# Token Sample Complexity of Attention

**Léa Bohbot**[1]  **Cyril Letrouit**[2]  **Gabriel Peyré**[3]  **François-Xavier Vialard**[4]

## Abstract

As context windows in large language models continue to expand, it is essential to characterize how attention behaves at extreme sequence lengths. We introduce token sample complexity: the rate at which attention computed on $n$ tokens converges to its infinite-token limit. We estimate finite-$n$ convergence bounds at two levels: pointwise uniform convergence of the attention map, and convergence of moments for the transformed token distribution. For compactly supported (and more generally sub-Gaussian) distributions, our first result shows that the attention map converges uniformly on a ball of radius $R$ at rate $C(R)/\sqrt{n}$, where $C(R)$ grows exponentially with $R$. For large $R$, this estimate loses practical value, and our second result addresses this issue by establishing convergence rates for the moments of the transformed distribution (the token output of the attention layer). In this case, the rate is $C'(R)/n^{\beta}$ with $\beta < \frac{1}{2}$, and $C'(R)$ depends polynomially on the size of the support of the distribution. The exponent $\beta$ depends on the attention geometry and the spectral properties of the token distribution. We also examine the regime in which the attention parameter tends to infinity and the softmax approaches a hardmax, and in this setting, we establish a logarithmic rate of convergence. Experiments on synthetic and real data support our predictions and show that the predicted slowdown is reflected in downstream accuracy.

## 1. Introduction

The remarkable success of Transformer models has been driven by attention mechanisms that flexibly handle very long input sequences (Vaswani et al., 2017; Devlin et al., 2019). Recent large-context Transformers extend context

---

[1]ENS, PSL Univ. [2]LMO, Univ. Paris-Saclay, CNRS [3]CNRS, ENS, PSL Univ. [4]LIGM, Univ. Gustave Eiffel, CNRS. Correspondence to: Léa Bohbot <lea.bohbot@ens.psl.eu>.

*Proceedings of the 43rd International Conference on Machine Learning*, Seoul, South Korea. PMLR 306, 2026. Copyright 2026 by the author(s).

windows from 100K to 10M tokens (Reid et al., 2024; Hooper et al., 2024), raising fundamental questions about how attention behaves in the regime of extremely large sequences, and how much benefit is gained by providing ever more tokens as input. In this work, we introduce the notion of *token sample complexity* to characterize the precision gained by feeding additional tokens into an already-trained Transformer. Unlike classical sample complexity, which studies performance as training data increases, we fix the Transformer parameters and examine a different asymptotic regime: for a fixed input distribution, we let the number $n$ of context tokens grow. Our goal is to understand how attention outputs behave in this limit, and how many tokens are required to approach the limiting behavior.

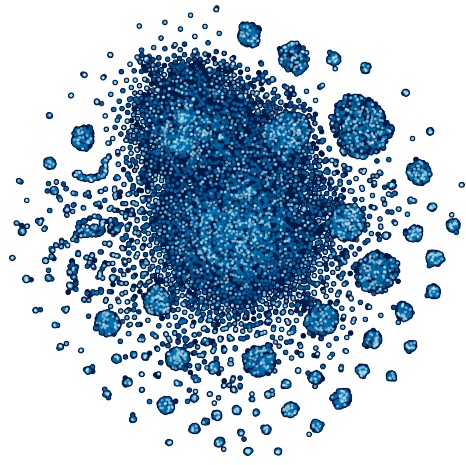

*Figure 1.* **Illustration of token distribution convergence toward a limit distribution** using t-SNE visualization of the first 50000 token embeddings from WikiText-103 (BigBird tokenizer). Colors (light/medium/dark blue) represent $n = 3000, 10000$, and $50000$ subsampled tokens: as $n$ grows, the token distribution $\nu_n$ covers the support more uniformly.

We formalize the infinite-token limit of a Transformer's self-attention as a continuous operator on probability measures. Following (De Bie et al., 2019; Pevny & Kovarik, 2019), we view a sequence of $n$ tokens as an empirical distribution $\nu_n$ that approximates an underlying token distribution $\nu$. Figure 1 illustrates this convergence, showing how $\nu_n$ approaches $\nu$ as $n$ grows. We define the attention mechanism in the limit of infinitely many tokens as an operator acting on $\nu$. This measure-theoretic perspective was pioneered by

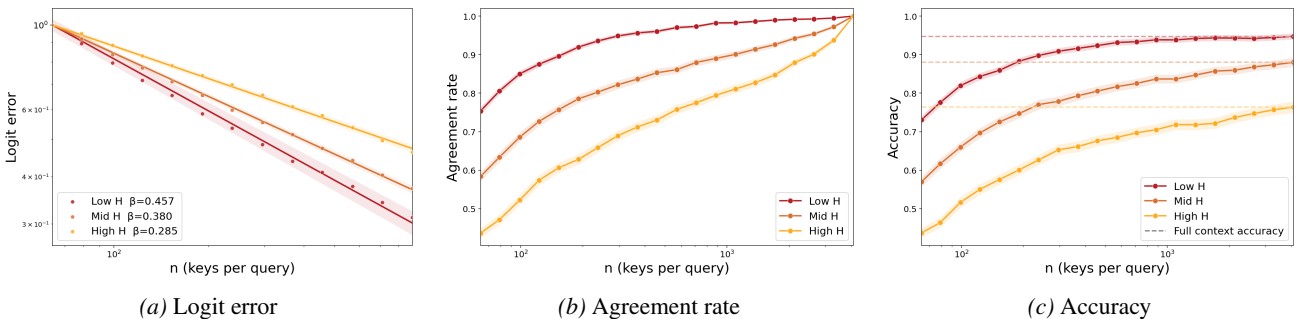

*(a)* Logit error         *(b)* Agreement rate         *(c)* Accuracy

*Figure 2.* **Downstream token sample complexity.** BigBird–RoBERTa–base is fine-tuned on arxiv-classification and evaluated by replacing dense attention with $n$ i.i.d. sampled keys per query. Curves are stratified by attention horizon $H$. Logit error, agreement with the dense model, and classification accuracy all show horizon-dependent convergence and diminishing returns as $n$ increases. Shaded bands indicate $\pm 3\sigma$ over Monte Carlo repetitions.

(Sander et al., 2022) and extended to study Transformers as interacting particle systems (Lu et al., 2019; Geshkovski et al., 2024; 2025; Castin et al., 2025; Burger et al., 2025; Bruno et al., 2025; Zimin et al., 2025; Chen et al., 2025).

We use this infinite-token limit to formalize *token sample complexity*: the rate at which finite-$n$ attention approaches its infinite-limit counterpart. With compactly supported token distributions, one naturally expects parametric convergence at rate $1/\sqrt{n}$. However, on both synthetic and real data, we observe slower sub-parametric rates $n^{-\beta}$ with $\beta < 1/2$. Our analysis identifies an effective *attention horizon* $H := \|\Sigma^{1/2} A \Sigma^{1/2}\|_2$ as a key quantity governing this slowdown, where $\Sigma$ is the covariance matrix of the token distribution, and $A := K^\top Q$ is the attention matrix, itself a product of the key $K$ and query $Q$ matrices. It can be interpreted as the largest query-key score attainable by two tokens in the unit ball of the covariance geometry (see Definition 5.1). When it grows large, convergence slows down, and when $H \to +\infty$, softmax attention converges to hardmax, in which case convergence is only logarithmic (see Section 5.4).

Figure 2 empirically connects our token sample complexity bounds to downstream errors on an end-to-end classification task. When dense attention is replaced by attention over $n$ sampled keys per query, the sparse model progressively recovers the dense model's logits and predictions as $n$ increases. Consistent with our theoretical analysis, this recovery is sub-parametric and visibly structured by the attention horizon: high-horizon regimes require more keys to approach dense-model behavior and reach lower accuracy. Thus, the slowdown identified by our analysis at the attention-output level propagates through the full network to downstream prediction quality. This yields a quantitative inference-time insight: in our experiment, using only 25% of keys already recovers most of the dense-model accuracy, while doubling the number of sampled keys beyond this point brings only a small additional gain (see 6.2).

More precisely, our key contributions are the following

- In Section 4, we quantify almost-sure convergence of the *attention map* itself at a given query $x$. Theorem 4.2 provides uniform convergence on balls $B_R$ for sub-Gaussian tokens, with explicit quantitative estimation of the rate.

- In Section 5, we study moment convergence, which matters for downstream tasks based on aggregate token statistics. Compact-support arguments yield $C(R)/\sqrt{n}$ rates, but with constants exponential in the radius $R$ and thus uninformative for large supports. Theorem 5.3 gives sharper sub-Gaussian rates $\mathcal{O}(n^{-\beta})$, $\beta < \frac{1}{2}$, with polynomial scale dependence governed by the attention horizon and token anisotropy, matching real-data behavior more closely. We also analyze the Gaussian hardmax limit, where Proposition 5.5 gives the exact logarithmic rate $1/\sqrt{\ln n}$.

- In Section 6, we empirically assess the tightness of bounds for Gaussian data and demonstrate their practical relevance on real text: using BigBird and BERT models on various long text inputs, we observe convergence rates strictly slower than $1/\sqrt{n}$, confirming that the sub-parametric regime governs attention behavior in practice. Finally, we show on an end-to-end classification task that the slowdown identified by our analysis is reflected in downstream behavior: sparse attention approaches dense inference at horizon-dependent sub-parametric rates.

## 2. Background and related work

**Classical PAC sample complexity.** In statistical learning theory, *sample complexity* refers to the number of training samples $N$ required for a learning algorithm to achieve a probably approximately correct (PAC) solution (Shalev-Shwartz & Ben-David, 2014). The classical question is: given i.i.d. examples of fixed dimension, how many samples are needed so that, with high probability, the learned model's error is within $\varepsilon$ of the optimum? Classical results show $N$ scales with hypothesis class complexity (VC dimension,

Rademacher complexity), but crucially assume each sample has fixed, finite dimension. This is the regime of standard PAC theory: we collect more fixed-size data points (images, sentences) to improve generalization.

**Token sample complexity (this work).** In our setting, we consider an entirely different kind of complexity question: the model is *already trained*, and we are not collecting new independent samples; instead, we are increasing the number of tokens in a single input sequence. Modern Transformer encoders like BERT process an input as a single sequence of tokens (e.g. words or subword tokens in a sentence) of length $n$. Importantly, $n$ is not fixed—the architecture allows variable-length input, and indeed self-attention inherently permits an unbounded number of tokens by design (Vaswani et al., 2017). Thus, we can think of $n$ as a parameter of the problem instance, akin to an input dimension that can grow.

This inference-time perspective aligns with the growing use of extended contexts at inference. In-context learning and chain-of-thought prompting rely on providing Transformers with sequences that are often much longer than those seen during training (Brown et al., 2020; Zhang et al., 2023; Sander et al., 2024; Huang et al., 2024; Zhang et al., 2025; Wei et al., 2022). As these approaches push the sequence length $n$ beyond training-time regimes, understanding how attention accuracy scales with $n$ at inference becomes increasingly important.

**Transformers and self-attention.** A Transformer relies on attention mechanisms to model interactions between tokens. Each encoder layer applies self-attention, where output vectors $z_i$ are computed as weighted combinations of value vectors based on token affinities (see Definition 3.1), followed by a feed-forward network. Unlike CNNs or RNNs, self-attention imposes no intrinsic limitation on sequence length $n$: arbitrarily many tokens can be processed. This ability to capture long-range interactions comes at quadratic cost $\mathcal{O}(n^2)$, motivating efficient attention variants.

**Sparse attention and efficient variants** Our results also give insight into the error created when the input tokens are randomly sub-sampled, which can be viewed as a simple model of attention sparsification. In practice, more advanced sparsification strategies are used, such as restricting tokens to local neighborhoods as in Longformer (Beltagy et al., 2020) and BigBird (Zaheer et al., 2020), or explicitly projecting keys and values to lower-dimensional subspaces as in Linformer (Wang et al., 2020). Although our findings do not cover those methods, they offer an initial step toward a theoretical understanding of such approaches.

**Transformers with arbitrarily many tokens.** The mean field model which corresponds to defining attention as operating over a probability distribution of tokens was initially presented in (De Bie et al., 2019) and (Pevny & Kovarik, 2019), and subsequently adapted to Transformers in (Vuckovic et al., 2020; Sander et al., 2022; Geshkovski et al., 2024; 2025). Building on this framework, (Geshkovski et al., 2024), (Geshkovski et al., 2025) and (Castin et al., 2025) develop a *Transformer PDE* for standard softmax attention and prove it is the mean-field limit of an $n$-particle system as $n \to \infty$. (Geshkovski et al., 2025) models tokens as points evolving on the *sphere* $\mathbb{S}^{d-1}$ to capture normalization effects and connects the resulting dynamics to models from mathematical physics. Related developments include (Karagodin et al., 2024) (clustering with causal masking) and (Bruno et al., 2025) (meta-stable clustering). Notably, (Castin et al., 2025) prove that the Transformer PDE preserves Gaussianity, reducing the dynamics to mean and covariance evolution. In contrast, we focus on a single attention layer in the infinite-token limit, using the Gaussian case as a tractable baseline to derive explicit convergence rates and motivate extensions to sub-Gaussian distributions.

**Relation to (Boursier & Boyer, 2025)** In an independent work, (Boursier & Boyer, 2025) also study token sample complexity. A first key difference with our work is that we consider the setting in which queries, keys, and values are drawn from the same token set, as is typically the case when modeling self-attention. This dependence between tokens makes the analysis substantially more involved, and requires the use of the chaining method in the proof of Theorem 4.2; see Section 4.3 for details. We also establish $O(1/n^\beta)$ error bounds with smaller values of $\beta$; in particular, when $A \to 0$ we recover the exponent $\beta = 1/2$, and we derive exact asymptotics in the hardmax limit $A \to \infty$. On the other hand, (Boursier & Boyer, 2025) establish concentration bounds for the Jacobian of the attention map, with applications to in-context linear regression under Gaussian inputs.

## 3. Mathematical preliminaries

### 3.1. Self-attention

Let $d \in \mathbb{N}$ be the hidden dimension of tokens embeddings that live in $\mathbb{R}^d$, with inner product $\langle \cdot, \cdot \rangle$ and $\| \cdot \|_2$ the Euclidean norm for vectors and the associated operator norm.

**Definition 3.1** (Single-head self-attention). Let $k, d \in \mathbb{N}$. Let $Q, K, V \in \mathbb{R}^{k \times d}$ the query, key and value matrices and set $\tilde{A} := K^\top Q / \sqrt{k} \in \mathbb{R}^{d \times d}$ and $A := K^\top Q \in \mathbb{R}^{d \times d}$ (where scaling is absorbed in either $K$ or $Q$). Given an $n$-sequence $X = (x_i)_{1 \leqslant i \leqslant n}$, embedded in dimension $d$, the single-head self-attention function $f_n : \mathbb{R}^d \to \mathbb{R}^k$ maps each token $x_i$ of the sequence to

$$f_n(x_i) = \frac{\sum_{j=1}^n e^{\langle Ax_i, x_j \rangle} V x_j}{\sum_{j'=1}^n e^{\langle Ax_i, x_{j'} \rangle}}, \tag{1}$$

which corresponds to a softmax operation defined for a vector $w \in \mathbb{R}^d$ by $\text{SoftMax}(w) := \left(\exp(w_i)/\sum_{j=1}^{n}\exp(w_j)\right)_{1 \leqslant i \leqslant n}$, and applied here for $w = \left(\langle Ax_i, x_j\rangle\right)_{1 \leqslant i \leqslant n}$. Note that $f_n$ depends on the entire context $X = (x_i)_{1 \leqslant i \leqslant n}$. For notational simplicity, we encode this dependence only through the index $n$, writing $f_n$ instead of the more precise $f_{(x_i)_{1 \leqslant i \leqslant n}}$ or $f_{\nu_n}$ when $X \sim \nu_n$ (see Section 5.1 for details on these notations).

All our results naturally extend to multihead self-attention (see Definition A.1) and all experiments are conducted in the multihead self-attention setting.

**Permutation equivariance and measure modeling.** Self-attention without positional information is equivariant to reindexing: for any permutation $\sigma$ of $\{1, \ldots, n\}$, if $(x_1, \ldots, x_n) \sim \nu_n$, then $(x_{\sigma(1)}, \ldots, x_{\sigma(n)}) \sim \nu_n$ and for every token $x$,

$$f_{(x_i)_{1 \leqslant i \leqslant n}}(x) = f_{(x_{\sigma(i)})_{1 \leqslant i \leqslant n}}(x)$$

Hence, one may represent the sequence via its empirical distribution $\nu_n := \frac{1}{n}\sum_{i=1}^{n}\delta_{x_i}$, and analyze attention at the level of probability measures. This naturally leads to formalizing the infinite-token limit by viewing attention as an operator on such measures.

**Definition 3.2** (Mean-field self-attention). Let $\mathcal{P}(\mathbb{R}^d)$ be the set of Borel probability measures. For $\nu \in \mathcal{P}(\mathbb{R}^d)$ and $x \in \mathbb{R}^d$, define

$$f(x) := \frac{\int_{\mathbb{R}^d} e^{\langle Ax, y\rangle}\, V\, y\, d\nu(y)}{\int_{\mathbb{R}^d} e^{\langle Ax, y\rangle}\, d\nu(y)} \in \mathbb{R}^k$$

When $\nu = \nu_n$, $f$ is equal to the empirical attention map $f_n$. As for $f_n$, $f$ depends on the context distribution $\nu$. As we fix $\nu$ in our work, we omit this dependence to alleviate notations.

**Reduction to the centered case.** All our results are stated for centered sub-Gaussian distributions, but they extend straightforwardly to non-centered sub-Gaussian distributions. The only modification in the main results is that the expectation of the distribution shows up in the bounds (in an explicit and controlled way).

# 4. Convergence of the attention map

In this section, we examine how fast $\left\|f_n(x) - f(x)\right\|_2$ converges to zero. We provide a quantitative uniform convergence rate over all tokens in a ball of radius $R$, highlighting the dependence of the estimated rate on the attention parameters and the token distribution properties.

A uniform—rather than simply pointwise (see Appendix D.2)—convergence result is required to translate convergence of the attention map into results on moments of the transformed distribution.

In the self-attention setting, the attention map itself is computed from the same set of tokens on which it is subsequently applied: the queries, keys, and values all originate from a common context. This induces a nontrivial autocorrelation structure, as the randomness governing the attention weights and that of the aggregated outputs are no longer independent. When the context size grows, moment estimates therefore involve aggregating attention outputs across a continuum of queries while the underlying attention operator simultaneously depends on the entire token set. As a consequence, the operator must be controlled uniformly over all queries $x$ in the relevant region of space: a single poorly controlled query can dominate the aggregate and ruin the global behavior of the attention operator across the distribution. In this sense, uniform control becomes indispensable to propagate convergence results to moment-level statements (see Appendix C.6 for details).

## 4.1. Sub-Gaussian Model

In transformer architectures, normalization layers (e.g. LayerNorm or RMSNorm) keep tokens bounded, hence sub-Gaussian. The sub-Gaussian framework therefore goes beyond compact-support assumptions, while also including important unbounded cases such as Gaussian distributions (Castin et al., 2025). More importantly, its parameter $\Sigma$ captures token anisotropy, yielding more informative convergence bounds in the regimes observed experimentally. We use the standard log-MGF characterization of sub-Gaussian random vectors (Boucheron et al., 2013).

**Definition 4.1** (Sub-Gaussian vector). A centered random vector $X \in \mathbb{R}^d$ with distribution $\nu$ is *sub-Gaussian with parameter matrix* $\Sigma \succ 0$ if $\forall t \in \mathbb{R}^d, \mathbb{E}[\exp\{\langle t, X\rangle\}] \leqslant \exp\left(\frac{1}{2}t^\top \Sigma t\right)$.

If no prior information is available aside from a compact support $\text{supp}(\nu) \subset B(0, R)$, one can take $\Sigma = R^2 \text{Id}$.

## 4.2. Uniform Convergence Result

Establishing an explicit quantitative estimation of the uniform convergence (Theorem 4.2) is challenging. First, the attention map is a ratio of empirical averages whose kernel $e^{\langle Ax, y\rangle}$ is not uniformly Lipschitz, leading to a function class $\mathcal{F} = \{y \mapsto ye^{\langle Ax, y\rangle}, x \in B_R\}$ with unbounded envelope $F(y) = \|y\|_2 e^{R\|A^\top y\|_2}$, outside the scope of standard empirical process theory.

Furthermore, to obtain explicit parameter-dependent bounds, we use Dudley's entropy integral and covering estimates, with a duality argument to handle the vector-valued empiri-

cal process.

Theorem 4.2 quantifies how the interaction between the token distribution geometry $\Sigma$ and the model parameter $A$ influences the convergence of the attention map. The key novelty and technical challenge of this result lies in capturing the behavior of SoftMax mappings, which are not uniformly Lipschitz. This prevents the use of standard uniform central limit theorems and requires a dedicated analysis.

**Theorem 4.2** (Uniform convergence of the attention map for sub-Gaussian tokens). *Let $X \sim \nu$ be centered and sub-Gaussian with parameter matrix $\Sigma \succ 0$. For any $R > 0$, $\delta > 0$, there exists a constant $C > 0$, such that for $n \geqslant n_{min}(\delta, \Sigma, A, R) := 4e^{2R^2 \|\Sigma^{1/2} A\|_2^2} \left( \frac{1}{\delta} - 1 \right)$, with probability at least $1 - \delta$,*

$$\sup_{x \in B_R} \left\| f_n(x) - f(x) \right\|_2 \leqslant q_{(\Sigma, A, V, R, d, \delta)} \cdot \frac{e^{8\,R^2 \|\Sigma^{1/2} A\|_2^2}}{\sqrt{n}},$$

(2)

*where* $q_{(\Sigma, A, V, R, d, \delta)} := \frac{C\sqrt{d}}{\delta} \|V\|_2 \cdot \|\Sigma\|_2^{1/2} \cdot (R\|A\|_2 \|\Sigma\|_2^{1/2} d + \mathrm{tr}(\Sigma)^{1/2} \|A\|_2 + 5^{d/2}\sqrt{d})$.

See Appendix C.2 for full proof, and Appendix D.8.1 for a specific estimate for Gaussians. Note that these rates are not sharp in full generality. For instance, in low dimension, faster asymptotic rates are expected; Appendix C.5 shows $1/\sqrt{n}$ rates in the one-dimensional case.

### 4.3. Sketch of proof of Theorem 4.2

The proof of Theorem 4.2 controls the numerator and denominator separately before recombining them uniformly in $x$, since the empirical denominator may approach zero and precludes a direct uniform delta-method argument. We first prove an expectation bound, then derive high-probability bounds.

Since the numerator and denominator follow the same proof structure, we detail only the numerator. The only additional difficulty is its vector-valued nature ($\mathbb{R}^d$), requiring a technical adaptation of Dudley's theorem to the multidimensional setting (see Step 2).

We reformulate the problem as the uniform convergence of a supremum over a parametric class of *unbounded Lipschitz functions*, where the parameter lies in a compact set. Define the function class $\mathcal{F} := \{f_x : x \in B_R\}$, where $f_x : y \mapsto y\,e^{\langle Ax, y \rangle} \in L^2(\nu; \mathbb{R}^d)$ (see Section B for empirical process theory's details). Our goal is to bound the uniform deviation $\|\mathbb{P}_n - \mathbb{P}\|_{\mathcal{F}} := \sup_{x \in B_R} \left\| \frac{1}{n} \sum_{i=1}^n f_x(Y_i) - \mathbb{E}[f_x(Y)] \right\|_2$. The proof of this intermediate result proceeds in two steps.

**Step 1: Symmetrization.** We first relate $\mathbb{E}[\|\mathbb{P}_n - \mathbb{P}\|_{\mathcal{F}}]$ to the Rademacher complexity of $\mathcal{F}$. Introducing an i.i.d.

ghost sample $(X_1, \ldots, X_n)$ and using Jensen's inequality, one shows that $\mathbb{E}[\|\mathbb{P}_n - \mathbb{P}\|_{\mathcal{F}}] \leqslant 2\,\Re_n(\mathcal{F})$, where

$$\Re_n(\mathcal{F}) := \mathbb{E}\Big[ \sup_{x \in B_R} \big\| \frac{1}{n} \sum_{i=1}^n \varepsilon_i f_x(Y_i) \big\|_2 \Big].$$

**Step 2: Bounding the Rademacher complexity.** To bound $\Re_n(\mathcal{F})$, we apply Dudley's entropy integral result (Theorem B.5). Since the process $Z_x := \frac{1}{n} \sum_{i=1}^n \varepsilon_i f_x(Y_i)$ is vector-valued, we use the dual norm representation $\|Z_x\|_2 = \sup_{z \in B_1} \langle z, Z_x \rangle$, and consider the scalar process $Z_{(z,x)} := \langle z, Z_x \rangle$, for $(z, x) \in \mathcal{G} := B_1 \times B_R$.

One verifies that $Z_{(z,x)}$ is sub-Gaussian with respect to an appropriate empirical metric $d_{\mathcal{G}}$ on $\mathcal{G}$. Dudley's theorem then yields

$$\mathbb{E}_\varepsilon \left[ \sup_{(z,x) \in \mathcal{G}} Z_{(z,x)} \right] \lesssim \frac{1}{\sqrt{n}} \int_0^{\mathrm{diam}(\mathcal{G})} \sqrt{\ln \mathcal{N}(\delta, \mathcal{G}, d_{\mathcal{G}})}\, d\delta.$$

To evaluate this integral, we bound three key quantities: the moments of the envelope function $F$ (Lemma C.8), the Lipschitz constant of functions in $\mathcal{F}$, (Lemmas C.7 and C.10), and the covering number $\mathcal{N}(\varepsilon, \mathcal{G}, d_{\mathcal{G}}((z_1, x_1), (z_2, x_2)))$ of $\mathcal{G}$ (Lemmas C.1 and C.2).

Combining these bounds and integrating yields a control of $\mathbb{E}[\|\mathbb{P}_n - \mathbb{P}\|_{\mathcal{F}}]$. A direct application of Markov's inequality then converts this expectation bound into the high-probability result of Proposition C.5.

Applying Steps 1–2 to the scalar function class $\mathcal{F}' := \{h_x : y \mapsto e^{\langle Ax, y \rangle}\}$ yields Proposition C.9, with a simpler analysis in dimension 1.

**Step 3: Combining the bounds.** Write $N_n$ and $D_n$ for the empirical numerator and denominator, and $N$ and $D$ for their expectations. The ratio error decomposes as

$$\left\| \frac{N_n}{D_n} - \frac{N}{D} \right\|_2 \leqslant \frac{\|N_n - N\|_2}{D_n} + \frac{\|N\|_2\,|D_n - D|}{D\,D_n}.$$

Applying Jensen's inequality, $D = \mathbb{E}[e^{\langle Ax, Y \rangle}] \geqslant 1$. Then, a control the deviation of $D_n$ around $D$ along with the application of Propositions C.5 and C.9 concludes the proof.

## 5. Convergence of the transformed distribution

### 5.1. Moments convergence

So far, we have analyzed the convergence of the attention map $\|f_n(x) - f(x)\|_2$ at a single query $x$ in a fixed compact set. However, it is also important to study the convergence of the output distribution itself, which corresponds to the self-attention setting (where the attention map is applied to the tokens themselves). Let $X$ (resp. $\hat{X}$) be a random

vector with distribution $\nu$ (resp. $\nu_n$). We therefore consider the convergence of the distribution $(f_n)_\sharp \nu_n$ of $f_n(\hat{X})$ toward the distribution $f_\sharp \nu$ of $f(X)$, where $f_\sharp$ denotes the push-forward operator. The convergence of the output distribution is particularly relevant for downstream tasks that rely on aggregate token statistics, such as classification or sequence-level predictions. Moreover, in deep transformer architectures, attention layers are stacked, so that the inputs to subsequent layers depend on the entire reweighted distribution produced by earlier attention mechanisms. Understanding distributional convergence is therefore crucial for analyzing error propagation across network depth.

A natural way to quantify this convergence is through moments, such as the mean, second-order statistics, and MSE, which capture aggregate properties of attention outputs used in downstream operations like average pooling, token aggregation, and sequence-level representations. We consider moments defined as expectations of functions $h : \mathbb{R}^d \to \mathbb{R}^{d'}$ applied to the attention outputs, and denote by $\mathbb{E}_n[h \circ f_n(\hat{X})]$ (resp. $\mathbb{E}[h \circ f(X)]$) the empirical (resp. exact) moments. Taking $h(x) = x$ recovers the mean; more generally, we focus on Lipschitz functions $h$, useful for bounding Fourier moments which is equivalent to controlling translation-invariant maximum mean discrepancies (MMD) (Gretton et al., 2012). We also extend this to quadratic functions to capture covariance.

Corollary D.6 broadens our results by controlling the MSE $\mathbb{E}\|f_n(X) - f(X)\|_2^2$, which measures errors at the level of individual token embeddings. Corollary D.7 further extends the theory to deep compositions of layers.

## 5.2. Sub-Gaussian Moment Convergence

For compactly supported token distributions ($\mathrm{supp}(\nu) \subset B(0,R)$) our uniform pointwise convergence result (Theorem 4.2) directly implies analogous rates of order $O(C(R)/\sqrt{n})$ for the convergence of distribution moments. However, the resulting constant $C(R)$ grows exponentially with $R$, which makes these bounds essentially useless in practice. Moreover, this analysis does not apply to the refined sub-Gaussian model introduced in Section 4.1, and therefore cannot account for the anisotropy of the token distribution encoded by $\Sigma$. We therefore pursue a more refined approach, aiming to obtain slower rates of the form $O(C'/n^\beta)$ with $\beta < 1/2$, but where the constant $C'$ depends only mildly on $\Sigma$. As illustrated by our numerical experiments, these slower rates are consistent with empirical observations. The rate $\beta$ depends on both the attention parameter and the token anisotropy $\Sigma$ through the attention horizon $H$ (see Definition 5.1 below). In the weaker setting where one only assumes compact support of the token distribution, we derive analogous rates in Appendix D.7.

**Definition 5.1** (Token-parameters coupling coefficient). Let

$X$ be centered and sub-Gaussian with parameter matrix $\Sigma \succ 0$. Recall $A = K^\top Q$, The *Attention Horizon H* is defined as

$$H := \left\| \Sigma^{1/2} A \Sigma^{1/2} \right\|_2,$$

where $\| \cdot \|_2$ denotes the operator norm.

*Remark* 5.2 (Intuition for attention horizon $H$). Since the tokens have covariance $\Sigma$, their natural geometry is the Mahalanobis geometry $\|x\|_{\Sigma^{-1}}^2 = x^\top \Sigma^{-1} x$. This motivates the definition of the horizon, which can be interpreted as the largest possible query-key score between two tokens in the unit ball of the covariance geometry:

$$H = \max_{\|x\|_{\Sigma^{-1}} \leqslant 1, \; \|y\|_{\Sigma^{-1}} \leqslant 1} \langle Qx, Ky \rangle.$$

Intuitively, the horizon $H$ quantifies the effective range of attention: how distant token keys can influence a given query, as determined by token geometry ($\Sigma$) and attention weights ($A$).

Theorem 5.3 establishes that the non-asymptotic convergence rate of attention moments is sub-parametric, with an exponent $\beta < 1/2$ that depends on $H$. Its main contribution is to provide, for the first time, a precise and quantitative characterization of how sparse attention mechanisms become harder to approximate with a limited number of tokens. The resulting bounds are not only tight on synthetic data, but are also supported by experiments on real data, where we show that $H$ governs the approximation rate even for end-to-end downstream tasks. This yields concrete and quantitative insight into the number of tokens required per layer to achieve a desired level of accuracy.

**Theorem 5.3** (Lipschitz functional convergence rate for sub-Gaussian tokens). *Let $X \sim \nu$ be centered and sub-Gaussian with parameter matrix $\Sigma \succ 0$, and let $\nu_n$ be the empirical measure associated with $n$ i.i.d. samples from $\nu$. Denote $\hat{X} \sim \nu_n$. Let $h$ be a $L_0$-Lipschitz function, squared integrable with respect to $\nu$. Let us denote by $\mathbb{E}$ the expectation w.r.t. $\nu$, and $\mathbb{E}_n$ the expectation w.r.t. $\nu_n$. For $n$ i.i.d. tokens, with $n \geqslant 4\left(\frac{1}{\delta} - 1\right)n^{1/8}$, with probability at least $1 - \delta$,*

$$\left\| \mathbb{E}_n[h \circ f_n(\hat{X})] - \mathbb{E}[h \circ f(X)] \right\|_2 \leqslant \frac{P(\sqrt{\ln n}, L_0)}{n^\beta},$$

*where $P$ is a polynomial function, and $\beta := \frac{1}{2(1 + 32 \cdot H^2)}$*

*Remark* 5.4 (Sharpness of the slow-rate result). Experiments confirm the predicted horizon-dependent slowdown; in the hardmax regime, Section 5.4 proves tight asymptotics with an even stronger slowdown.

See Appendix C.6 for proof, and Appendix Section A for a discussion on the i.i.d. hypothesis.

## 5.3. Sketch of proof for Theorem 5.3

**Step 1: Error decomposition.** By the triangle inequality, we decompose the error into two terms in order to bound each separately

$$\left\|\mathbb{E}_n[h \circ f_n(\hat{X})] - \mathbb{E}[h \circ f(X)]\right\|_2 \leqslant \mathcal{I} + \mathcal{J},$$

$$\text{where} \quad \mathcal{I} := \left\|\mathbb{E}_n[h \circ f_n(\hat{X}) - h \circ f(X)]\right\|_2,$$

$$\mathcal{J} := \left\|\mathbb{E}_n[h \circ f(X)] - \mathbb{E}[h \circ f(X)]\right\|_2.$$

The term $\mathcal{J}$ captures the sampling error of the population attention, while $\mathcal{I}$ captures the approximation error from using finite context. The term $\mathcal{J}$ is straightforward to bound, so we will focus on establishing the bound for $\mathcal{I}$.

**Step 2: Bounding $\mathcal{I}$ (approximation error).** The key idea is that sub-Gaussian concentration confines most tokens to an *effective radius*, slowly growing with $n$ and the attention scale, while large-norm tokens are exponentially rare.

We split the error between tokens inside the ellipsoid $\|Bx_i\|_2 \leqslant R$ and those outside it, where $B$ is an auxiliary invertible matrix optimized later and get: $\mathcal{I} \leqslant \mathcal{I}_1 + \mathcal{I}_2 + \mathcal{I}_3$.

Here $\mathcal{I}_1$ is the inside contribution, while $\mathcal{I}_2$ and $\mathcal{I}_3$ control the tail contributions for $f_n$ and $f$, respectively. On $K_B(R) := \{x : \|Bx\|_2 \leqslant R\}$, the Lipschitz property of $h$ and Lemma C.11 yield

$$\mathcal{I}_1 \leqslant L_0 \cdot \sup_{x \in K_B} \|f_n(x) - f(x)\|_2 = O\Big(\frac{e^{8R^2\|\Sigma^{1/2}AB^{-1}\|_2^2}}{\sqrt{n}}\Big).$$

For the tail terms $\|Bx_i\|_2 > R$, the attention map satisfies $\|f_n(x)\|_2 \leqslant M_n := \max_i \|x_i\|_2$, since $f_n$ is a convex combination of token embeddings; the analogous bound for $f$ follows from the importance-sampling form of attention in Lemma D.1. Hoeffding's inequality controls the tail proportion $N_R/n$, while a union bound with sub-Gaussian tails controls $M_n$ (see Proposition C.14), yielding $\mathcal{I}_2 + \mathcal{I}_3 = O\Big(e^{-cR^2/\|\Sigma^{1/2}B^\top\|_2}\Big).$

**Step 3: Optimizing over $R$.** Thus, $\mathcal{I}$ is bounded by two competing terms: the compact-region term $\mathcal{I}_1$, which grows like $\exp(cR^2)$ through uniform convergence, and the tail terms $\mathcal{I}_2 + \mathcal{I}_3$, which decay like $\exp(-cR^2)$ by sub-Gaussian concentration. Balancing them with $R^\star = \Theta\left(\sqrt{\frac{\ln n}{\|\Sigma^{1/2}AB^{-1}\|_2^2 + \|\Sigma^{1/2}B^\top\|_2^{-1}}}\right)$ and optimizing over invertible $B$ gives the final rate.

## 5.4. Hardmax regime

To complete the analysis, we study the infinite-horizon, or hardmax, regime and derive exact asymptotics. This limit is relevant to real Transformers, where large matrix norms can make softmax behave like a hard-max selector. In the simplified one-dimensional Gaussian case, when $\|A\sigma\|_2 \gg 1$, attention concentrates on extreme tokens, so the empirical output reduces to a combination of the sample maximum and minimum. This phenomenon leads to a remarkable cancellation effect that makes the convergence rate governed by Gaussian extreme value theory.

Let $\bar{f}_n := \frac{1}{n}\sum_{i=1}^n f_n(X_i)$, the empirical mean of the attention outputs.

**Proposition 5.5** (Hardmax convergence rate). *As $n \to \infty$, taking the expectation w.r.t. $X_1, ..., X_n$ :*

$$\mathbb{E}[|\bar{f}_n|] = \frac{\ln(4)\,\sigma}{2\sqrt{2\ln n}}[1 + o(1)], \mathbb{E}[\bar{f}_n^2] = \frac{\pi^2\,\sigma^2}{24\ln n}[1 + o(1)]$$

*Therefore, the expectation decreases exactly at the rate $\sigma/\sqrt{\ln n}$.*

**Cancellation phenomenon** In the wide-horizon regime, softmax behaves like a hard maximum that focuses on extremes. For Gaussian inputs, the maximum and minimum are asymptotically $\pm\sqrt{2\ln n}$, so the mechanism selects either one with roughly equal probability as $n$ grows, and their contributions cancel, yielding a mean near zero. This symmetry-driven cancellation is specifically Gaussian and implies a slow decay of order $\sigma/\sqrt{\ln n}$, much slower than any fixed polynomial rate. Observe that real token embeddings (e.g., in BigBird) do not display the above saturation. Our interpretation is that their distributions can easily deviate from Gaussian symmetry through skewness, kurtosis, anisotropy, or dependence for example. These departures disrupt the tail balance, removing the cancellation and the characteristic $1/\sqrt{\ln n}$ behavior.

*Remark* 5.6 (Heavy-tailed distributions). The sub-Gaussian assumption is necessary for hardmax attention to converge; Appendix C.9 gives a heavy-tailed counterexample where convergence fails.

# 6. Experiments

We validate our rates on Gaussian data, real tokens from BigBird–RoBERTa–base (Zaheer et al., 2020) and BERT, and an end-to-end classification task showing that the slowdown propagates to downstream quality. Code is available at https://github.com/leabbt/token-sample-complexity. We use the following experimental protocol common to all experiments. See Appendix Section A for additional details.

**Token distribution $\nu$** We study how the empirical mean and covariance of attention outputs, computed from $n$ sampled tokens, converge to their limiting values. For synthetic data, the limiting distribution $\nu$ is known; for real

datasets, we approximate it by the discrete distribution $\nu = \sum_{i=1}^{N_{\max}} \delta_{x_i}$ over a large token set, and form $\nu_n$ by sampling $n \ll N_{\max}$ i.i.d. tokens.

**Convergence rate computation** Our Monte Carlo protocol estimates how fast finite-token attention approaches its limiting distribution. For each subsample size $n \in [n_{\min}, n_{\max}]$, we uniformly draw $n$ tokens from $\nu$, compute the corresponding attention outputs, and measure the errors of their empirical mean and covariance relative to the limiting values. We take $N_{\max} \gg n_{\max}$ so that $\nu$ accurately approximates the limit, while $n_{\max}$ is large enough for convergence to be visible. Rates are obtained by fitting the error against $n$ on a log–log scale.

### 6.1. Tight bounds for Gaussian data

For synthetic experiments, we take the limiting token distribution to be $\nu = \mathcal{N}(0, \Sigma)$. For fixed $\Sigma$, we estimate the empirical rate $\hat{\beta}$ by Monte Carlo simulations and compare it, across varying horizons $H := \|\Sigma^{1/2} A \Sigma^{1/2}\|_2$, to the Gaussian prediction $\beta_{\text{th}} = \frac{1}{2(1+H^2)}$ from Proposition D.23.

Figure 3 shows strong agreement between the empirical rates and the prediction of Corollary D.4 over a wide range of horizons, suggesting that the bound captures the correct rate in this Gaussian setting. For larger $H$, the mean error becomes too slow to be reliably fitted by a polynomial rate, marking the transition toward the large-horizon regime analyzed in Section 5.4. Additional high-dimensional experiments, for both Gaussian tokens and a non-Gaussian sub-Gaussian distribution, are reported in Appendix D (Figures 7 and 8).

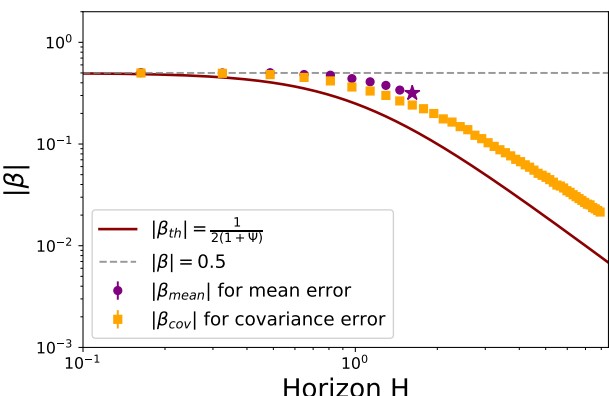

*Figure 3.* **Empirical and theoretical convergence rates** $|\hat{\beta}|$ **and** $|\beta_{th}|$ **vs horizon** $H$ **for Gaussian token** embeddings $\mathcal{N}(0, \Sigma)$ in dimension 4 with $\Sigma = 0.1 \cdot I_4$. Gray dotted line indicates the classical rate exponent $|\beta_0| = 0.5$ corresponding to the parametric rate in $1/\sqrt{n}$.

### 6.2. Slow rates on real data

**Model.** We use BigBird–RoBERTa–base (Zaheer et al., 2020), a sparse-attention Transformer with 12 layers, 768

hidden dimensions, and 12 heads. Positional interpolation (Chen et al., 2023) extends its 4096-token context to sequences of up to $N_{\max} \approx 500{,}000$, from which we construct $\nu$. We repeat the experiment on BERT to verify that the slowdown is not specific to the sparsity of the BigBird model (Appendix D).

**Text distributions.** To study the impact of textual structure on attention convergence, we evaluate seven datasets drawn from Wikipedia English, German, and Chinese ((Lhoest et al., 2021)), as well as CC-News ((Nagel, 2016)). These datasets are chosen to probe the effects of domain shift, language, and syntactic structure on convergence behavior.

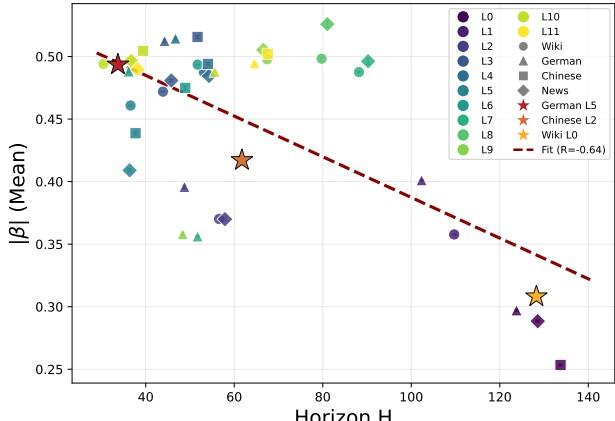

*Figure 4.* **Mean convergence rate** $|\beta|$ **versus horizon** $H$ across BigBird's layers 0–11 and all text distributions. Markers represent (layer, text configuration) pairs; shapes denote text sources (Wikipedia EN/DE/ZH and CC-News) and colors indicate layers. Larger $|\beta|$ implies faster convergence. Star markers denote the configurations whose detailed convergence curves are displayed in Figure 5.

As shown in Figure 4, convergence rates deteriorate systematically as the horizon $H$ increases. The rates also vary across text distributions at a fixed layer, reflecting the dependence of our bounds on the token geometry through $\Sigma$. In particular, $H = \|\Sigma^{1/2} A \Sigma^{1/2}\|_2$ does not measure anisotropy alone, but the interaction between the token distribution and the attention parameter $A$: convergence slows when the directions emphasized by attention align with high-variance directions of the token distribution. These results therefore reflect the combined influence of the model and the data. Analogous covariance results are reported in Figure 9 of Appendix D.

Figure 5 zooms in on the star-marked configurations. The linear behavior in log–log scale confirms a clear power-law convergence regime. Differences in slope across the three curves reveal the combined effect of layer depth and dataset anisotropy on convergence rates.

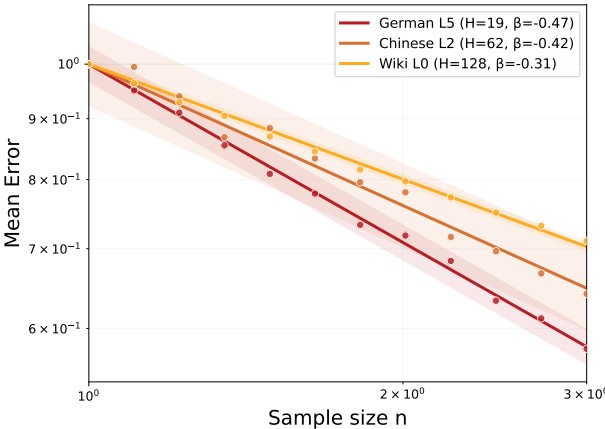

*Figure 5.* **Mean error convergence versus subsample size** $n$**.** For the starred BigBird-RoBERTa configurations in Figure 4, we plot $\|\mathbb{E}_n[f_n(x)] - \mathbb{E}[f(x)]\|_2$ over $k = 500$ subsamples on German Wikipedia. Log–log fits give exponents $|\beta|$, with error scaling as $n^{-|\beta|}$; larger horizons yield slower convergence.

**Window-based sampling.** We also examine whether the horizon-dependent slowdown persists under structured key sampling. Keeping the same experimental protocol as above, we replace uniform i.i.d. subsampling with two window-based rules: a local window around each query supplemented with $r = 50$ uniformly sampled out-of-window keys, and a hybrid rule that splits the key budget evenly between local-window and out-of-window random keys. These patterns are closer to practical sparse-attention mechanisms such as BigBird. We again observe sub-parametric convergence (see Figure 11b and Figure 11a in Appendix), with rates deteriorating as the horizon increases, suggesting that the slowdown is not specific to i.i.d. subsampling but also appears under structured, window-based sparsification.

**Downstream classification task** We finally test whether the finite-token convergence behavior observed at the attention-output level is reflected in end-to-end prediction quality. We fine-tuned BigBird–RoBERTa–base for 3 epochs on the arxiv-classification task (11 categories, dense accuracy of 86.4%) using mean pooling, and evaluated inference with $n$ i.i.d. sampled keys per query, for $n$ ranging from 64 to 4096 ($K = 5$ Monte Carlo repetitions, over 2400 test documents). We compare sampled-key and dense inference through downstream errors, prediction agreement, defined as the fraction of examples where sparse and dense predictions agree, and classification accuracy. Figure 6 shows that the error metrics follow sub-parametric power laws $n^{-\beta}$ with $\beta < \frac{1}{2}$, showing that the slowdown identified at the attention-output level also propagates through the full network to downstream prediction quality. When test examples are stratified by attention horizon, high-horizon groups converge more slowly, both in downstream errors and in the recovery of dense-model accuracy (see Figure 2). The sub-parametric convergence of the error curves can translate

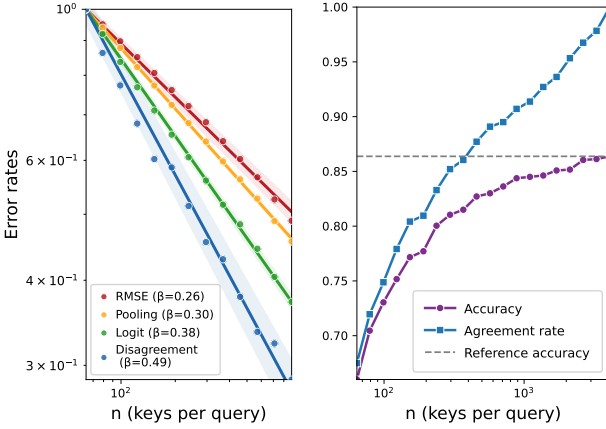

*(a)* Downstream errors.     *(b)* Accuracy/agreement.

*Figure 6.* **Downstream token sample complexity.** BigBird–RoBERTa–base is evaluated on arxiv-classification by replacing dense attention with attention over $n$ i.i.d. sampled keys per query. Downstream errors decrease sub-parametrically (RMSE/pooling/logit/disagreement exponents: $\beta = 0.27, 0.30, 0.37, 0.46$), while agreement rate and accuracy approach the dense model with diminishing returns.

into quantifiable insight into downstream accuracy: with only 25% of keys ($n = 1101$), the model already achieves 84.5% accuracy, and doubling $n$ from 1101 to 2124 yields only $+0.7\%$ gain.

## 7. Conclusion

In this paper, we introduced the notion of *token sample complexity* to characterize the convergence of the attention outputs as the sequence length $n$ increases. While parametric convergence rates can be derived for compactly supported token distributions, these rates are asymptotic and fail to describe the finite-context regimes encountered in practice. Encompassing compactly supported distributions, we extend our analysis to sub-Gaussian tokens, allowing us to capture the combined effect of data anisotropy and attention geometry on the convergence behavior of attention mechanisms. We derive slower finite-sample convergence rates that match what is observed in both synthetic and real data. From an inference perspective, these rates quantify how quickly sampled-key attention recovers dense-model behavior, with the attention horizon identifying regimes where logits, predictions, and accuracy are slower to stabilize. Our upper bound applies to a large class of distributions and describes the observed slow convergence rate on real data. However, showing the tightness of our upper bound in the Gaussian setting, as suggested by our numerical experiments, remains widely open. More generally, the precise rate of convergence of attention crucially depends on the token distribution, and identifying classes of distributions and their corresponding tight rates is left for future work.

## Acknowledgements

This work was granted access to the HPC resources of IDRIS under the allocation 2025-[A0181016159] made by GENCI. The work of Gabriel Peyré was supported by the European Research Council (ERC project WOLF) and the French government under the management of Agence Nationale de la Recherche as part of the "France 2030" program, reference ANR-23-IACL-0008 (PRAIRIE-PSAI). The work of Léa Bohbot was supported by the Fondation CFM, through the Jean-Pierre Aguilar fellowship.

## Impact Statement

This paper presents work whose goal is to advance the field of Machine Learning. There are many potential societal consequences of our work, none which we feel must be specifically highlighted here.

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

# A. Additional figures

This section presents additional experimental results on both synthetic and real data in high dimension. All synthetic experiments were conducted using multi-head attention following Definition A.1.

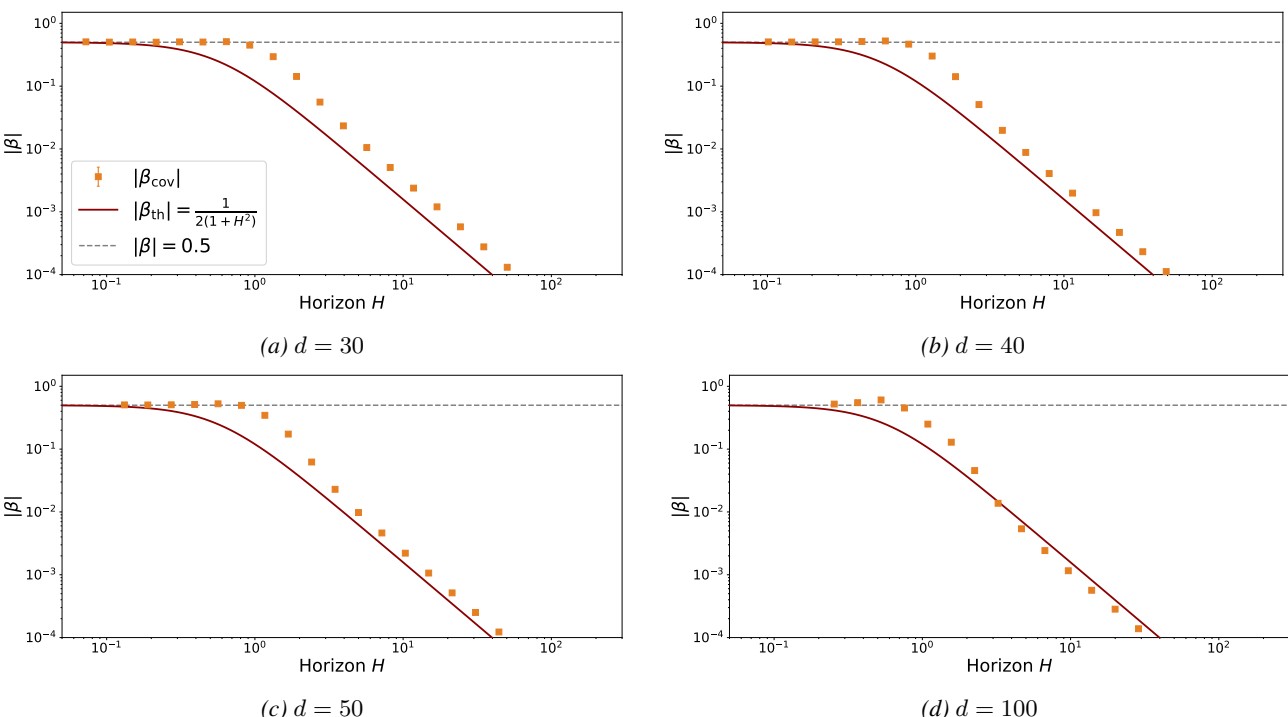

*(a) $d = 30$*  *(b) $d = 40$*

*(c) $d = 50$*  *(d) $d = 100$*

*Figure 7.* **Covariance convergence rate $|\beta|$ vs horizon $H$ for Gaussian token** embeddings $\mathcal{N}(0, \Sigma)$, with $\Sigma = \lambda_{max}\mathbf{I}$, across dimensions $d \in \{20, 30, 40, 50, 100\}$. We vary the horizon by rescaling a fixed matrix $A$, to capture both low and high-horizon regimes. Diagonal covariance matrices have identical spectral properties of $\Sigma$ across dimensions. The results demonstrate strong agreement with theoretical rate of Proposition D.8.

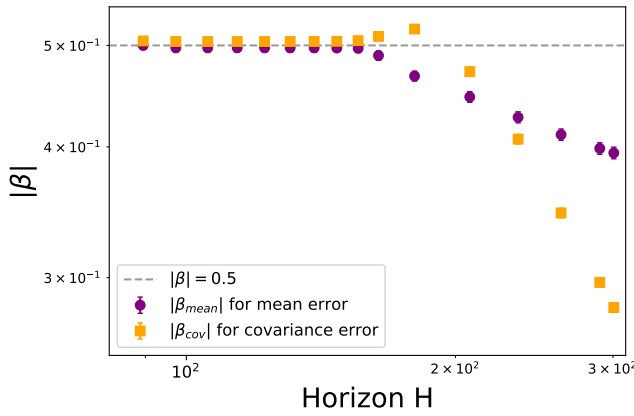

*Figure 8.* **Empirical mean and covariance convergence rate exponents** $|\beta_{mean}|$ **and** $|\beta_{cov}|$ as a function of attention horizon $H$ for uniform token embeddings on a unit sphere in dimension 50. Gray dotted line indicates the classical rate exponent $|\beta| = 0.5$ corresponding to the parametric rate in $1/\sqrt{n}$.

**Definition A.1** (Multi-head self-attention). Let $H_{\text{heads}}$ divide $d$ and set $k = d/H_{\text{heads}}$. For each head $h \in \{1, \dots, H_{\text{heads}}\}$, let $Q^{(h)}, K^{(h)}, V^{(h)} \in \mathbb{R}^{k \times d}$ and an output projection $W^{(h)} \in \mathbb{R}^{d \times k}$. Denoting by $f_n^{(h)}$ the single-head map associated with $\left(Q^{(h)}, K^{(h)}, V^{(h)}\right)$, the multi-head operator is

$$f_n^{\text{MH}}(X) = \sum_{h=1}^{H_{\text{heads}}} W^{(h)} f_n^{(h)}(X) \in \left(\mathbb{R}^d\right)^n.$$

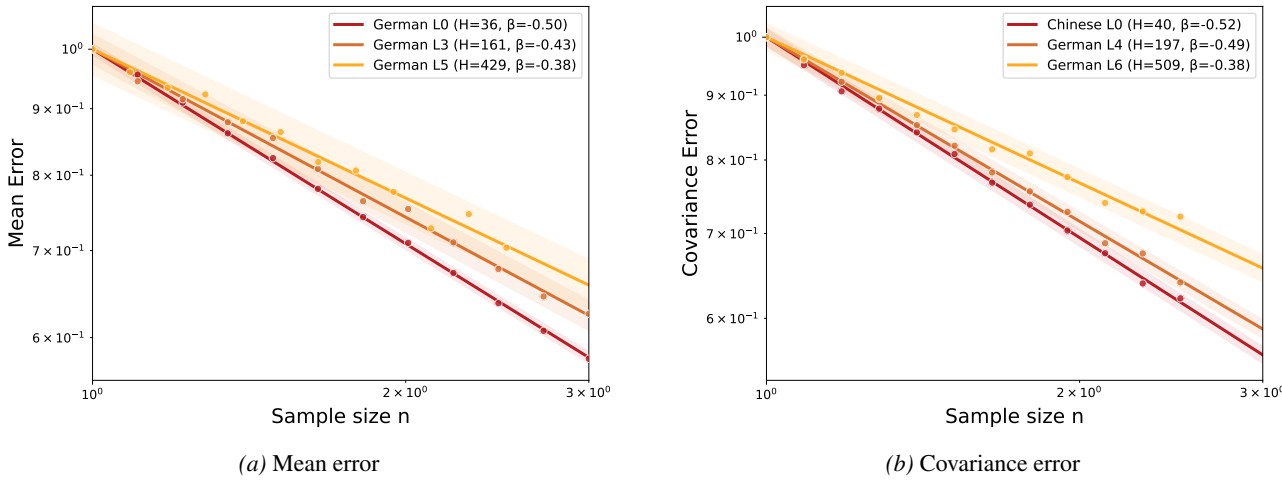

*Figure 9.* **Additional convergence-rate experiments on BigBird and BERT.** Top row: covariance convergence for BigBird-RoBERTa across layers and text distributions. The left panel reports the fitted convergence rate $|\beta|$ versus horizon $H$; the right panel shows the detailed covariance-error curves for the starred configurations. Bottom row: convergence rate $|\beta|$ versus horizon $H$ for BERT-base-uncased, for mean error and covariance error across Wikipedia text distributions. Markers indicate (layer, text configuration) pairs, colors indicate layers, and larger $|\beta|$ means faster convergence.

*Figure 10.* **Convergence curves for starred layers.** Mean error (left) and covariance error (right) versus subsample size $n$ for the three starred configurations from Figure 9, using BERT-base-uncased (dense attention). Markers show the averaged $\ell_2$ errors over $k = 500$–$1000$ subsamples, with $\pm 3$ standard-error bands. Log–log linear fits yield power-law exponents $|\beta|$ (legend), with error scaling as $n^{-|\beta|}$. From dark red to light orange as the horizon $H$ increases, confirming the slowdown predicted by Theorem 5.3.

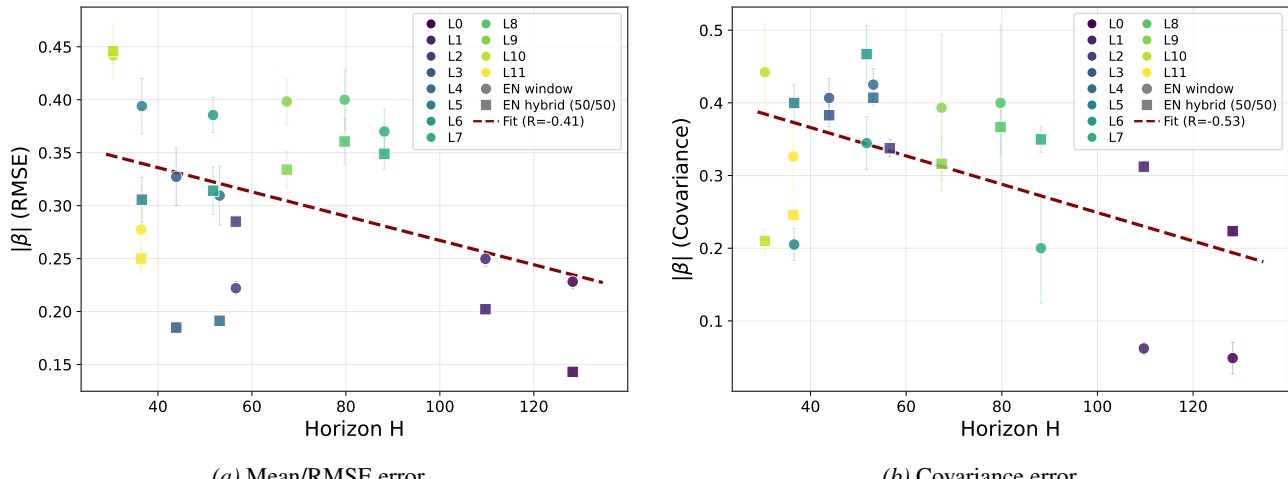

*(a)* Mean/RMSE error           *(b)* Covariance error

*Figure 11.* **Convergence rate $|\beta|$ versus horizon $H$ under non-i.i.d. windowed sampling.** BigBird–RoBERTa layers 0–11 are evaluated on English Wikipedia with two structured schemes: *window* ($n - r - 1$ local keys plus $r = 50$ random keys) and *hybrid* (half local, half random). In both mean/RMSE and covariance errors, larger horizons still correspond to slower convergence, confirming that the slowdown persists beyond i.i.d. sampling.

**Implementation details.** All experiments use $N_{\max} \approx 5 \times 10^5$ reference tokens of each dataset as the reference limit distribution, and pass them through BigBird–RoBERTa–base to compute the empirical mean and covariance of the attention outputs at each layer, which serve as approximations of the infinite-context limits. For subsample sizes $n \in [1000, 3000]$, we randomly sample $n$ tokens from the full sequence and estimate the corresponding mean and covariance errors $\left\| \mathbb{E}_n[f_n(\hat{X})] - \mathbb{E}[f(X)] \right\|_2$ and $\left\| \operatorname{Cov}_n(f_n(\hat{X})) - \operatorname{Cov}(f(X)) \right\|_2$ of Corollaries D.8 and D.4 . Each Monte Carlo estimate is repeated 500 times to reduce variance.

**Discussion on the i.i.d. assumption** Having tokens drawn from a distribution with Markovian dependencies does not contradict our theory: as long as tokens are sampled i.i.d. from that distribution — which is the case in our real-data experiments — the theorem applies directly. Nonetheless, in practice, subsampling is often structured, e.g. via windowing, and in that case the i.i.d. assumption no longer holds. Obtaining theoretical guarantees in this setting is significantly harder, which motivates our simplifying assumption. We addressed this concern empirically by conducting an additional experiment using windowing-based subsampling on BigBird (see Section 6 and Figure 11), and observe the same slowdown in convergence, suggesting that our theoretical predictions remain informative beyond the i.i.d. setting.

## B. Tools from empirical process theory

Before diving into the proof of Theorem 4.2, we recall a few results and definitions from empirical process theory that will be useful to the proof.

First, recall the definition of the Rademacher complexity.

**Definition B.1** (Rademacher complexity in $\mathbb{R}^d$). Let $\mathcal{F} \subset \{f : \mathcal{Y} \to \mathbb{R}^d\}$ be a class of (measurable) functions and $S = (Y_1, \ldots, Y_n) \sim \nu^n$ an i.i.d. sample. Let $\varepsilon_1, \ldots, \varepsilon_n$ be i.i.d. Rademacher variables, independent of $S$, with $\mathbb{P}(\varepsilon_i = 1) = \mathbb{P}(\varepsilon_i = -1) = \frac{1}{2}$. The *empirical* Rademacher complexity of $\mathcal{F}$ with respect to $S$ is:

$$\widehat{\mathfrak{R}}_S(\mathcal{F}) \; := \; \mathbb{E}_\varepsilon \left[ \sup_{f \in \mathcal{F}} \left\| \frac{1}{n} \sum_{i=1}^n \varepsilon_i \, f(Y_i) \right\|_2 \right].$$

The (expected) Rademacher complexity at sample size $n$ is

$$\mathfrak{R}_n(\mathcal{F}) \; := \; \mathbb{E}_{S \sim \nu^n} \left[ \widehat{\mathfrak{R}}_S(\mathcal{F}) \right].$$

If $\mathcal{F} = \{f_\theta : \theta \in \Theta\}$ is parameterized by $\Theta$, replace $\sup_{f \in \mathcal{F}}$ with $\sup_{\theta \in \Theta}$. For $d = 1$, this reduces to the usual scalar definition.

Intuitively, the Rademacher complexity quantifies the ability of the function class $\mathcal{F}$ to fit random noise on the specific sample $S \sim \nu$. Notice that the expression resembles the empirical expectation we aim to bound. The symmetrization step consists in artificially introducing Rademacher samples $\varepsilon_i$ to relate our supremum over $\mathcal{F}$ to a measure of the richness of the class, that relies on measuring the alignment between function outputs $f(Y_i)$ and random noise.

Next, we recall some useful definitions about sub-Gaussian processes and the Dudley's entropy integral, central in the proof Theorem 4.2.

**Definition B.2** (Sub-Gaussian $\psi_2$ norm). For a real-valued random variable $Z$, its $\psi_2$ (sub-Gaussian) norm is

$$\|Z\|_{\psi_2} := \inf\Big\{\sigma > 0: \ \mathbb{E}\Big[\exp\Big(\frac{Z^2}{\sigma^2}\Big)\Big] \leqslant 2\Big\}.$$

For a random vector $X \in \mathbb{R}^d$,

$$\|X\|_{\psi_2} := \sup_{v \in \mathbb{S}^{d-1}} \|\langle X, v\rangle\|_{\psi_2},$$

where $\mathbb{S}^{d-1} = \{v \in \mathbb{R}^d : \|v\|_2 = 1\}$. We say $X$ is sub-Gaussian if $\|X\|_{\psi_2} < \infty$.

**Definition B.3** (Sub-gaussian increments). Consider a random process $(X_t)_{t \in T}$ on a metric space $(T, d)$. We say that the process has *sub-Gaussian increments* if there exists $K \geqslant 0$ such that

$$\|X_t - X_s\|_{\psi_2} \leqslant K d(t, s) \quad \text{for all } t, s \in T, \tag{3}$$

where $\|\cdot\|_{\psi_2}$ denotes the sub-Gaussian norm.

**Definition B.4** ($\varepsilon$-net and covering numbers). Let $(T, d)$ be a metric space. Consider a set $T' \subset T$ and a number $\varepsilon > 0$. A subset $\mathcal{N} \subset T'$ is an $\varepsilon$-net of $T$ if balls of radius $\varepsilon$ centered at points in $\mathcal{N}$ cover $T'$.
The smallest cardinality of an $\varepsilon$-net of $T'$ is called the covering number of $K$ and is denoted $\mathcal{N}(\varepsilon, T', d)$. Equivalently, $\mathcal{N}(\varepsilon, T', d)$ is the smallest number of closed balls with centers in $T'$ and radius $\varepsilon$ whose union covers $T'$.

**Theorem B.5** (Dudley's entropy integral). *((Vershynin, 2018), see Theorem 8.1.3, p. 221, 2nd ed.) Let $(X_t)_{t \in T}$ be a centered random process on a metric space $(T, d)$ with sub-Gaussian increments as in (3), with $K \geqslant 0$ its sub-Gaussian parameter. Then, there exists a constant $C \geqslant 0$ such that:*

$$\mathbb{E}[\sup_{t \in T} X_t] \leqslant CK \int_0^{diam(T)} \sqrt{\ln \mathcal{N}(\varepsilon, T, d)}\, d\varepsilon, \tag{4}$$

*where $\mathcal{N}(\varepsilon, T, d)$ denotes the covering number of $T$ at scale $\varepsilon$ introduced in Definition B.4.*

**Definition B.6** (Data-dependent distance). Let $Y_1, \ldots, Y_n$ be a sample of random vectors in $\mathbb{R}^d$ and let $\mathbb{P}_n := \frac{1}{n}\sum_{i=1}^n \delta_{Y_i}$ denote the empirical measure. For any measurable $g$, define the empirical $L^2(\mathbb{P}_n)$ norm under the samples $Y_1, \ldots, Y_n$ by

$$\|g\|_{L^2(\mathbb{P}_n)} := \Big(\frac{1}{n}\sum_{i=1}^n \|g(Y_i)\|_2^2\Big)^{1/2}.$$

For the indexed class $\mathcal{F} = \{f_x : x \in B_R\}$, the data-dependent distance between $f_x$ and $f_{x'}$ is

$$\|f_x - f_{x'}\|_{L^2(\mathbb{P}_n)} = \Big(\frac{1}{n}\sum_{i=1}^n \|f_x(Y_i) - f_{x'}(Y_i)\|_2^2\Big)^{1/2}.$$

**Lemma B.7** (Empirical Lipschitz control). *Assume there exists a measurable envelope $L : \mathbb{R}^d \to [0, \infty)$ such that for all $x, x' \in B_R$ and all $y \in \mathbb{R}^d$,*

$$\|f_x(y) - f_{x'}(y)\|_2 \leqslant L(y)\|x - x'\|_2.$$

*Then, for any samples $Y_1, \ldots, Y_n$,*

$$\|f_x - f_{x'}\|_{L^2(\mathbb{P}_n)} \leqslant \|L\|_{L^2(\mathbb{P}_n)}\|x - x'\|_2.$$

*Proof.* By the pointwise Lipschitz condition and the definition of $\| \cdot \|_{L^2(\mathbb{P}_n)}$,

$$\|f_x - f_{x'}\|^2_{L^2(\mathbb{P}_n)} = \frac{1}{n}\sum_{i=1}^n \|f_x(Y_i) - f_{x'}(Y_i)\|^2_2 \leqslant \frac{1}{n}\sum_{i=1}^n L(Y_i)^2 \|x - x'\|^2_2 = \|L\|^2_{L^2(\mathbb{P}_n)}\|x - x'\|^2_2.$$

The claim follows by taking square roots. $\qquad\square$

**Lemma B.8** (Bound on empirical Lipschitz envelope). *Let $Y_1, \dots, Y_n$ be i.i.d. with law $\nu$. Assume there exists a function $L$ as defined in Lemma B.7 such that $\mathbb{E}[L(Y_1)^2] < \infty$. Then*

$$\mathbb{E}\big[\|L\|_{L^2(\mathbb{P}_n)}\big] \leqslant \big(\mathbb{E}[\|L(Y_1)\|^2_2]\big)^{1/2} =: \|L\|_{L^2(\nu)}.$$

*Proof.* By Cauchy–Schwarz applied to $Z = \|L\|_{L^2(\mathbb{P}_n)}$:

$$\mathbb{E}\big[\|L\|_{L^2(\mathbb{P}_n)}\big] \leqslant \Big(\mathbb{E}\big[\|L\|^2_{L^2(\mathbb{P}_n)}\big]\Big)^{1/2}.$$

Using the fact that $Y_1, \dots, Y_n$ are i.i.d,

$$\mathbb{E}\big[\|L\|^2_{L^2(\mathbb{P}_n)}\big] = \mathbb{E}\left[\frac{1}{n}\sum_{i=1}^n L(Y_i)^2\right] = \frac{1}{n}\sum_{i=1}^n \mathbb{E}[L(Y_i)^2] = \mathbb{E}[L(Y_1)^2].$$

$\qquad\square$

# C. Proofs

## C.1. Proofs of technical Lemmas for Theorem 4.2

**Lemma C.1** (Bound on the covering number of a product space). *Let $\mathcal{F} := \{f_x, x \in \Theta\}$ be a parametric class of Lipschitz functions with respect to $x$. Let $\| \cdot \|^2_{L^2(\mathbb{P}_n)}$ defined in Definition B.6. Consider the product space $\mathcal{G} = \mathcal{F} \times B_1$ and the associated distance defined in Lemma C.3. The covering number of the product space is bounded by:*

$$\mathcal{N}(\varepsilon, \mathcal{F} \times B_1, d_{\mathcal{G}}((z_1, x_1), (z_2, x_2))) \leqslant \mathcal{N}\left(\frac{\varepsilon}{2}, \mathcal{F}, \| \cdot \|_{L^2(\mathbb{P}_n)}\right) \times \mathcal{N}\left(\frac{\varepsilon}{2\|F(\cdot)\|_{L^2(\mathbb{P}_n)}}, B_1, \| \cdot \|_2\right).$$

*Proof.* Let $\mathcal{N}(\varepsilon, \mathcal{F} \times B_1, d_{\mathcal{G}}((z_1, x_1), (z_2, x_2)))$ be the covering number of $\mathcal{G}$ for the distance $d_{\mathcal{G}}$.

Consider an $\frac{\varepsilon}{2}$-cover of $\mathcal{F}$ and a $\frac{\varepsilon}{2\|F(\cdot)\|_{L^2(\mathbb{P}_n)}}$-cover of $B_1$. We can write,

$$\begin{aligned}
\|z_1^\top f_{x_1}(\cdot) - z_2^\top f_{x_2}(\cdot)\|_{L^2(\mathbb{P}_n)} &\leqslant \|z_1^\top f_{x_1}(\cdot) - z_1^\top f_{x_2}(\cdot) + z_1^\top f_{x_2}(\cdot) - z_2^\top f_{x_2}(\cdot)\|_{L^2(\mathbb{P}_n)} \\
&\leqslant \|f_{x_1}(\cdot) - f_{x_2}(\cdot)\|_{L^2(\mathbb{P}_n)} + \|z_1 - z_2\|_2 \|F(\cdot)\|_{L^2(\mathbb{P}_n)} \\
&\leqslant \frac{\varepsilon}{2} + \frac{\varepsilon}{2\|F(\cdot)\|_{L^2(\mathbb{P}_n)}}\|F(\cdot)\|_{L^2(\mathbb{P}_n)} = \varepsilon.
\end{aligned}$$

The conclusion of the lemma follows immediately. $\qquad\square$

**Lemma C.2** (Bound on the covering number). *Let $\mathcal{F}$ be a parametric class of function $\{f_x, x \in \Theta\}$ such that any $f_x \in \mathcal{F}$ is Lipschitz with respect to its index parameter $x$. Let $\| \cdot \|^2_{L^2(\mathbb{P}_n)}$ be defined by Definition B.6, then:*

$$\mathcal{N}(\delta, \mathcal{F}, \| \cdot \|_{L^2(\mathbb{P}_n)}) \leqslant \mathcal{N}\left(\frac{\delta}{\|L\|_{L^2(\mathbb{P}_n)}}, \Theta, \| \cdot \|_2\right).$$

*Proof.* Let $x_1, \dots, x_M$ form an $\frac{\delta}{\|L\|_{L^2(\mathbb{P}_n)}}$ cover for $\Theta$. Then $\{f(x_i, \cdot) : i = 1, \dots, M\}$ form an $\delta$ cover for $\mathcal{F}$. Hence the result. $\qquad\square$

The following Lemmas prove that the empirical processes $Z_h$ and $Z_{(z,x)}$ defined in the proof of Theorem 4.2 in Appendix C.2 are sub-Gaussian processes w.r.t. well-defined metrics.

**Lemma C.3** (Sub-gaussian increments in dimension $d$). *Conditionally on the $Y_i$, $Z_{(z,x)} = \frac{1}{n} \sum_{i=1}^n \varepsilon_i \langle z, f_x(Y_i) \rangle$ is a sub-Gaussian process with respect to the distance $d_\mathcal{G}((z_1, x_1), (z_2, x_2))$ of the metric space $\mathcal{G} = B_1 \times \mathcal{F}$ defined by:*

$$d_\mathcal{G}((z_1, x_1), (z_2, x_2)) := \|f_{x_1}(\cdot) - f_{x_2}(\cdot)\|_{L^2(\mathbb{P}_n)} + \|z_1 - z_2\|_2 \|F(\cdot)\|_{L^2(\mathbb{P}_n)}.$$

*Proof.* By Hoeffding's Lemma:

$$\mathbb{E}[\exp(\lambda(Z_{(z_1,x_1)} - Z_{(z_2,x_2)}))|Y_1, \ldots, Y_n]$$
$$= \prod_{i=1}^n \mathbb{E}\left[\exp\left(\frac{\lambda}{\sqrt{n}} \varepsilon_i (\langle z_1, f_{x_1}(Y_i)\rangle - \langle z_2, f_{x_2}(Y_i)\rangle)\right) \Big| Y_1, \ldots, Y_n\right]$$
$$\leqslant \exp\left(\frac{\lambda^2}{2} \|z_1^\top f_{x_1}(\cdot) - z_2^\top f_{x_2}(\cdot)\|_{L^2(\mathbb{P}_n)}^2\right).$$

This norm admits the following expression:

$$\|z_1^\top f_{x_1}(\cdot) - z_2^\top f_{x_2}(\cdot)\|_{L^2(\mathbb{P}_n)} = \|z_1^\top f_{x_1}(\cdot) - z_1^\top f_{x_2}(\cdot) + z_1^\top f_{x_2}(\cdot) - z_2^\top f_{x_2}(\cdot)\|_{L^2(\mathbb{P}_n)}$$
$$\leqslant \|z_1\|_2 \|f_{x_1}(\cdot) - f_{x_2}(\cdot)\|_{L^2(\mathbb{P}_n)} + \|z_1 - z_2\|_2 \|f_{x_2}(\cdot)\|_{L^2(\mathbb{P}_n)}$$
$$\leqslant \|f_{x_1}(\cdot) - f_{x_2}(\cdot)\|_{L^2(\mathbb{P}_n)} + \|z_1 - z_2\|_2 \|F(\cdot)\|_{L^2(\mathbb{P}_n)}.$$

This leads to the announced distance on the product space $\mathcal{G}$. $\qquad\square$

**Lemma C.4** (Sub-gaussian increments in dimension 1). *Conditionally on the $Y_i$, $Z_h = \frac{1}{\sqrt{n}} \sum_{i=1}^n \varepsilon_i h_x(Y_i)$ is a sub-Gaussian process with respect to the norm $\|\cdot\|_{\mathcal{L}_2(\mathbb{P}_n)}$.*

*Proof.* The proof follows from Hoeffding's lemma, as for Lemma C.3. $\qquad\square$

### C.2. Proof of Theorem 4.2

The strategy of the proof of Theorem 4.2 is to analyze the numerator and denominator of the attention ratio separately, and then combine the results. We will prove this Theorem in three parts. We first establish a concentration in expectation and subsequently derive a high-probability bound via Markov's inequality. These steps rely on classical tools from the empirical processes literature.

**Part 1: Uniform concentration of the numerator of $f_n(x)$**    Recall that the continuous attention map is given by

$$f(x) = \frac{\mathbb{E}[Y\, e^{\langle Ax, Y\rangle}]}{\mathbb{E}[e^{\langle Ax, Y\rangle}]}.$$

Consider the numerator. It is the expectation of $y \mapsto y\, e^{\langle Ax, y\rangle}$, for an $x \in \mathbb{R}^d$, over the sub-Gaussian random vector $Y$.

Define the function class $\mathcal{F} := \{f_x,\ x \in B_R\}$, where, for all $x \in \mathbb{R}^d, y \in \mathbb{R}^d$,

$$f_x : y \mapsto y\, e^{\langle Ax,\, y\rangle} \in L^2(\nu; \mathbb{R}^d). \tag{5}$$

Our goal is to prove a uniform concentration result over the function class $\mathcal{F}$, indexed by $x$ in the ball $B_R$.

Let

$$\|\mathbb{P}_n - \mathbb{P}\|_\mathcal{F} := \sup_{f \in \mathcal{F}} \Big\| \frac{1}{n} \sum_{i=1}^n f_x(Y_i) - \mathbb{E}_{Y \sim \nu}[f_x(Y)] \Big\|_2,$$

and define the envelope function of the class $\mathcal{F}$,

$$F : y \mapsto \sup_{x \in B_R} \|f_x(y)\|_2 \leqslant \|y\|_2\, e^{R\|A^\top y\|_2}. \tag{6}$$

**Proposition C.5** (Bound on numerator). *With probability at least $1 - \delta$,*

$$\|\mathbb{P}_n - \mathbb{P}\|_{\mathcal{F}} \leqslant \frac{C}{\delta} \, d \left( R\|A\|_2 \sqrt{\|\Sigma\|_2 d} + 5^{d/2} \right) \cdot \frac{\sqrt{\|\Sigma\|_2}}{\sqrt{n}} \cdot e^{8R^2 \, \|\Sigma^{1/2} A\|_2^2}.$$

*Proof.* We will first bound $\mathbb{E}[\|\mathbb{P}_n - \mathbb{P}\|_{\mathcal{F}}]$, and then apply Markov's inequality to obtain a high-probability concentration bound.

The proof follows two main steps. First, we use a symmetrization lemma (Lemma C.6 below) to relate $\mathbb{E}[\|\mathbb{P}_n - \mathbb{P}\|_{\mathcal{F}}]$ to the Rademacher complexity of the function class $\mathcal{F}$. Next, we use Dudley's entropy integral together with the Lipschitz property of $f_x$ to bound this Rademacher complexity via covering numbers.

**1. Symmetrization of $\mathbb{E}[\|\mathbb{P}_n - \mathbb{P}\|_{\mathcal{F}}]$**
We first derive a symmetrization result that will allow us to link $\mathbb{E}[\|\mathbb{P}_n - \mathbb{P}\|_{\mathcal{F}}]$ to the Rademacher complexity of the class $\mathcal{F}$ defined in Section B.

**Lemma C.6** (Symmetrization of $\mathbb{E}[\|\mathbb{P}_n - \mathbb{P}\|_{\mathcal{F}}]$)**.**

$$\mathbb{E}[\|\mathbb{P}_n - \mathbb{P}\|_{\mathcal{F}}] \leqslant 2\,\mathfrak{R}_n(\mathcal{F}).$$

*Proof.* Let $(X_1, ..., X_n)$ i.i.d. copies of $Y$ independent of $Y$.

$$\|\mathbb{P}_n - \mathbb{P}\|_{\mathcal{F}} = \sup_{f \in \mathcal{F}} \left\| \frac{1}{n} \sum_{i=1}^{n} f(Y_i) - \mathbb{E}_Y[f(Y)] \right\|_2$$

$$= \sup_{f \in \mathcal{F}} \left\| \frac{1}{n} \sum_{i=1}^{n} (f(Y_i) - \mathbb{E}_{X_i}[f(X_i)]) \right\|_2$$

$$= \sup_{f \in \mathcal{F}} \left\| \mathbb{E}_X \left[ \frac{1}{n} \sum_{i=1}^{n} f(Y_i) - f(X_i) \right] \right\|_2.$$

Using Jensen inequality, we get:

$$\mathbb{E}[\|\mathbb{P}_n - \mathbb{P}\|_{\mathcal{F}}] = \mathbb{E}_Y \left[ \sup_{f \in \mathcal{F}} \left\| \mathbb{E}_X \left[ \frac{1}{n} \sum_{i=1}^{n} (f(Y_i) - f(X_i)) \right] \right\|_2 \right]$$

$$\leqslant \mathbb{E}_{X,Y} \left[ \sup_{f \in \mathcal{F}} \left\| \frac{1}{n} \sum_{i=1}^{n} (f(Y_i) - f(X_i)) \right\|_2 \right].$$

Let $\varepsilon_1, \ldots, \varepsilon_n$ be i.i.d. Rademacher signs, independent of $S = (Y_1, \ldots, Y_n)$. The random vector of components $\varepsilon_i(f(Y_i) - f(X_i))$ has the same joint distribution as the vector with components $f(Y_i) - f(X_i)$. Hence,

$$\mathbb{E}_{X,Y} \left[ \sup_{f \in \mathcal{F}} \left\| \frac{1}{n} \sum_{i=1}^{n} (f(Y_i) - f(X_i)) \right\|_2 \right] = \mathbb{E}_{X,Y,\varepsilon} \left[ \sup_{f \in \mathcal{F}} \left\| \frac{1}{n} \sum_{i=1}^{n} \varepsilon_i(f(Y_i) - f(X_i)) \right\|_2 \right]$$

$$\leqslant 2\mathbb{E}_{Y,\varepsilon} \left[ \sup_{f \in \mathcal{F}} \left\| \frac{1}{n} \sum_{i=1}^{n} \varepsilon_i f(Y_i) \right\|_2 \right] := 2\mathfrak{R}_n(\mathcal{F}).$$

This yields $\mathbb{E}[\|\mathbb{P}_n - \mathbb{P}\|_{\mathcal{F}}] \leqslant 2\mathfrak{R}_n(\mathcal{F})$. $\qquad\square$

**2. Bounding the Rademacher complexity $\mathfrak{R}_n(\mathcal{F})$**
Now that we have bounded $\mathbb{E}[\|\mathbb{P}_n - \mathbb{P}\|_{\mathcal{F}}]$ by the Rademacher complexity, the next step of the proof is to use Dudley's entropy integral to bound this Rademacher complexity $\mathfrak{R}_n(\mathcal{F})$.

In order to do that, we will use some useful definitions introduced in Section B, as well as the Dudley's entropy integral Theorem B.5.

Recall that the Rademacher complexity we want to bound is:

$$\mathfrak{R}_n(\mathcal{F}) \; := \; \mathbb{E}_{S,\varepsilon}[\sup_{x \in B_R} \underbrace{\left\| \frac{1}{n} \sum_{i=1}^{n} \varepsilon_i \, f_x(Y_i) \right\|_2}_{:=\|Z_x\|_2}] =: \mathbb{E}_S[\underbrace{\mathbb{E}_\varepsilon[\sup_{x \in B_R} \|Z_x\|_2]}_{(\star)}].$$

The strategy of the rest of the proof is to apply Dudley's Theorem B.5 to $(\star)$. In the one-dimensional case, Dudley's theorem can be applied directly to the scalar process $\frac{1}{\sqrt{n}} \sum_{i=1}^{n} \varepsilon_i \, f_x(Y_i)$, which is a sub-Gaussian process conditioned on the $Y_i$ with respect to the associated empirical norm $\|f\|_{L^2(\mathbb{P}_n)} := \left( \frac{1}{n} \sum_{i=1}^{n} f(Y_i)^2 \right)^{1/2}$ (see Lemma C.4). In the present setting, however, the process takes values in $\mathbb{R}^d$. To use the same concentration inequality, we therefore project it onto scalar directions—that is, we use the dual norm considering:

$$Z_{(z,x)} := \frac{1}{n} \sum_{i=1}^{n} \varepsilon_i \langle z, f_x(Y_i) \rangle := z^\top Z_x. \tag{7}$$

We must then verify that this scalar process $Z_{(z,x)}$ remains sub-Gaussian, though, now, with respect to an appropriate metric $d_{\mathcal{G}}$ defined on the product space $(\mathcal{G}, d_{\mathcal{G}})$, where $\mathcal{G} := \mathcal{F} \times B_1$ (Lemma C.3). Once this sub-Gaussian property is established, Dudley's theorem B.5 can be applied to $Z_{(z,x)}$ and the rest of the analysis will require bounds on:

(i) the moments of the envelope function $F$ (Lemma C.8 ),

(ii) the Lipschitz constant of functions in $\mathcal{F}$, (Lemmas C.7 and C.10), and

(iii) the covering number of $\mathcal{G}$ (Lemmas C.1 and C.2).

In order to do this we will need the definition B.6 of the empirical distance and Lemmas B.7 and B.8 introduced in Section B, which we will be referring to.

**Lemma C.7** (Bound on Lipschitz constant for numerator). *Let $f \in \mathcal{F} := \{f_x, \, x \in B_R\}$ as defined in (5). With an abuse of notations, denote $f : (x, y) \mapsto y \, e^{\langle Ax, y \rangle}$. Then, $f(\cdot, y)$ is Lipschitz with respect to $x$. Let $y \mapsto L(y)$ be its Lipschitz constant. Consider $Y \sim \nu$, a sub-gaussian random vector in $\mathbb{R}^d$ of matrix parameter $\Sigma$, and scalar parameter $\sigma := \sqrt{\|\Sigma\|_2}$. Then, $L$ verifies*

$$\|L\|_{L^2(\nu)} \leqslant C\|A\|_2 \, \sigma^2 \, d \, \exp\!\left(2 \, R^2 \|\Sigma^{1/2}A\|_2^2\right),$$

*for some constant $C > 0$.*

*Proof.* Consider $f_t(y) := y \, e^{\langle t, y \rangle}$ for $t \in \mathbb{R}^d$ - we will then apply the result to $t = Ax$ using $\nabla_x f_x(y) = A^\top (\nabla_t f_t)(Ax, y)$. The Jacobian of $f(\cdot, y)$ at $t$ is

$$\nabla_t f_t(y) \; = \; y \, y^\top \, e^{\langle t, y \rangle} \in \mathbb{R}^{d \times d}.$$

Let

$$\|\nabla_t f_t\|_{L^2(\nu)} := \left( \mathbb{E}_{Y \sim \nu} \|\nabla_t f_t(Y)\|_2^2 \right)^{1/2} = \left( \mathbb{E}_{Y \sim \nu} \left[ \|Y\|_2^4 \, e^{2\langle t, Y \rangle} \right] \right)^{1/2},$$

where we used the fact that $YY^\top$ is a rank one matrix, hence $\|YY^\top\| = \|Y\|_2^2$.

By Cauchy–Schwarz and the sub-Gaussian property of $Y$,

$$\|\nabla_t f_t\|_{L^2(\nu)} \leqslant \left( \mathbb{E}_{Y \sim \nu} \left[ \|Y\|_2^8 \right] \right)^{1/4} \left( \mathbb{E}_{Y \sim \nu} e^{4\langle t, Y \rangle} \right)^{1/4}$$

$$\leqslant \left( \mathbb{E}_{Y \sim \nu} \left[ \|Y\|_2^8 \right] \right)^{1/4} \exp\!\left(2 \, t^\top \Sigma t\right).$$

Use $t^\top \Sigma t = x^\top A^\top \Sigma A \, x \leqslant \|\Sigma^{1/2}A\|_2^2 \|x\|_2^2$, and take the supremum over $\|x\|_2 \leqslant R$:

$$\sup_{\|x\|_2 \leqslant R} \|\nabla_x f_x\|_{L^2(\nu)} \leqslant \|A\|_2 \left( \mathbb{E}_{Y \sim \nu} \|Y\|_2^8 \right)^{1/4} \exp\!\left(2 \, R^2 \|\Sigma^{1/2}A\|_2^2\right).$$

To express the moment factor,

$$\left(\mathbb{E}_{Y\sim\nu}\|Y\|_2^8\right)^{1/4} = \left(\left(\mathbb{E}_{Y\sim\nu}\|Y\|_2^8\right)^{1/8}\right)^2 \underbrace{\leqslant}_{\star} (c\,\sigma\,\sqrt{d})^2,$$

where $\star$ is an extension to dimension $d$ of a classical bound on sub-Gaussian moments, with $c$ a universal constant (see Lemma 5.5 in (Vershynin, 2012a)).

Hence

$$\|L\|_{L^2(\nu)} := \sup_{\|x\|_2\leqslant R} \|\nabla_x f_x\|_{L^2(\nu)} \leqslant C\|A\|_2\,\sigma^2\,d\,\exp\!\left(2\,R^2\|\Sigma^{1/2}A\|_2^2\right)$$

which concludes the proof. $\qquad\square$

Then, we will need a bound on the envelope function of the class $\mathcal{F}$.

**Lemma C.8** (Bound on envelope function). *Let $F$ be the envelope function defined in equation (6), and $Y \sim \nu$, a sub-gaussian random vector in $\mathbb{R}^d$ of parameter $\sigma = \sqrt{\|\Sigma\|_2}$. Then,*

$$\mathbb{E}[F(Y)^2]^{1/2} \leqslant C5^{d/2}\,\sqrt{d}\,\sigma\,e^{8R^2\,\|\Sigma^{1/2}A\|_2^2}.$$

*Proof.* By Cauchy–Schwarz,

$$\mathbb{E}[F(Y)^2]^{1/2} = \mathbb{E}\big[\|Y\|_2^2\,e^{2R\|A^\top Y\|_2}\big]^{1/2} \leqslant \left(\mathbb{E}\|Y\|_2^4\right)^{1/4}\left(\mathbb{E}e^{4R\|A^\top Y\|_2}\right)^{1/4}.$$

The first term $\left(\mathbb{E}\|Y\|_2^4\right)^{1/4}$ is controlled by the same argument used for $\left(\mathbb{E}_{Y\sim\nu}\|Y\|_2^8\right)^{1/4}$ in Lemma C.7. Now, let $\mathbb{S}^{d-1}$ be the unit sphere and $U \subset \mathbb{S}^{d-1}$ an $\varepsilon$-net with $|U| \leqslant (1+2/\varepsilon)^d$. By a standard estimate (e.g. (Vershynin, 2018), Lemma 4.4.1, p. 116, 2nd ed.) we have,

$$\|g\|_2 = \sup_{\|u\|=1}\langle g,u\rangle \leqslant \frac{1}{1-\varepsilon}\max_{u\in U}\langle g,u\rangle,$$

hence for $g = A^\top Y$,

$$e^{4R\|A^\top Y\|_2} \leqslant \sum_{u\in U}\exp\!\left(\tfrac{4R}{1-\varepsilon}\langle Au,Y\rangle\right).$$

Taking expectations and using the sub-Gaussian moment generative function bound of Definition 4.1 with $t = \frac{4R}{1-\varepsilon}Au$,

$$\mathbb{E}\,e^{4R\|A^\top Y\|_2} \leqslant (1+2/\varepsilon)^d\,\exp\!\left(\tfrac{1}{2}\Big\|\tfrac{4R}{1-\varepsilon}\Sigma^{1/2}A\Big\|_2^2\right) = (1+2/\varepsilon)^d\,\exp\!\left(\frac{8R^2}{(1-\varepsilon)^2}\|\Sigma^{1/2}A\|_2^2\right).$$

Finally, for $\varepsilon = \frac{1}{2}$,

$$\left(\mathbb{E}[F(Y)^2]\right)^{1/2} \leqslant C5^{d/2}\,\sqrt{d}\,\sigma\,e^{8R^2\,\|\Sigma^{1/2}A\|_2^2}$$

which concludes the proof. $\qquad\square$

Proposition B.7 holds for all $f \in \mathcal{F}$. That is for all $x, x' \in B_R$,

$$\|f_x - f_{x'}\|_{L^2(\mathbb{P}_n)}^2 \leqslant \|L\|_{L^2(\mathbb{P}_n)}^2\|x - x'\|_2^2.$$

We formally prove in Lemma C.3 that the empirical process defined in equation (7) is a sub-Gaussian process, with respect to the distance associated to the product space $\mathcal{G} = (\mathcal{F} \times B_1)$ defined in Lemma C.3. We also need a bound on the covering number of $\mathcal{G}$. Corresponding Lemmas with their proofs can be found in Appendix C.1.

Now that we have gathered all the necessary tools to bound the Rademacher complexity $\mathfrak{R}_n(\mathcal{F})$ we can end the proof of Proposition C.5.

*Proof of Proposition C.5*

Let $R_{L,F} = 2R\|L\|_{L^2(\mathbb{P}_n)} + 2\|F\|_{L^2(\mathbb{P}_n)}$ and apply Theorem B.5 to the centered sub-Gaussian process $Z_{(z,x)}$ with respect to the metric space $(\mathcal{G}, d_{\mathcal{G}})$, in order to bound $\mathbb{E}_\varepsilon[\sup_{(z,x)\in\mathcal{G}} Z_{(z,x)}]$,

$$\mathbb{E}_\varepsilon\big[\sup_{(z,x)\in\mathcal{G}} Z_{(z,x)}\big]$$

$$\leqslant \frac{C}{\sqrt{n}} \int_0^{\mathrm{diam}(\mathcal{G})} \sqrt{\ln \mathcal{N}(\delta, \mathcal{G}, d_{\mathcal{G}})}\, d\delta$$

$$\leqslant \frac{C}{\sqrt{n}} \int_0^{R_{L,F}} \sqrt{\ln \mathcal{N}\Big(\frac{\delta}{2\|L\|_{L^2(\mathbb{P}_n)}}, B_R, \|\cdot\|_2\Big)} + \sqrt{\ln \mathcal{N}\Big(\frac{\delta}{2\|F(\cdot)\|_{L^2(\mathbb{P}_n)}}, B_1, \|\cdot\|_2\Big)}\, d\delta,$$

where we used Lemma C.1 and C.2.

Taking the expectation over $Y_i$, and using the symmetrization Lemma C.6,

$$\mathbb{E}[\|\mathbb{P}_n - \mathbb{P}\|_{\mathcal{F}}] \leqslant \mathfrak{R}_n(\mathcal{F}) := \mathbb{E}_Y \mathbb{E}_\varepsilon\big[\sup_{(z,x)\in\mathcal{G}} Z_{(z,x)}|Y_1,...,Y_n\big] \leqslant \frac{c}{\sqrt{n}}(\mathcal{I} + \mathcal{J})$$

where

$$\mathcal{I} = \mathbb{E}_Y\Big[\int_0^{R_{L,F}} \sqrt{\ln \mathcal{N}\Big(\frac{\delta}{2\|L\|_{L^2(\mathbb{P}_n)}}, B_R, \|\cdot\|_2\Big)}\Big]d\delta,$$

and

$$\mathcal{J} = \mathbb{E}_Y\Big[\int_0^{R_{L,F}} \sqrt{\ln \mathcal{N}\Big(\frac{\delta}{2\|F(\cdot)\|_{L^2(\mathbb{P}_n)}}, B_1, \|\cdot\|_2\Big)}\, d\delta\Big].$$

For the term $\mathcal{I}$, set the change of variable $u = \frac{\delta}{2R\|L\|_{L^2(\mathbb{P}_n)}}$ for $0 \leqslant u \leqslant 1$. For the second term $\mathcal{J}$, set $u = \frac{\delta}{2\|F\|_{L^2(\mathbb{P}_n)}}$

$$\mathcal{I} \leqslant \mathbb{E}\left[\int_0^{1+\frac{\|F\|_n}{R\|L\|_n}} \sqrt{\ln \mathcal{N}(Ru, B_R, \|\cdot\|_2)}\, 2R\|L\|_{L^2(\mathbb{P}_n)}\, du\right],$$

$$\mathcal{J} \leqslant \mathbb{E}\left[\int_0^{1+\frac{R\|L\|_n}{\|F\|_n}} \sqrt{\ln \mathcal{N}(u, B_1, \|\cdot\|_2)}\, 2\|F\|_{L^2(\mathbb{P}_n)}\, du\right],$$

where we can denote $\|F\|_n := \|F\|_{L^2(\mathbb{P}_n)}$ and $\|L\|_n := \|L\|_{L^2(\mathbb{P}_n)}$ in the integrand to alleviate the notation.

Recall that $\ln \mathcal{N}(\varepsilon, B_r, \|\cdot\|) \leqslant d\ln(1 + \frac{2r}{\varepsilon})$ (see Corollary 4.2.11. in (Vershynin, 2018)), hence

$$\mathcal{I} \leqslant 4\sqrt{2d}\, R\, \mathbb{E}[\|L\|_{L^2(\mathbb{P}_n)}] + 2\sqrt{d\ln 3}\, \mathbb{E}[\|F\|_{L^2(\mathbb{P}_n)}]. \tag{8}$$

The same steps for the second term give,

$$\mathcal{J} \leqslant 4\sqrt{2d}\, \mathbb{E}[\|F\|_{L^2(\mathbb{P}_n)}] + 2R\sqrt{d\ln 3}\, \mathbb{E}[\|L\|_{L^2(\mathbb{P}_n)}]. \tag{9}$$

Detailed derivations of (8) and (9) can be found in Appendix C.3. Finally,

$$\mathbb{E}[\|\mathbb{P}_n - \mathbb{P}\|_{\mathcal{F}}] \leqslant \frac{c\sqrt{d}}{\sqrt{n}}\Big(R\,\mathbb{E}[\|L\|_{L^2(\mathbb{P}_n)}] + \mathbb{E}[\|F\|_{L^2(\mathbb{P}_n)}]\Big). \tag{10}$$

Applying Lemma B.8, $\mathbb{E}[\|L\|_{L^2(\mathbb{P}_n)}] \leqslant \left(\mathbb{E}\left[\|L\|_{L^2(\mathbb{P}_n)}^2\right]\right)^{1/2} = (\mathbb{E}[L(Y_1)^2])^{1/2} = \|L\|_{L^2(\nu)}$, and using Lemma C.7, along with Lemma C.8 it comes,

$$\mathbb{E}[\|\mathbb{P}_n - \mathbb{P}\|_{\mathcal{F}}] \leqslant \frac{c\sqrt{d}}{\sqrt{n}}\Big(R\, C_1\, \|A\|_2\sigma^2\, d e^{2R^2\|\Sigma^{1/2}A\|_2^2} + 5^{d/2}C_2\sigma\sqrt{d}\, e^{8R^2\|\Sigma^{1/2}A\|_2^2}\Big)$$

$$\leqslant \frac{C\, d\, (R\|A\|_2\sigma\sqrt{d} + 5^{d/2})\sigma}{\sqrt{n}}\, e^{8R^2\|\Sigma^{1/2}A\|_2^2} =: \frac{C_0(\Sigma,\, A,\, \delta,\, R)}{\sqrt{n}}.$$

for some positive constant $C$.

To conclude, we apply the Markov inequality, which yields, with probability at least $1-\delta$, $\|\mathbb{P}_n - \mathbb{P}\|_{\mathcal{F}} \leqslant \frac{1}{\delta}\,\mathbb{E}[\|\mathbb{P}_n - \mathbb{P}\|_{\mathcal{F}}] \leqslant \frac{C_0(\Sigma, A, \delta, R)}{\delta \cdot \sqrt{n}}$. This ends the proof of Proposition C.5. $\qquad\square$

**Part 2: Uniform concentration of the denominator empirical process** An analogous result as Proposition C.5 holds for the denominator with an identical proof using the function class

$$\mathcal{F}' := \{h_x : y \mapsto e^{\langle Ax, y \rangle}, x \in B_R\}.$$

**Proposition C.9** (Bound on denominator). *For all $R > 0$ and $\delta \in (0, 1)$, with probability at least $1 - \delta$,*

$$\|\mathbb{P}_n - \mathbb{P}\|_{\mathcal{F}'} \leqslant \frac{C'}{\delta} \cdot \frac{\|A\|_2 \, \sigma \sqrt{d}}{\sqrt{n}} e^{2 \, R^2 \|\Sigma^{1/2} A\|_2^2}.$$

*Proof.* The strategy of the proof follows the one of Proposition C.5, in dimension 1. The first step of symmetrization is identical. Then, we detail a few differences compared to the numerator.

First, the Lipschitz constant calculation slightly differs, as shown in Lemma C.10 below.

**Lemma C.10** (Bound on Lipschitz constant for denominator). *Let $f \in \mathcal{F}'$, and $L$ defined in Lemma B.7*

$$\|L\|_{L^2(\nu)} \leqslant C \|A\|_2 \sigma \sqrt{d} \, \exp\!\big(2 \, R^2 \|\Sigma^{1/2} A\|_2^2\big),$$

*for some constant $C > 0$.*

*Proof.* Here, we apply the exact same proof as for Lemma C.7, with $h(t, y) := e^{\langle t, y \rangle}$, using $\Big(\mathbb{E}_{Y \sim \nu} \|Y\|_2^4\Big)^{1/4} \leqslant C \sigma \sqrt{d}$. $\qquad\square$

Then, the sub-Gaussian process under consideration is now

$$Z_h = \frac{1}{\sqrt{n}} \sum_{i=1}^{n} \varepsilon_i h_x(Y_i), \tag{11}$$

(see Lemma C.4 for justification).

In dimension 1, we apply Theorem B.5 to the sub-Gaussian process $Z_h$ - defined in equation (11) - with respect to the metric space $(\mathcal{F}', \|\cdot\|_{\mathcal{L}_2(\mathbb{P}_n)})$ to bound $\mathbb{E}_\varepsilon[\sup_{h \in \mathcal{F}'} Z_h]$. Note that Dudley's entropy bound extends to $\mathbb{E}_\varepsilon[\sup_{h \in \mathcal{F}'} |Z_h|]$ using that $\mathbb{E}_\varepsilon[\sup_{h \in \mathcal{F}'} |Z_h|] = \sup_{h \in \mathcal{F}'} \max\{Z_h, -Z_h\}$ (both statement can be found in the literature). Then,

$$\mathbb{E}[\sup_{h \in \mathcal{F}'} |Z_h|] \leqslant \frac{C}{\sqrt{n}} \int_0^{\mathrm{diam}(\mathcal{F}')} \sqrt{\ln N(\delta, \mathcal{F}', L^2(\mathbb{P}_n))} \, d\delta$$

$$\leqslant \frac{C}{\sqrt{n}} \int_0^{\mathrm{diam}(B_R)\|L\|_{L^2(\mathbb{P}_n)}} \sqrt{\ln N\big(\frac{\delta}{\|L\|_{L^2(\mathbb{P}_n)}}, B_R, \|\cdot\|_2\big)} \, d\delta,$$

where we used Lemma C.2.

Taking the expectation over $Y_i$, and using the symmetrization Lemma C.6 (see Appendix C.3 for details),

$$E[\|\mathbb{P}_n - \mathbb{P}\|_{\mathcal{F}'}] \leqslant \frac{C \|A\|_2 \, \sigma \sqrt{d}}{\delta \sqrt{n}} \, e^{2 \, R^2 \|\Sigma^{1/2} A\|_2^2} := C_0'(\Sigma, A, R; \delta). \tag{12}$$

Finally we also conclude using Markov inequality. $\qquad\square$

**Part 3: Uniform concentration of the attention map (ratio bound)** The last step of the proof is to combine Propositions C.5 and C.9 to bound the attention's ratio. For the sake of clarity, we denote the empirical and continuous attention's numerator and denominator: $N_n(x) := \frac{1}{n} \sum_{i=1}^{n} y_i e^{\langle Ax, y_i \rangle}$, $D_n(x) := \frac{1}{n} \sum_{i=1}^{n} e^{\langle Ax, y_i \rangle}$, and $N(x) := \int y e^{\langle Ax, y \rangle} d\nu(y)$, $D(x) := \int e^{\langle Ax, y \rangle} d\nu(y)$. In the sequel, we omit the dependence in $x \in B_R$ most of the time to alleviate notation. Then, we can write: $f_n(x) = \|V\|_2 \cdot \frac{N_n}{D_n}$ and $f(x) = \|V\|_2 \cdot \frac{N}{D}$. Putting everything on the same denominator yields:

$$\|\frac{N_n}{D_n} - \frac{N}{D}\|_2 = \|\frac{N_n D - ND + ND - ND_n}{DD_n}\|_2 \leqslant \frac{\|N_n - N\|_2}{D_n} + \frac{\|N\|_2 \, |D_n - D|}{DD_n}.$$

To bound below the denominator $D_n(x)$ uniformly in $x$, we define the following additional events:

$$\mathcal{E}_3 = \{ \sup_{x \in B_R} |D_n(x) - D(x)| \leqslant \frac{1}{2} \}$$

$$\mathcal{E}_4 = \{ \forall x \in B_R, D_n(x) \geqslant \frac{1}{2} \} = \{ \inf_{x \in B_R} D_n(x) \geqslant \frac{1}{2} \}$$

First, notice that $\mathcal{E}_3 \subseteq \mathcal{E}_4$. Let $\omega \in \mathcal{E}_3$. For all $x \in B_R$, $D_n(x, \omega) \geqslant D(x) - |D_n(x, \omega) - D(x)| \geqslant \frac{1}{2}$, using $D(x) = \mathbb{E}[e^{\langle Ax, Y \rangle}] \geqslant 1$ by Jensen's inequality. Hence $\mathcal{E}_3 \subseteq \mathcal{E}_4$.

By Proposition C.9, if

$$C_2(R) := \frac{C'}{\delta_1} \cdot \frac{\|A\|_2 \, \sigma \sqrt{d}}{\sqrt{n}} e^{2 R^2 \|\Sigma^{1/2} A\|_2^2}$$

then the event

$$\mathcal{E}_2 = \{ \sup_{x \in B_R} |D_n(x) - D(x)| \leqslant C_2(R) \}$$

is verified with probability $1 - \delta_1$. Note that, for

$$n \geqslant n_{\min}(\delta, \Sigma, A, R, d) := 4 \frac{C'^2}{\delta_1^2} \|A\|_2^2 \sigma^2 d e^{4R^2 \|\Sigma^{1/2} A\|_2^2},$$

one has $C_2(R) \leqslant \frac{1}{2}$, hence $\mathcal{E}_2 \subseteq \mathcal{E}_3$, and therefore $\mathbb{P}(\mathcal{E}_3) \geqslant \mathbb{P}(\mathcal{E}_2) \geqslant 1 - \delta_1$. On $\mathcal{E}_3$, the event $\mathcal{E}_4$ is also verified by the inclusion $\mathcal{E}_3 \subseteq \mathcal{E}_4$, which allows us to bound $D_n(x) \geqslant \frac{1}{2}$ uniformly in $x \in B_R$. Therefore, with probability $1 - \delta_1$, and with $n \geqslant n_{min}(\delta_1, \Sigma, A, R, d)$,

$$\sup_{x \in B_R} \| \frac{N_n}{D_n} - \frac{N}{D} \|_2 \leqslant 2( \sup_{x \in B_R} \|N_n - N\|_2 + \sup_{x \in B_R} \|N\|_2 \cdot C_2(R)). \tag{13}$$

where we used $\sup_{x \in B_R}(\|N\|_2 \cdot |D_n(x) - D(x)|) \leqslant \sup_{x \in B_R} \|N\|_2 \cdot \sup_{x \in B_R} |D_n(x) - D(x)|$, and $\sup_{x \in B_R} |D_n(x) - D(x)| \leqslant C_2(R)$ on $\mathcal{E}_2$.

By Proposition C.5, the event

$$\mathcal{E}_1 = \{ \sup_{x \in B_R} \|N_n(x) - N(x)\|_2 \leqslant C_1(R) \}$$

is verified with probability $1 - \delta_2$, where

$$C_1(R) := \frac{C_1}{\delta_2} \, d \, (R \|A\|_2 \sqrt{\|\Sigma\|_2 d} + 5^{d/2}) \cdot \frac{\sqrt{\|\Sigma\|_2}}{\sqrt{n}} \cdot e^{8R^2 \|\Sigma^{1/2} A\|_2^2}. \tag{14}$$

Then, to bound $\|N\|_2$, notice that: $\|N\|_2 = \|\mathbb{E}[Y \, e^{\langle Ax, Y \rangle}]\|_2 \leqslant \mathrm{tr}(\Sigma)^{1/2} \, e^{2R^2 \|\Sigma^{\frac{1}{2}} A\|_2^2}$. Combining events $\mathcal{E}_1$ and $\mathcal{E}_2$, and for $n \geqslant n_{min}(\delta, \Sigma, A, R, d)$, we have with probability $1 - \delta$ (taking $\delta_1 = \delta_2 = \frac{\delta}{2}$),

$$\sup_{x \in B_R} \| \frac{N_n}{D_n} - \frac{N}{D} \|_2 \leqslant \|V\|_2 \cdot \frac{2C \sqrt{d} \, (R \|A\|_2 \sqrt{\|\Sigma\|_2} d + \mathrm{tr}(\Sigma)^{1/2} \|A\|_2 + 5^{d/2} \sqrt{d})}{\delta} \cdot \frac{\sqrt{\|\Sigma\|_2}}{\sqrt{n}} \cdot e^{8 R^2 \|\Sigma^{1/2} A\|_2^2}.$$

This concludes the proof of Theorem 4.2.

### C.3. Equations of Theorem 4.2

Below, we detail the derivation of some inequalities in the proof Theorem 4.2 (Appendix C.2). We recall that $\ln \mathcal{N}(\varepsilon, B_r, \|\cdot\|) \leqslant d \ln (1 + \frac{2r}{\varepsilon})$.

**Equation (8).** Let $t = \frac{\|F\|_n}{R\|L\|_n}$, then

$$
\begin{aligned}
\mathcal{I} &\leqslant 2R\,\mathbb{E}\left[\|L\|_{L^2(\mathbb{P}_n)} \int_0^{1+t} \sqrt{d\ln\left(1+\frac{2}{u}\right)}\,du\right] \\
&= 2R\,\mathbb{E}\left[\|L\|_{L^2(\mathbb{P}_n)}\left(\int_0^1 \sqrt{d\ln\left(1+\frac{2}{u}\right)}\,du + \int_1^{1+t} \sqrt{d\ln\left(1+\frac{2}{u}\right)}\,du\right)\right] \\
&\leqslant 2R\,\mathbb{E}\left[\|L\|_{L^2(\mathbb{P}_n)}\left(\int_0^1 \frac{\sqrt{2d}}{\sqrt{u}}\,du + \int_1^{1+t} \sqrt{d\ln 3}\,du\right)\right] \\
&\leqslant 2R\,\mathbb{E}\left[\|L\|_{L^2(\mathbb{P}_n)}\left(2\sqrt{2d} + t\sqrt{d\ln 3}\right)\right] \\
&= 4\sqrt{2d}\,R\,\mathbb{E}[\|L\|_{L^2(\mathbb{P}_n)}] + 2\sqrt{d\ln 3}\,\mathbb{E}[\|F\|_{L^2(\mathbb{P}_n)}].
\end{aligned}
$$

**Equation (9).** Let $t' = \frac{R\|L\|_n}{\|F\|_n}$, then by the same arguments as above,

$$
\begin{aligned}
\mathcal{J} &\leqslant \mathbb{E}\left[2\|F\|_{L^2(\mathbb{P}_n)} \int_0^{1+\frac{R\|L\|_n}{\|F\|_n}} \sqrt{d\ln\left(1+\frac{2}{u}\right)}\,du\right] \\
&\leqslant \mathbb{E}\left[\|F\|_{L^2(\mathbb{P}_n)}\left(4\sqrt{2d} + 2t'\sqrt{d\ln 3}\right)\right] \\
&= 4\sqrt{2d}\,\mathbb{E}[\|F\|_{L^2(\mathbb{P}_n)}] + 2R\,\sqrt{d\ln 3}\,\mathbb{E}[\|L\|_{L^2(\mathbb{P}_n)}].
\end{aligned}
$$

**Equation (12).** Taking the expectation over $Y_i$, and using the symmetrization Lemma C.6,

$$
\begin{aligned}
\mathbb{E}[\|\mathbb{P}_n - \mathbb{P}\|_{\mathcal{F}}] &\leqslant \mathbb{E}_Y\mathbb{E}_\varepsilon\left[\sup_{h\in\mathcal{F}'} |Z_h| \mid Y_1,...,Y_n\right] \\
&\leqslant \frac{C}{\sqrt{n}}\mathbb{E}_Y\left[\int_0^{2R\|L\|_{L^2(\mathbb{P}_n)}} \sqrt{\ln\mathcal{N}\left(\frac{\delta}{\|L\|_{L^2(\mathbb{P}_n)}}, B_R, \|\cdot\|_2\right)}\,d\delta\right] \\
&\leqslant \frac{C}{\sqrt{n}}\mathbb{E}_Y\left[\int_0^2 \sqrt{\ln\mathcal{N}\left(Ru, B_R, \|\cdot\|_2\right)}\,R\,\|L\|_{L^2(\mathbb{P}_n)}\,du\right] \\
&\leqslant \frac{C}{\sqrt{n}}R\,\mathbb{E}_Y\left[\|L\|_{L^2(\mathbb{P}_n)}\int_0^2 \sqrt{d\ln\left(1+\frac{2}{u}\right)}\,du\right] \\
&\leqslant \frac{C}{\sqrt{n}}R\int_0^2 \sqrt{\frac{2d}{u}}\,du\,\mathbb{E}_Y[\|L\|_{L^2(\mathbb{P}_n)}] \\
&\leqslant \frac{c}{\delta}\frac{\left(\mathbb{E}_Y[L(Y_1)^2]\right)^{1/2}}{\sqrt{n}} \\
&= \frac{C\,\|A\|_2\,\sigma\sqrt{d}}{\delta\sqrt{n}}\,\exp\left(2\,R^2\|\Sigma^{1/2}A\|_2^2\right).
\end{aligned}
$$

The last inequality follows directly from Lemma B.8 and Lemma C.7.

## C.4. Generalization of Theorem 4.2

**Theorem C.11** (Uniform convergence of the attention map for sub-Gaussian tokens over a general set). *Let $X \sim \nu$ be centered and sub-Gaussian with parameter matrix $\Sigma \succ 0$. Let $B$ an invertible matrix, let $R > 0$ and denote $K_R := \{x \in \mathbb{R}^d \mid \|Bx\|_2 \leqslant R\}$. For any $\delta > 0$, there exists a constant $C > 0$, such that for $n \geqslant n_{min}(\delta, \Sigma, A, R) := 4e^{2R^2\|\Sigma^{1/2}AB^{-1}\|_2^2}\left(\frac{1}{\delta}-1\right)$, with probability at least $1 - \delta$,*

$$
\sup_{x\in K_R} \left\|f_n(x) - f(x)\right\|_2 \leqslant q_{(\Sigma,A,V,R,d,\delta)} \cdot \frac{e^{8\,R^2\|\Sigma^{1/2}AB^{-1}\|_2^2}}{\sqrt{n}}, \tag{15}
$$

*where* $q_{(\Sigma, A, V, R, d, \delta)} := \frac{C\sqrt{d}}{\delta}\|V\|_2 \cdot \|\Sigma\|_2^{1/2} \cdot (R\|A\|_2\|\Sigma\|_2^{1/2}d + \mathrm{tr}(\Sigma)^{1/2}\|A\|_2 + 5^{d/2}\sqrt{d}).$

*Proof.* It suffices to apply Theorem 4.2 with $\tilde{A} := AB^{-1}$, and taking the supremum over $y \in \mathrm{Im}B \cap B_R$, which is a compact set. $\qquad\square$

### C.5. Sharp bounds for two Dirac masses

In this Appendix, we show that when $\nu$ is a sum of two Dirac masses, the bound provided in Theorem 4.2 is not sharp with respect to the parameters $R$ and $R_0$, and one can actually prove a sharp dependence rate in these parameters. We consider the case $d = 1$, $A = \beta\mathrm{Id}$ (i.e., $A = \beta \in \mathbb{R}$ is a scalar). Let $\nu = p\delta_{-R_0} + (1-p)\delta_{R_0}$ for some $p \in (0,1)$ and $R_0 > 0$. When $p = 1/2$, the sub-Gaussian parameter $\Sigma$ is equal to $R_0^2$. Therefore Theorem 4.2 yields a bound of the form $\frac{e^{8\beta^2 R_0^2 R^2}}{\sqrt{n}}$. We show below a sharp bound which is polynomial in $R_0$ and does not depend on $R$. Emphasising the dependence in $p$ and $R_0$, we write $f(p, R_0, x)$ and $f_n(p, R_0, x)$ instead of $f(x)$ and $f_n(x)$.

**Theorem C.12.** *There exist $C, C'$ universal (in particular, independent of $p, \delta, R_0$) such that for any $\delta > 0$, any $p \in (0,1), R, R_0 > 0$, and any $n \geqslant \frac{C R_0^2}{\delta \min(p^6, (1-p)^6)}$, there holds with probability at least $1 - \delta$*

$$C'\delta\sqrt{p(1-p)}\frac{R_0^2}{\sqrt{n}} \leqslant \sup_{x \in B_R} |f(p, R_0, x) - f_n(p, R_0, x)| \leqslant \sqrt{2\log(2/\delta)}\frac{R_0^2}{\sqrt{n}}. \tag{16}$$

*for any $R > 0$.*

*Proof.* We have

$$f(p, R_0, x) = \frac{(1-p)e^{\beta R_0 x}R_0 - pe^{-\beta R_0 x}R_0}{(1-p)e^{\beta R_0 x} + pe^{-\beta R_0 x}} = R_0 - 2R_0\frac{1}{1 + \frac{1-p}{p}e^{2\beta R_0 x}}. \tag{17}$$

Let us now compute $f_n(p, R_0, x)$ for some $n \in \mathbb{N}$. Since $\nu_n$ is obtained by taking $n$ random draws according to $\nu$, we denote by $m$ the (random) number of times that $-R_0$ appears, and thus $n - m$ is the number of times that $R_0$ appears. Thus $\nu_n = \frac{m}{n}\delta_{-R_0} + \frac{n-m}{n}\delta_{R_0}$. We let $p' = \frac{m}{n}$. It is not difficult to verify that

$$f_n(p, R_0, x) = f(p', R_0, x).$$

Our goal is to estimate the quantity $|f(p', R_0, x) - f(p, R_0, x)|$.

**Lower bound.** We have $\sup_{x \in B_R} |f(p, R_0, x) - f_n(p, R_0, x)| \geqslant |f(p, R_0, 0) - f_n(p, R_0, 0)| = 2R_0|p - p'|$ according to (17). By Berry-Esseen's theorem we have for some $c > 0$ independent of $R_0$ and $p$

$$\forall \varepsilon > 0, \qquad \mathbb{P}\left(|p' - p| \geqslant \varepsilon\frac{R_0\sqrt{p(1-p)}}{\sqrt{n}}\right) \geqslant 1 - c\left(\varepsilon + \frac{1}{\sqrt{p(1-p)n}}\right).$$

Hence, with probability $\geqslant 1 - \delta$, we have $|p' - p| \geqslant \frac{\delta}{c}\frac{R_0\sqrt{p(1-p)}}{\sqrt{n}} - \frac{R_0}{n}$. Our assumption on $n$ then implies that $|p' - p| \geqslant \frac{\delta}{2c}\frac{R_0\sqrt{p(1-p)}}{\sqrt{n}}$ (by choosing $C$ large) and finally the lower bound in (16) follows.

**Upper bound.** By Taylor's remainder formula,

$$|f_n(p', R_0, x) - f(p, R_0, x)| \leqslant |p' - p|\left|\frac{\partial f}{\partial p}(p, R_0, x)\right| + \frac{1}{2}|p' - p|^2\left\|\frac{\partial^2 f}{\partial p^2}(\cdot, R_0, x)\right\|_{L^\infty([p,p'])}.$$

It follows from (17) that

$$\frac{\partial f}{\partial p}(p, R_0, x) = -\frac{2R_0}{(pe^{-\beta R_0 x} + (1-p)e^{\beta R_0 x})^2} \qquad \text{and} \qquad \frac{\partial f^2}{\partial p^2}(q, R_0, x) = \frac{4R_0 e^{2\beta R_0 x}(1 - e^{2\beta R_0 x})}{(q + (1-q)e^{2\beta R_0 x})^3}.$$

Hence

$$\left|\frac{\partial f}{\partial p}(p, R_0, x)\right| \leqslant R_0 \qquad \text{and} \qquad \left|\frac{\partial f^2}{\partial p^2}(q, R_0, x)\right| \leqslant 4R_0\max(q^{-3}, (1-q)^{-3})$$

(the second inequality follows by treating separately then cases $x \geqslant 0$ and $x \leqslant 0$). Due to Hoeffding's inequality we have

$$\mathbb{P}(|p' - p| \geqslant \varepsilon) \leqslant 2 \exp(-2n\varepsilon^2 / R_0^2)$$

hence on an event $A_\delta$ of probability $\geqslant 1 - \delta$ we have

$$|p' - p| \leqslant \frac{R_0 \sqrt{\log(2/\delta)}}{\sqrt{2n}}. \tag{18}$$

Our assumption on $n$ implies that $n \geqslant \frac{2R_0^2 \log(2/\delta)}{\min(p^2,(1-p)^2)}$ hence we have under the event $A_\delta$ that $p' \geqslant \frac{p}{2}$ and $1 - p' \geqslant \frac{1-p}{2}$. We deduce easily that under $A_\delta$ there holds

$$\left\| \frac{\partial^2 f}{\partial p^2}(\cdot, R_0, x) \right\|_{L^\infty([p,p'])} \leqslant 32 R_0 \left( \frac{1}{p^3} + \frac{1}{(1-p)^3} \right). \tag{19}$$

In particular, with probability $1 - \delta$, up to choosing a larger $C_1$,

$$|f(p', R_0, x) - f(p, R_0, x)| \leqslant \frac{R_0^2 \sqrt{\log(2/\delta)}}{\sqrt{2n}} + 8 \frac{R_0^3 \log(2/\delta)}{n} \left( \frac{1}{p^3} + \frac{1}{(1-p)^3} \right)$$

By our assumption on $n$, we have $\frac{R_0^2 \sqrt{\log(2/\delta)}}{\sqrt{2n}} \geqslant 8 \frac{R_0^3 \log(2/\delta)}{n} \left( \frac{1}{p^3} + \frac{1}{(1-p)^3} \right)$, thus

$$|f(p', R_0, x) - f(p, R_0, x)| \leqslant \frac{R_0^2 \sqrt{2\log(2/\delta)}}{\sqrt{n}}.$$

This gives the upper bound in (16). $\qquad\square$

*Remark* C.13. The same example of two Dirac masses can be considered in any dimension, and the lower and upper bounds are obviously the same, without any dimensional dependence.

## C.6. Proof of Theorem 5.3

*Proof.* Let $h$ a Lipschitz function of parameter $L_0$. By considering $h - h(0)$ instead of $h$ if necessary, we can assume without loss of generality that $h(0) = 0$.

**1. Decomposition of the error.** We reformulate the mean error to facilitate our analysis. The quantity of interest is

$$\left\| \mathbb{E}_n[h \circ f_n(\hat{X})] - \mathbb{E}[h \circ f(X)] \right\|_2.$$

Applying the triangle inequality and adding–subtracting identical terms, we obtain

$$\left\| \mathbb{E}_n[h \circ f_n(\hat{X})] - \mathbb{E}_x[h \circ f(X)] \right\|_2$$
$$= \left\| \mathbb{E}_n[h \circ f_n(\hat{X})] - \mathbb{E}_n[h \circ f(X)] + \mathbb{E}_n[h \circ f(X)] - \mathbb{E}[h \circ f(X)] \right\|_2$$
$$\leqslant \mathcal{I} + \mathcal{J}$$

where

$$\mathcal{I} = \left\| \mathbb{E}_n[h \circ f_n(\hat{X}) - h \circ f(X)] \right\|_2$$

and

$$\mathcal{J} = \left\| \mathbb{E}_n[h \circ f(X)] - \mathbb{E}[h \circ f(X)] \right\|_2$$

**2. Analysis of the second term $\mathcal{J}$.** To bound $\mathcal{J}$, we follow the same methodology as in the proof of Proposition D.3: we apply the CLT, followed by the derivation of an asymptotic confidence bound using the quantile of the $\chi^2(d)$ law (see Appendix D.2 for more details). We obtain that, with probability at least $1 - \delta$,

$$\mathcal{J} \leqslant \frac{q_{d,1-\delta}^{1/2} \lambda_{\max}(\mathrm{Cov}(h \circ f(X)))^{1/2}}{\sqrt{n}},$$

where the bound scales with the $\chi_2$ quantile $q_{d,1-\delta}$, and the maximum variance direction of $h \circ f(X)$.

Let $\alpha := 2 \|V\|_2 \|\Sigma^{1/2}\|_2 \|\Sigma^{1/2}A\|_2$, and let us show that

$$\lambda_{\max}(\mathrm{Cov}(h \circ f(X))) \leqslant \alpha^2 L_0^2 \cdot \|\Sigma\|_2,$$

from which $\mathcal{J} = \mathcal{O}(n^{-1/2})$ follows.

Using the at-most-linear growth of $\Gamma_\nu$ of Lemma D.1, we get,

$$
\begin{aligned}
\lambda_{\max}&(\mathrm{Cov}(h \circ f(X))) \\
&\leqslant \mathrm{tr}(\mathrm{Cov}(h \circ f(X))) \\
&= \mathrm{tr}(\mathbb{E}[(h \circ f(X) - \mathbb{E}[h \circ f(X)])(h \circ f(X) - \mathbb{E}[h \circ f(X)])^\top]) \\
&= \mathbb{E}[\mathrm{tr}((h \circ f(X) - \mathbb{E}[h \circ f(X)])^\top (h \circ f(X) - \mathbb{E}[h \circ f(X)]))] \\
&= \mathbb{E}[\|h \circ f(X) - \mathbb{E}[h \circ f(X)]\|_2^2] \\
&\leqslant \mathbb{E}[\|h \circ f(X)\|_2^2] \leqslant \alpha^2 L_0^2 \cdot \mathbb{E}[\|X\|_2^2] = \alpha^2 L_0^2 \cdot \|\Sigma\|_2.
\end{aligned}
$$

**3. Analysis of the first term $\mathcal{I}$.** To bound the first term $\mathcal{I}$, the key insight is to split the analysis based on whether $\|Bx_i\|_2$ exceeds a threshold $R$ or not, where $B$ is any matrix satisfying the hypothesis of Theorem C.11. The threshold on $R$ will then be optimized later. We have

$$\mathcal{I} = \left\| \frac{1}{n} \sum_{i=1}^n h \circ f_n(x_i) - h \circ f(x_i) \right\|_2 \leqslant \mathcal{I}_1 + \mathcal{I}_2 + \mathcal{I}_3, \tag{20}$$

where

$$\mathcal{I}_1 = \frac{1}{n} \sum_{|Bx_i| < R} \left\| h \circ f_n(x_i) - h \circ f(x_i) \right\|_2$$

$$\mathcal{I}_2 = \frac{1}{n} \sum_{|Bx_i| > R} \left\| h \circ f_n(x_i) \right\|_2$$

$$\mathcal{I}_3 = \frac{1}{n} \sum_{|Bx_i| > R} \left\| h \circ f(x_i) \right\|_2.$$

For $\mathcal{I}_1$, we apply Theorem 4.2, and get with probability at least $1 - \delta_1$, For $\mathcal{I}_1$, we apply Theorem C.11, and get with probability at least $1 - \delta_1$,

$$\mathcal{I}_1 \leqslant \frac{1}{n} \sum_{|Bx_i| < R} L_0 \cdot \sup_{x \in B_R} \left\| f_n(x) - f(x) \right\|_2 \leqslant L_0 \cdot \frac{C_1(\Sigma, A, B, V, R, \delta_1, d)}{\sqrt{n}}.$$

For $\mathcal{I}_2$, we prove the following Proposition,

**Proposition C.14.** *Recall $x_1, \ldots, x_n$ are i.i.d. $\mathbb{R}^d$-valued sub-Gaussian random vectors, with parameter $\sqrt{\|\Sigma\|_2}$. Let $B$ an invertible matrix satisfying the hypothesis of Theorem C.11, let $\sigma_B := \sqrt{\|B\Sigma B^\top\|_2} = \|\Sigma^{1/2}B^\top\|_2$ and $R > 0$. For all $n > 0$, and some constants $c, C > 0$, with probability at least $1 - \delta$,*

$$\frac{1}{n} \sum_{\|Bx_i\|_2 > R} \|f_n(x_i)\|_2 \leqslant \left( e^{-c(\frac{R}{\sigma_B} - C\sqrt{d})^2} + \sqrt{\frac{1}{2n} \ln\left(\frac{4}{\delta}\right)} \right) \cdot c\sigma_B \left( \sqrt{d} + \sqrt{\ln \frac{2n}{\delta}} \right).$$

*Proof.* Let

$$N_R := \sum_{i=1}^n \mathbf{1}_{\{\|Bx_i\|_2 > R\}}, \qquad M_n := \max_{1 \leqslant i \leqslant n} \|x_i\|_2, \qquad p_R := \mathbb{P}(\|BX\|_2 > R).$$

Since $f_n(x)$ is a convex combination of $\{x_j\}_{j=1}^n$,

$$\left\| f_n(x) \right\|_2 \leqslant M_n.$$

Hence,

$$\frac{1}{n} \sum_{\|Bx_i\|_2 > R} \left\| f_n(x_i) \right\|_2 \leqslant \frac{N_R}{n} M_n. \tag{21}$$

Our task therefore reduces to finding high-probability bounds for: (i) the maximum norm $M_n$, and (ii) the fraction $N_R/n$ of samples with norm above $R$. We begin with a lemma that controls the norm of a sub-Gaussian vector $X$.

**Lemma C.15** ((Vershynin, 2018) see Exercise 6.3.5 p.114, 1st ed.)**.** *There exist universal constants $C, c > 0$ such that for all $R > 0$, if $X$ is a sub-Gaussian random vector in $\mathbb{R}^d$ with sub-Gaussian norm at most $\sqrt{\|\Sigma\|_2}$, then $BX$ is sub-Gaussian with sub-Gaussian norm at most $\sigma_B := \sqrt{\|B\Sigma B^\top\|_2} = \|\Sigma^{1/2} B^\top\|_2$ and*

$$p_R := \mathbb{P}(\|BX\|_2 > R) \leqslant e^{-c(\frac{R}{\sigma_B} - C\sqrt{d})^2}.$$

*with explicitly, $c = \frac{1}{4}$ and $C = 4$.*

Then we bound separately $\frac{N_R}{n}$ and $M_n$.

**Step 1: Concentration of $\frac{N_R}{n}$** The indicators $\mathbf{1}_{\{\|Bx_i\|_2 > R\}}$ are i.i.d. Bernoulli($p_R$). By Hoeffding's inequality, for every $t > 0$,

$$\mathbb{P}\left( \left| \frac{N_R}{n} - p_R \right| \geqslant t \right) \leqslant 2 \exp(-2nt^2).$$

Equivalently, for any $\delta_1 \in (0, 1)$, with probability at least $1 - \delta_1$,

$$\left| \frac{N_R}{n} - p_R \right| \leqslant \sqrt{\frac{1}{2n} \ln\left(\frac{2}{\delta_1}\right)}.$$

Then, we move on to bounding the maximum of the $x_i$'s norm.

**Step 2: Union bound for $M_n = \max_{1 \leqslant i \leqslant n} \|x_i\|_2$.** The union bound gives

$$\mathbb{P}(M_n \geqslant t) \leqslant \mathbb{P}\left( \bigcup_{i=1}^n \{\|x_i\|_2 \geqslant t\} \right) \leqslant \sum_{i=1}^n \mathbb{P}(\|x_i\|_2 \geqslant t) = n\, \mathbb{P}(\|X\|_2 \geqslant t).$$

Applying Lemma C.15 we get,

$$\mathbb{P}(M_n \geqslant t) \leqslant n\, e^{-\left(\frac{t}{c\,\sigma_B} - \sqrt{d}\right)^2}.$$

For any $\delta_2 \in (0, 1)$, solving $n\, e^{-(\frac{t}{c\,\sigma_B} - \sqrt{d})^2} = \delta_2$ yields, with probability at least $1 - \delta_2$,

$$M_n \leqslant c\sigma_B\left(\sqrt{d} + \sqrt{\ln \frac{n}{\delta_2}}\right).$$

Combining this result with the step 1, we get with probability at least $1 - \delta_1 - \delta_2$,

$$\frac{1}{n} \sum_{\|Bx_i\|_2 > R} \left\| f_n(x_i) \right\|_2 \leqslant \left( e^{-c(\frac{R}{\sigma_B} - C\sqrt{d})^2} + \sqrt{\frac{1}{2n} \ln\left(\frac{2}{\delta_1}\right)} \right) \cdot c\sigma\left(\sqrt{d} + \sqrt{\ln \frac{n}{\delta_2}}\right).$$

Taking $\delta_1 = \delta_2 = \frac{\delta}{2}$ concludes the proof of Proposition C.14. $\qquad\square$

We deduce from Proposition C.14, and using the same bound for $N_R/n$ as in Proposition C.14, that with probability at least $1 - \delta_2$

$$\mathcal{I}_2 \leqslant \frac{1}{n} \sum_{\|Bx_i\|_2 > R} L_0 \cdot \|f_n(x_i)\|_2$$

$$\leqslant L_0 \cdot \frac{N_R}{n} M_n$$

$$\leqslant L_0 \cdot C_2(\Sigma, R, B, \delta_2, d, n)$$

where

$$C_2(\Sigma, R, B, \delta_2, d, n) = \left( e^{-c\left(\frac{R}{\sigma_B} - C\sqrt{d}\right)^2} + \sqrt{\frac{1}{2n} \ln\left(\frac{4}{\delta_2}\right)} \right) \cdot c\sigma_B \left( \sqrt{d} + \sqrt{\ln \frac{2n}{\delta_2}} \right).$$

For $\mathcal{I}_3$, applying Lemma D.1:

$$\frac{1}{n} \sum_{i=1}^{n} \mathbf{1}_{\{\|Bx_i\|_2 > R\}} \|h \circ f(x_i)\|_2 \leqslant L_0 \cdot \underbrace{2 \|V\|_2 \sqrt{\|\Sigma\|_2} \|\Sigma^{1/2}A\|_2}_{\alpha} \cdot \frac{1}{n} \sum_{i=1}^{n} \mathbf{1}_{\{\|Bx_i\|_2 > R\}} \|x_i\|_2,$$

$$\leqslant L_0 \cdot \alpha \cdot \frac{N_R}{n} M_n$$

where we used the notations of Proposition C.14. The same bound on $\frac{N_R}{n} M_n$ holds and gives, with probability $1 - \delta_3$

$$\frac{1}{n} \sum_{i=1}^{n} \mathbf{1}_{\{\|Bx_i\|_2 > R\}} \|h \circ f(x_i)\|_2 \leqslant L_0 \cdot \alpha \cdot C_2(\Sigma, R, B, \delta_3, d, n).$$

We return to our initial inequality (20), where, with probability $1 - \delta$ (taking $\delta_1 = \delta_2 = \delta_3 = \frac{\delta}{3}$, and denoting $\delta' = \frac{\delta}{3}$),

$$\left\| \mathbb{E}_n[h \circ f_n(\hat{X}) - h \circ f(X)] \right\|_2 \leqslant \mathcal{I}_1 + \mathcal{I}_2 + \mathcal{I}_3 \leqslant f(R).$$

Here, $f(R) = L_0 \cdot \frac{C_1(\Sigma, A, B, R, \delta', d)}{\sqrt{n}} + L_0 \cdot (1 + \alpha) \cdot C_2(\Sigma, A, B, R, \delta', n)$ with,

$$C_1(\Sigma, A, B, V, R, \delta', d) = P_1(\Sigma, A, B, V, R, \delta', d) \cdot \exp\left\{ 8\|\Sigma^{1/2}AB^{-1}\|^2 R^2 \right\}$$

$$C_2(\Sigma, R, B, \delta', d, n) = \left( e^{-c\left(\frac{R}{\sigma_B} - C\sqrt{d}\right)^2} + \sqrt{\frac{1}{2n} \ln\left(\frac{4}{\delta'}\right)} \right) \cdot c\sigma_B \left( \sqrt{d} + \sqrt{\ln \frac{2n}{\delta'}} \right).$$

where

$$P_1(\Sigma, A, V, R, \delta', d) = \frac{C\sqrt{d}}{\delta} \|V\|_2 \cdot \|\Sigma\|_2^{1/2} \cdot (R\|A\|_2\|\Sigma\|_2^{1/2}d + \text{tr}(\Sigma)^{1/2}\|A\|_2 + 5^{d/2}\sqrt{d})$$

is linear in $R$. To make the two terms of $f(R)$ of the same order, we choose

$$R^\star = \sqrt{\frac{\ln(n)}{16 \|\Sigma^{1/2}AB^{-1}\|^2 + \frac{2c}{\sigma_B^2}}}.$$

To be valid, our upper bound on $\mathcal{I}_1$ derived above requires (since we applied Theorem 4.2) that $n \geqslant 4e^{2R^{\star 2}\|\Sigma^{1/2}AB^{-1}\|_2^2}\left(\frac{1}{\delta} - 1\right)$. This is the case for the choice $R = R^\star$ for $n$ large enough since

$$4e^{2R^{\star 2}\|\Sigma^{1/2}AB^{-1}\|_2^2}\left(\frac{1}{\delta} - 1\right) \leqslant 4\left(\frac{1}{\delta} - 1\right)n^{1/8} \ll n.$$

Noting $\tau_B := \|\Sigma^{\frac{1}{2}}AB^{-1}\|^2$, we get the bound

$$f(R^\star) = \left( K_1 + K_2 \cdot \sqrt{\ln(n)} \right) n^{-\frac{1}{2}\left(1 - \frac{16\tau_B^2}{16\tau_B^2 + \frac{2c}{\sigma_B^2}}\right)}.$$

Recalling that $c = \frac{1}{4}$ (see Lemma C.15), we get $f(R^\star) = \Theta(n^{-\frac{1}{2(1+32 \cdot \rho_B)}} \sqrt{\ln n})$ with $\rho_B := \|\Sigma^{1/2} A B^{-1}\|_2^2 \|\Sigma^{1/2} B^\top\|_2^2$. The final step is to optimize this result over B. Notice that, for any invertible $B$,

$$\rho_B \geqslant \|\Sigma^{\frac{1}{2}} A \Sigma^{\frac{1}{2}}\|_2^2 =: H^2$$

with equality attained for $B = \Sigma^{-1/2}$, which is indeed invertible. The final rate becomes

$$f(R^\star) = \Theta(n^{-\frac{1}{2(1+32 \cdot H^2)}} \sqrt{\ln n})$$

with $H = \|\Sigma^{\frac{1}{2}} A \Sigma^{\frac{1}{2}}\|_2$, which we call the horizon parameter. This concludes the proof of Theorem 5.3. $\qquad \square$

## C.7. Proof of Proposition 5.5

We consider a set of tokens $X_1, \ldots, X_n \overset{\text{iid}}{\sim} \mathcal{N}(0, \sigma^2)$. Let $M_n^+ := \max_{j \leqslant n} X_j$ and $M_n^- := \min_{j \leqslant n} X_j$ be the maximum and minimum of the token set $(X_i)_{i \in [1,n]}$.

The hardmax-attention is defined by,

$$f_n(X_i) := \begin{cases} M_n^+ & \text{if } X_i > 0, \\ M_n^- & \text{if } X_i < 0. \end{cases}$$

This corresponds to a limit behavior where $A = +\infty$, and thus the attention map $f_n(X_i)$ selects only the token with same sign as $X_i$ and largest modulus.

The empirical mean is

$$\bar{f}_n := \frac{1}{n} \sum_{i=1}^n f_n(X_i) = \frac{I_+}{n} M_n^+ + \frac{I_-}{n} M_n^-$$

where $I_+ = \#\{i : X_i > 0\}$, and $I_- = n - I_+$ denote the number of positive and negative tokens $X_i$. Note that $I_+ \sim \text{Bin}(n, \frac{1}{2})$.

Our main result in the hardmax regime derives the bias and standard deviation of the absolute value of the empirical mean, both of which are of order $1/\sqrt{\ln(n)}$.

*Remark* C.16. Notice that the symmetry of $X$ yields $\mathbb{E}[\bar{f}_n] = 0$.

In order to prove Proposition 5.5, let's first decompose the empirical mean into two components:

$$\bar{f}_n = \frac{I_+}{n} M_n^+ + \frac{I_-}{n} M_n^- = \frac{1}{2}(M_n^+ + M_n^-) + \left(\frac{2I_+ - n}{2n}\right)(M_n^+ - M_n^-).$$

Setting

$$V_n := \frac{1}{2}(M_n^+ + M_n^-), \quad U_n := \left(\frac{2I_+ - n}{2n}\right)(M_n^+ - M_n^-), \tag{22}$$

we get $\bar{f}_n = V_n + U_n$.

The key insight is that $V_n$ captures the main asymptotic behavior while $U_n$ represents a smaller-order deviation term.

Before proving Proposition 5.5, we establish several technical results on extreme values of Gaussian random variables.

### C.7.1. EXTREME VALUE ANALYSIS

In this subsection, we derive precise asymptotic rates for $V_n$ and establish bounds on $U_n$. Extreme value theory provides weak convergence of the maximum and minimum (Lemmas C.21 and C.22 in the Appendix). To obtain convergence in expectation, we establish uniform integrability of both extremes in Lemma C.17.

Let $a_n$ and $b_n$ be the scaling parameters of extreme values defined by

$$a_n := \sigma(\sqrt{2 \ln n} - \frac{1}{2}\sqrt{2 \ln n} \, (\log \log n + \log 4\pi)) \quad \text{and} \quad b_n = \frac{\sigma}{\sqrt{2 \ln n}}.$$

**Lemma C.17** (Uniform Integrability of Extremes). *Let $Z_n = \frac{M_n^+ - a_n}{b_n}$ and $Y_n = -\frac{M_n^- + a_n}{b_n}$. Then $\{Z_n\}_{n \geqslant 1}$ and $\{Y_n\}_{n \geqslant 1}$ are uniformly integrable.*

The definition of uniform integrability is recalled in Appendix C.8.

*Proof of Lemma C.17.* We establish the result for $Z_n$ in several steps. The same argument applies to $Y_n$ by symmetry.

*Step 1:* From equation (1.5) in (Tanguy, 2015), we have the concentration inequality:

$$\mathbb{P}(|M_n^+ - \mathbb{E}[M_n^+]| > t) \leqslant 6e^{-ct\sqrt{\ln n}} \tag{23}$$

*Step 2:* We show that $\sup_n \mathbb{E}[|Z_n|] < \infty$.

Let $M_n'$ be an independent copy of $M_n^+$ and set $Z_n' = \frac{M_n' - a_n}{b_n}$. From Step 1:

$$\sup_n \frac{1}{b_n} \mathbb{E}[|M_n^+ - \mathbb{E}[M_n^+]|] < \infty. \tag{24}$$

Using the convexity of $|\cdot|$ and Jensen's inequality,

$$\mathbb{E}_{Z_n'}[|Z_n - Z_n'|] \geqslant |Z_n - \mathbb{E}[Z_n']|.$$

Therefore,

$$\mathbb{E}[|Z_n|] \leqslant \mathbb{E}[|Z_n - Z_n'|] + |\mathbb{E}[Z_n']|$$

From (24), we can deduce that $\sup_n \mathbb{E}[|Z_n - Z_n'|] < \infty$. Indeed

$$|Z_n - Z_n'| = \frac{1}{b_n}|M_n - M_n'| \leqslant \frac{1}{b_n}|M_n - \mathbb{E}[M_n]| + \frac{1}{b_n}|M_n' - \mathbb{E}[M_n']|.$$

Since, on the other hand $\sup_n |\mathbb{E}[Z_n]| < \infty$, this gives $\sup_n \mathbb{E}[|Z_n|] < \infty$.

*Step 3:* We transfer the concentration around $\mathbb{E}[M_n^+]$ to a concentration around $a_n$.

Define $\delta_n = \mathbb{E}[M_n^+] - a_n$. Then $|\delta_n| = O(b_n)$, so $|\delta_n| \to 0$. From the event inclusion

$$\{|M_n^+ - a_n| > t\} \subset \{|M_n^+ - \mathbb{E}[M_n^+]| > t - |\delta_n|\}$$

and (23), we deduce

$$\mathbb{P}(|M_n^+ - a_n| > t) \leqslant 6e^{-c(t-|\delta_n|)\sqrt{\ln n}}.$$

For $n$ large enough, it holds for all $t \geqslant 1$ that $t - |\delta_n| \geqslant t/2$, giving

$$\mathbb{P}(|Z_n| > t) \leqslant 6e^{-c't}.$$

*Step 4:* Using the tail integral formula,

$$\mathbb{E}[|Z_n| \mathbf{1}_{\{|Z_n| > A\}}] = \int_A^{+\infty} \mathbb{P}(|Z_n| > x)\,dx \leqslant \int_A^{+\infty} 6e^{-c'x}dx = \frac{6}{c'}e^{-c'A} \to 0$$

as $A \to \infty$, uniformly in $n$. $\qquad\square$

**Lemma C.18** (Asymptotic behavior of $V_n$). *There holds*

$$\mathbb{E}|V_n| = \frac{\ln(4)\,\sigma}{2\sqrt{2\ln n}}[1 + o(1)] \quad and \quad \mathbb{E}[V_n^2] = \frac{\sigma^2 \pi^2}{24\ln n}[1 + o(1)].$$

*Proof.* The random variables $Z_n = \frac{M_n^+ - a_n}{b_n}$ and $Y_n = -\frac{M_n^- + a_n}{b_n}$ jointly converge weakly to the joint distribution of two independent Gumbel random variables, which we denote $G_+$ and $G_-$ (see Lemma C.21 in the Appendix).

By Lemma C.17, both $|Z_n|$ and $|Y_n|$ are asymptotically uniformly integrable. Since $|Z_n - Y_n| \leqslant |Z_n| + |Y_n|$, it follows that $|Z_n - Y_n|$ is also asymptotically uniformly integrable. Combining the weak convergence of the sum (Lemma C.22)

with the dominated convergence theorem (([van der Vaart, 1998](#)), Theorem 2.20, Section 2.5), we conclude that $|Z_n - Y_n|$ converges in expectation.

Recall that $V_n = \frac{M_n^+ + M_n^-}{2}$, which can be rewritten as $V_n = \frac{\sigma(Z_n - Y_n)}{2\sqrt{2\ln n}}$. We can therefore deduce that

$$\mathbb{E}\left[\frac{2\sqrt{2\ln n}|V_n|}{\sigma}\right] \to \mathbb{E}\left[|G_+ - G_-|\right]. \tag{25}$$

Since $L := G_+ - G_-$ follows a centered logistic law, we have

$$\mathbb{E}[|L|] = \ln 4,$$

which gives:

$$\lim_{n\to\infty} \frac{2\sqrt{2\ln n}}{\sigma}\mathbb{E}|V_n| = \ln 4.$$

The variance result follows similarly using $\mathrm{Var}(L) = \pi^2/3$. $\qquad\square$

**Lemma C.19** (Upper bound on $U_n$). *Let $U_n$ and $V_n$ defined in* (22)*, it holds:*

$$\mathbb{E}[|U_n|] \leqslant \sigma\sqrt{\frac{\ln n}{n}}, \quad \textit{and} \quad \mathbb{E}[U_n^2] = O\left(\frac{\ln n}{n}\right).$$

*Proof.* Using $\mathrm{Var}(2I_+ - n) = n$ and $\mathbb{E}[|M_n^+ - M_n^-|^2] = O(a_n^2)$, we have:

$$\mathbb{E}[|U_n|] \leqslant \left(\mathbb{E}\left(\left|\frac{2I_+ - n}{2n}\right|^2\right) \cdot \mathbb{E}[|M_n^+ - M_n^-|^2]\right)^{1/2} \leqslant \sqrt{\frac{1}{4n}} \cdot 2a_n = \sigma\sqrt{\frac{\ln n}{n}}.$$

For the second moment, using that $\mathbb{E}[|M_n^+ - M_n^-|^4] = O(a_n^4)$, we get

$$\mathbb{E}[U_n^2] \leqslant \left(\mathbb{E}\left[\left|\frac{2I_+ - n}{2n}\right|^4\right] \cdot \mathbb{E}[|M_n^+ - M_n^-|^4]\right)^{1/2} = O\left(\frac{\ln n}{n}\right)$$

which concludes the proof. $\qquad\square$

### C.7.2. CONCLUSION

We first establish the convergence of the mean. Using the triangle inequality,

$$|V_n| - |U_n| \leqslant |\bar{f}_n| \leqslant |V_n| + |U_n|.$$

Taking expectations and using $\mathbb{E}|U_n| = o(\mathbb{E}|V_n|)$ from Lemmas C.18 and C.19:

$$(1 - o(1))\,\mathbb{E}|V_n| \;\leqslant\; \mathbb{E}|\bar{f}_n| \;\leqslant\; (1 + o(1))\,\mathbb{E}|V_n|$$

which provides the announced equivalent for the mean in Lemma C.18.

For the variance, we decompose:

$$\mathrm{Var}(\bar{f}_n) = \mathrm{Var}(V_n) + \mathrm{Var}(U_n) + 2\,\mathrm{Cov}(U_n, V_n).$$

By Cauchy–Schwarz,

$$|\mathrm{Cov}(U_n, V_n)| \;\leqslant\; \sqrt{\mathrm{Var}(U_n)\,\mathrm{Var}(V_n)}.$$

In addition, from Lemma C.19, $\mathbb{E}[U_n^2] = O(\ln n/n)$, and from Lemma C.18, $\mathrm{Var}(V_n) = \Theta(1/\ln n)$. This gives $\mathrm{Var}(U_n) = o(\mathrm{Var}(V_n))$ and $|\mathrm{Cov}(U_n, V_n)| = |\mathrm{Cov}(U_n, V_n)| \leqslant \sqrt{O\left(\frac{\ln n}{n}\right)\Theta\left(\frac{1}{\ln n}\right)} = O\left(\frac{1}{\sqrt{n}}\right) = o\left(\frac{1}{\ln n}\right)$ from which $\mathrm{Var}(\bar{f}_n) \sim \mathrm{Var}(V_n)$ follows. Since $\mathbb{E}(\bar{f}_n) = \mathbb{E}(V_n) = 0$, this concludes the proof.

## C.8. Lemmas for hardmax convergence rate

In this section, we introduce some useful lemmas about extreme value theory used in the proof of Proposition 5.5.

**Definition C.20** (Asymptotic uniform integrability)**.** A family $(X_i)_{i \in I}$ of random variables is *asymptotically uniformly integrable* if:

$$\lim_{A \to \infty} \sup_{i \in I} \mathbb{E}\big[\,|X_i|\,\mathbf{1}_{\{|X_i| > A\}}\big] = 0.$$

**Lemma C.21** (Joint Convergence)**.** *Let* $Z_n = \frac{M_n^+ - a_n}{b_n}$ *and* $Y_n = \frac{-M_n^- - a_n}{b_n}$. *Joint convergence holds:*

$$\left(Z_n,\, Y_n\right) \xrightarrow{\mathcal{L}} (G_+, G_-)$$

*with* $G_+, G_-$ *independent, centered Gumbel random variables (variance* $\pi^2/6$*).*

*Proof of Lemma C.21.* A statement of the weak convergence of the marginals can be found in Theorem 1 of (Tanguy, 2015). For joint convergence, see Theorem 1.8.3 of (Nadarajah, 2000) applied to Gaussian distribution. □

**Lemma C.22** (Sum Convergence)**.**

$$\frac{2\sqrt{2 \ln n}}{\sigma} V_n = \frac{M_n^+ + M_n^-}{\sigma} \sqrt{2 \ln n} \xrightarrow{d} L$$

*where* $L := G_+ - G_-$ *follows a standard logistic law with variance* $\pi^2/3$.

*Proof.* This follows from Lemma C.21 and continuity of the mapping $(a, b) \mapsto a + b$. □

## C.9. Counter-examples for heavy-tailed distributions

This subsection shows that the sub-Gaussian assumption is mandatory in order for the hardmax attention to converge. To prove so, we exhibit a counter-example of a heavy-tailed distribution.

The key idea follows from the same decomposition as in the proof of Theorem 5.5: the empirical attention mean splits into a fluctuation term $\frac{M_n^+ - M_n^-}{\sqrt{n}}$ and a mean-shift term $\frac{M_n^+ + M_n^-}{2}$. In the Gaussian case, symmetry forces the latter to zero. For heavy-tailed distributions, this term fails to converge Consider a distribution with density

$$\mu(x) = c_k(1 + |x|)^{-k}, \qquad k \geqslant 4.$$

Taking the same notation as in Appendix B.6, for $x > 0$,

$$\mathbb{P}(M_n^+ > x) = 1 - \left(1 - c_k' x^{-k+1}\right)^n.$$

Denoting by $p_n(\alpha, \beta)$ the probability that

$$M_n^+ \in \left[(\alpha n)^{1/(k-1)}, (\beta n)^{1/(k-1)}\right],$$

we have

$$p_n(\alpha, \beta) = \left(1 - \frac{c_k' \alpha}{n}\right)^n - \left(1 - \frac{c_k' \beta}{n}\right)^n \xrightarrow[n \to +\infty]{} e^{-c_k' \alpha} - e^{-c_k' \beta}.$$

In other words, $n^{-1/(k-1)} M_n^+$ converges in law to a non-trivial explicit distribution. Since $n^{-1/(k-1)} M_n^-$ converges to the same law,

$$\frac{M_n^+ - M_n^-}{\sqrt{n}} \to 0 \qquad \text{a.s.}$$

However,

$$\frac{M_n^+ + M_n^-}{2}$$

does not converge to zero: it has strictly positive variance, and the probability that it exceeds $1$ is positive. Therefore, studying the rate of convergence makes no sense in this setting: $\mathbb{E}\,\Gamma_n$ does not converge. This shows that the sub-Gaussian assumption is not merely a technical convenience; it is necessary for the convergence of the attention output to hold.

# D. Additional results and proofs

## D.1. At most linear growth of the attention map

We start with a preliminary result that quantifies the growth of the attention map norm with the query norm. Lemma D.1 establishes that the attention map $f(x)$ cannot grow faster than linearly in the query norm $\|x\|_2$. This linear growth property is crucial for controlling the behavior of attention mechanisms and will be instrumental in establishing convergence rates.

Let us denote by $\|\Sigma\|_2 := \lambda_{\max}(\Sigma)$ the largest eigenvalue of $\Sigma$, where $\|\cdot\|_2$ denotes both the matrix norm and the Euclidean norm (clear from the context).

**Lemma D.1** (At-most linear growth of $\Gamma$). *Let $X \in \mathbb{R}^d$ be centered and sub-Gaussian with parameter matrix $\Sigma \succ 0$. For any matrices $A \in \mathbb{R}^{d \times m}$ and $V \in \mathbb{R}^{p \times d}$ and any $x, t \in \mathbb{R}^m$, define*

$$f(x) := V \nabla K(Ax), \qquad K(t) := \ln \mathbb{E}[e^{\langle t, X \rangle}].$$

*Then*

$$\left\| f(x) \right\|_2 \leqslant 2 \|V\|_2 \|\Sigma^{1/2}\|_2 \left\| \Sigma^{1/2} A \right\|_2 \|x\|_2 .$$

*Proof.* Following the proof in (Bobkov & Götze, 2025), we define the auxiliary function $P(t) = \frac{1}{2}\langle t, \Sigma t \rangle - K(t)$ which satisfies $P(t) \geqslant 0$ by definition of the sub-Gaussiannity. By Jensen's inequality, we have $K(t) \geqslant 0$ which implies $P(t) \leqslant \frac{1}{2}\langle t, \Sigma t \rangle$. Moreover, by convexity of $K(t)$, we have $P'' \leqslant \Sigma$. By Taylor expansion, this implies

$$0 \leqslant P(h + t) = P(t) + \langle \nabla P(t), h \rangle + \frac{1}{2}\langle h, \Sigma h \rangle .$$

Minimizing over $h$ the r.h.s. leads to

$$\frac{1}{2}\langle \nabla P(t), \Sigma^{-1} \nabla P(t) \rangle \leqslant P(t) .$$

Therefore, we get by the triangle inequality

$$\|\nabla K(t)\|_{\Sigma^{-1}} = \|\Sigma t - \nabla P\|_{\Sigma^{-1}} \leqslant \|t\| + \sqrt{2P(t)} \leqslant 2\|t\|_\Sigma ,$$

where we used the Mahalanobis norm $\|\cdot\|_S$ defined for a positive definite matrix S by $\|x\|_S = x^\top S x = \|S^{1/2}x\|_2^2$.

Converting this estimate for the standard Euclidean norm leads to

$$\|\nabla K(Ax)\|_2 = \left\| \Sigma^{1/2}\Sigma^{-1/2}\nabla K(Ax) \right\|_2 \leqslant \left\| \Sigma^{1/2} \right\|_{\text{op}} \left\| \Sigma^{-1/2}\nabla K(Ax) \right\|_2$$
$$\leqslant 2 \|\Sigma\|_2^{1/2} \|\Sigma^{1/2}Ax\|_2.$$

Coming back to the attention map, we have $f(x) = V \nabla K(Ax)$, which yields the claimed linear growth. $\square$

*Remark* D.2. The continuous self-attention map $f$ admits a probabilistic interpretation as an exponentially tilted expectation, or equivalently as an importance-sampling transform. More precisely, it can be rewritten as

$$f(X) \;=\; V \frac{\mathbb{E}\big[ X\, e^{\langle Ax, X \rangle} \big]}{\mathbb{E}\big[ e^{\langle Ax, X \rangle} \big]} \;=\; V \, \mathbb{E}_{\mathbb{Q}_{Ax}}[X], \qquad d\mathbb{Q}_{Ax} = \frac{e^{\langle Ax, X \rangle}}{\mathbb{E}[e^{\langle Ax, X \rangle}]}\, d\mathbb{P} .$$

where we use the *Esscher transform* for $t \in \mathbb{R}^d$, defined by the tilted measure

$$d\mathbb{Q}_t := \frac{e^{\langle t, X \rangle}}{\mathbb{E}[e^{\langle t, X \rangle}]}\, d\mathbb{P}.$$

Thus, the finite-$n$ self-attention $f_n$ is the Monte Carlo importance-sampling estimator of this tilted expectation.

Gaussian distributions saturate the sub-Gaussian bound of Lemma D.1 (see Lemma D.14): $K(t) = \frac{1}{2}t^\top \Sigma t$ exactly, so $\nabla K$ is linear and Lemma D.1 is tight up to constants. As a result, the class of Gaussian distributions is preserved layer-wise (Castin et al., 2025), which allows to follow the evolution of tokens by tracking only means and covariances. This makes Gaussians a calibrating case for both growth bounds and deviation inequalities (see for example (Wainwright, 2019), and (Vershynin, 2018)).

Note that, when $\nu$ has a compact support, the transformed distribution of the tokens stays bounded.

## D.2. Pointwise convergence of the attention map

Here we prove that the pointwise concentration of the attention map exhibits the classical $O(\frac{1}{\sqrt{n}})$ asymptotic rate for empirical processes, combined with an exponential dependence on $\|\Sigma^{1/2}Ax\|_2^2$ that captures how attention mechanisms amplify signals along high-variance directions. This pointwise analysis provides relevant insights into how individual attention outputs stabilize with increasing context length, with potential applications in attention sketching and approximation methods.

**Proposition D.3** (Pointwise convergence of the attention map for sub-Gaussian tokens)**.** *Let $X \sim \nu$ be centered and sub-Gaussian with parameter matrix $\Sigma \succ 0$. For any fixed $x \in \mathbb{R}^m$, the empirical attention computed from $n$ i.i.d. tokens satisfies, for $n$ large enough, with probability at least $1 - 2\delta$,*

$$\left\|f_n(x) - f(x)\right\|_2 \leqslant q_{\Sigma,V,d,\delta} \cdot \frac{e^{\frac{5}{2}\|\Sigma^{1/2}Ax\|_2^2}}{\sqrt{n}}, \tag{26}$$

*where $q_{\Sigma,V,d,\delta} := (2 + \sqrt{3})\|V\|_2 \|\Sigma\|_2^{1/2} \cdot q_{d,1-\delta}^{1/2}$.*

*Here $q_{d,1-\delta}$ denotes the $(1-\delta)$-quantile of the $\chi^2(d)$ distribution, which satisfies in particular $q_{d,1-\delta} \leqslant d + 2\sqrt{d\ln(1/\delta)} + 2\ln(1/\delta)$.*

The proof of this result combines classical tools of statistics with properties of sub-Gaussian distributions.

*Proof.* Let $\nu$ be as in the statement of Proposition D.3, and let $(x_1, \ldots, x_n)$ be i.i.d. samples from $\nu$. We let $A = K^\top Q$ where $K$ and $Q$ are the key and query matrices in the attention mechanism, and consider the random vector $Z_j(x) := \left(e^{\langle Ax, x_j\rangle} x_j, e^{\langle Ax, x_j\rangle}\right) \in \mathbb{R}^{d+1}$.

We introduce notation for the numerator and denominator of the attention maps $f_n(x)$ and $f(x)$:

$$f_n(x) = \frac{N_n(x)}{D_n(x)} \quad \text{where} \quad N_n(x) = \frac{1}{n}\sum_{j=1}^n e^{\langle Ax, x_j\rangle} x_j \in \mathbb{R}^d, \quad D_n(x) = \frac{1}{n}\sum_{j=1}^n e^{\langle Ax, x_j\rangle} \in \mathbb{R}$$

$$f(x) = \frac{N(x)}{D(x)} \quad \text{where} \quad N(x) = \mathbb{E}_{X\sim\nu}\left[e^{\langle Ax, X\rangle} X\right] \in \mathbb{R}^d, \quad D(x) = \mathbb{E}_{X\sim\nu}\left[e^{\langle Ax, X\rangle}\right] \in \mathbb{R}$$

where $X \sim \nu$.

**Step 1: Joint Central Limit Theorem (CLT) and Delta Method.** Under the assumptions that $\mathbb{E}\left[\|e^{\langle Ax, X\rangle} X\|_2\right] < \infty$ and $\mathbb{E}[e^{\langle Ax, X\rangle}] < \infty$, the strong law of large numbers ensures almost sure convergence

$$N_n(x) \xrightarrow{\text{a.s.}} N(x), \qquad D_n(x) \xrightarrow{\text{a.s.}} D(x) \quad \text{as } n \to \infty.$$

The random vectors $Z_j(x) = (e^{\langle Ax, x_j\rangle} x_j, e^{\langle Ax, x_j\rangle}) \in \mathbb{R}^{d+1}$ are i.i.d. with mean $\mathbb{E}[Z_1(x)] = (N(x), D(x)) \in \mathbb{R}^{d+1}$. Assuming $\mathbb{E}[\|Z_1(x)\|^2] < \infty$, the multivariate central limit theorem yields

$$\sqrt{n}\left(\begin{pmatrix} N_n(x) \\ D_n(x) \end{pmatrix} - \begin{pmatrix} N(x) \\ D(x) \end{pmatrix}\right) \xrightarrow{d} \mathcal{N}_{d+1}\left(0, \mathrm{Cov}(Z_1(x))\right)$$

where $\mathrm{Cov}(Z_1(x)) = \begin{pmatrix} \Sigma_{11}(x) & \Sigma_{12}(x) \\ \Sigma_{21}(x) & \Sigma_{22}(x) \end{pmatrix} \in \mathbb{R}^{(d+1)\times(d+1)}$, and

$$\Sigma_{11}(x) = \mathrm{Cov}(e^{\langle Ax, X\rangle} X) = \mathbb{E}[X^\top X e^{2\langle Ax, X\rangle}] - \left(\mathbb{E}[X e^{\langle Ax, X\rangle}]\right)^2 \in \mathbb{R}^{d\times d}$$

$$\Sigma_{22}(x) = \mathrm{Var}(e^{\langle Ax, X\rangle}) = \mathbb{E}[e^{2\langle Ax, X\rangle}] - \left(\mathbb{E}[e^{\langle Ax, X\rangle}]\right)^2 \in \mathbb{R}$$

$$\Sigma_{12}(x) = \Sigma_{21}(x)^\top = \mathrm{Cov}(e^{\langle Ax, X\rangle} X, e^{\langle Ax, X\rangle}) = \mathbb{E}[X e^{2tX}] - \mathbb{E}[X e^{tX}]\mathbb{E}[e^{tX}] \in \mathbb{R}^d.$$

Applying the multivariate delta method to $g : \mathbb{R}^d \times \mathbb{R}_*^+ \to \mathbb{R}^d$ defined by $g(a, b) = a/b$, we obtain,

$$\sqrt{n}\left(\frac{N_n(x)}{D_n(x)} - \frac{N(x)}{D(x)}\right) \xrightarrow{d} \mathcal{N}_d\left(0, \mathrm{Cov}(x)\right),$$

where the asymptotic covariance is denoted $\mathrm{Cov}(x)$ and is equal to

$$\mathrm{Cov}(x) = \nabla g_{(N(x),D(x))} \, \mathrm{Cov}(Z_1(x)) \nabla g_{(N(x),D(x))}^{\top} \in \mathbb{R}^{d \times d}$$

The gradient of $g$ at $(N(x), D(x))$ is

$$\nabla g_{(N(x),D(x))} = \left( \frac{1}{D(x)} I_d \quad -\frac{N(x)}{D(x)^2} \right) \in \mathbb{R}^{d \times (d+1)},$$

where $I_d$ is the $d \times d$ identity matrix. The variance $\mathrm{Cov}(x)$ expands as

$$\mathrm{Cov}(x) = \frac{1}{D(x)^2} \Sigma_{11}(x) - \frac{1}{D(x)^3} \left( N(x)\Sigma_{12}(x)^{\top} + \Sigma_{12}(x)N(x)^{\top} \right) + \frac{1}{D(x)^4} \Sigma_{22}(x) N(x) N(x)^{\top}.$$

**Step 2: Chi-Square Bound.** From the asymptotic normality, the squared Euclidean norm converges in distribution to a chi-squared random variable

$$\left\| \sqrt{n} \, \mathrm{Cov}(x)^{-1/2}(f_n(x) - f(x)) \right\|_2^2 \xrightarrow{d} \chi^2(d).$$

For any $\delta \in (0, 1)$, for $n$ large enough , we have

$$\mathbb{P} \left( \left\| \sqrt{n} \, \mathrm{Cov}(x)^{-1/2}(f_n(x) - f(x)) \right\|_2^2 \leqslant q_{d,1-\delta} \right) \geqslant 1 - \delta \tag{27}$$

where $q_{d,1-\delta} := F_{\chi^2(d)}^{-1}(1 - \delta)$ is the $(1 - \delta)$-quantile of the $\chi^2(d)$ distribution.

To derive the desired inequality for $\|f_n(x) - f(x)\|_2^2$, we use a spectral inequality.

**Step 3: Spectral Bound.** Let $\tilde{z}_n = \sqrt{n}(f_n(x) - f(x))$. Applying the Rayleigh-Ritz theorem, we get

$$\|\tilde{z}_n\|_2^2 \leqslant \lambda_{\max}(\mathrm{Cov}(x)) \| \mathrm{Cov}(x)^{-1/2} \tilde{z}_n \|_2^2.$$

This yields the event inclusion

$$\left\{ \| \mathrm{Cov}(x)^{-1/2} \tilde{z}_n \|_2^2 \leqslant q_{d,1-\delta} \right\} \subset \left\{ \|\tilde{z}_n\|_2^2 \leqslant \lambda_{\max}(\mathrm{Cov}(x)) q_{d,1-\delta} \right\}. \tag{28}$$

From (27) and (28), we obtain that for $n$ large enough,

$$\mathbb{P} \left( \|\tilde{z}_n\|_2^2 \leqslant \lambda_{\max}(\mathrm{Cov}(x)) q_{d,1-\delta} \right) \geqslant 1 - \delta.$$

Therefore:

$$\|f_n(x) - f(x)\|_2 \leqslant \frac{\lambda_{\max}(\mathrm{Cov}(x))^{1/2} q_{d,1-\delta}^{1/2}}{\sqrt{n}} \quad \text{with probability } 1 - \delta.$$

This spectral inequality allows us to convert bounds on quadratic forms back to Euclidean norm bounds, completing the bridge between the chi-squared concentration and our desired result. The quantile $q_{d,1-\delta}$ can be upper bounded using the Laurent-Massart concentration inequality for chi-squared random variables (see (Laurent & Massart, 2000, Section 4.1, Lemma 1)) stating that, for any $x > 0$,

$$\mathbb{P}(\chi^2(d) \geqslant d + 2\sqrt{dx} + 2x) \leqslant e^{-x}.$$

Setting $x = \ln(1/\delta)$ in this inequality yields

$$\mathbb{P} \left( \chi^2(d) \geqslant d + 2\sqrt{d\ln(1/\delta)} + 2\ln(1/\delta) \right) \leqslant \delta.$$

By definition of the quantile function,

$$q_{d,1-\delta} \leqslant d + 2\sqrt{d\ln(1/\delta)} + 2\ln(1/\delta).$$

This yields our refined bound, with probability at least $1 - \delta$,

$$\|f_n(x) - f(x)\|_2 \leqslant \frac{\lambda_{\max}(\operatorname{Cov}(x))^{1/2} \left(d + 2\sqrt{d\ln(1/\delta)} + 2\ln(1/\delta)\right)^{1/2}}{\sqrt{n}}.$$

**Step 4: Moment Generating Function Representation.** We express the variance components using the moment generating function (MGF) $M(t) = \mathbb{E}[e^{t^\top X}]$ and its derivatives. For a centered sub-Gaussian $X$ with parameter $\Sigma$:

$$M(t) = \mathbb{E}[e^{\langle t, X\rangle}], \quad M'(t) = \mathbb{E}[Xe^{\langle t, X\rangle}], \quad M''(t) = \mathbb{E}[XX^\top e^{\langle t, X\rangle}].$$

The variance components become

$$\Sigma_{11}(x) = M''(2Ax) - M'(Ax)M'(Ax)^\top.$$
$$\Sigma_{22}(x) = M(2Ax) - M(Ax)^2.$$
$$\Sigma_{12}(x) = M'(2Ax) - M'(Ax)M(Ax).$$

Since $D(x) = M(Ax)$ and $N(x) = M'(Ax)$, the covariance $\operatorname{Cov}(x)$ can be written as

$$\operatorname{Cov}(x) \quad = \quad \frac{M''(2Ax)}{M(Ax)^2} \quad - \quad \frac{M'(Ax)M'(2Ax)^\top + M'(2Ax)M'(Ax)^\top}{M(Ax)^3} \quad + \quad \frac{M(2Ax)}{M(Ax)^4} M'(Ax)M'(Ax)^\top.$$

Let $X$ be $\Sigma$-sub-Gaussian with $\|\Sigma\|_2^{1/2} = \sqrt{\lambda_{\max}(\Sigma)}$. For any $t \in \mathbb{R}^d$ and unit vector $u$:

$$|\langle u, M'(t)\rangle| \leqslant \|\Sigma\|_2^{1/2} e^{t^\top \Sigma t}. \tag{29}$$
$$\langle u, M''(t)u\rangle \leqslant \sqrt{3}\|\Sigma\|_2 e^{t^\top \Sigma t}. \tag{30}$$

Let us prove this result. Let $Y = \langle u, X\rangle$. By the Cauchy-Schwarz inequality,

$$|\langle u, M'(t)\rangle| = |\mathbb{E}[Ye^{\langle t, X\rangle}]| \leqslant (\mathbb{E}[Y^2])^{1/2}(\mathbb{E}[e^{2\langle t, X\rangle}])^{1/2}.$$

Since $X$ is $\Sigma$-sub-Gaussian, $Y$ is $\sigma_u$-sub-Gaussian with $\sigma_u = \sqrt{u^\top \Sigma u} \leqslant \|\Sigma\|_2^{1/2}$. Thus $\mathbb{E}[Y^2] \leqslant \|\Sigma\|_2$ and $\mathbb{E}[e^{2\langle t, X\rangle}] \leqslant e^{2t^\top \Sigma t}$.

The second inequality follows similarly using the fourth moment bound $\mathbb{E}[Y^4] \leqslant 3\|\Sigma\|_2^2$ for sub-Gaussian random variables.

**Step 5: Main spectral bound.** We will prove the following spectral bound for attention variance. For any $x \in \mathbb{R}^d$,

$$\lambda_{\max}(\operatorname{Cov}(x)) \leqslant (2 + \sqrt{3})\|\Sigma\|_2 e^{5(Ax)^\top \Sigma Ax}.$$

Let $u$ be a unit vector and $t = Ax$. Using $M(t) \geqslant 1$, and $\frac{M(2t)}{M(t)^4} \leqslant \frac{1}{M(t)^2} \leqslant 1$,

$$|u^\top \operatorname{Cov}(x)u| \leqslant |u^\top M''(2t)u| + |u^\top M'(t)|^2 + 2|u^\top M'(2t)||u^\top M'(t)|.$$

Substituting the bounds from (29) and (30), with $p(t) := t^\top \Sigma t$,

$$|u^\top \operatorname{Cov}(x)u| \leqslant \|\Sigma\|_2 \left(\sqrt{3}e^{4p(t)} + e^{2p(t)} + 2e^{5p(t)}\right).$$

Since the exponential terms are dominated by $e^{5p(t)}$, we obtain

$$|u^\top \operatorname{Cov}(x)u| \leqslant (2 + \sqrt{3})\|\Sigma\|_2 e^{5p(t)}.$$

Since this holds for all unit vectors $u$, the result follows.

**Step 6: Conclusion.** Combining our results, we obtain that, for any $x \in \mathbb{R}^d$ and $\delta \in (0, 1)$, with probability at least $1 - \delta$,

$$\|f_n(x) - f(x)\|_2 \leqslant \frac{C(\Sigma, A, V, x, \delta, d)}{\sqrt{n}},$$

where

$$C(\Sigma, A, V, x, \delta, d) = (2 + \sqrt{3})\|V\|_2\|\Sigma^{1/2}\|_2 \, e^{\frac{5}{2}(Ax)^\top \Sigma Ax}\left(d + 2\sqrt{d\ln(1/\delta)} + 2\ln(1/\delta)\right)^{1/2}.$$

This concludes the proof. □

In the particular case of Gaussians, a specific estimation can be found in Appendix D.8.1.

## D.3. Mean convergence rate

We highlight the particular case of the mean of the distribution, which is a corollary of the previous result (with $f = \mathrm{id}$).

**Corollary D.4** (Mean convergence rate for sub-Gaussian tokens). *Let $X$ be centered and sub-Gaussian with parameter matrix $\Sigma \succ 0$ and define $H$ as in Theorem 5.1. Let us denote by $\mathbb{E}$ the expectation w.r.t. $\nu$ and $\mathbb{E}_n$ the expectation w.r.t. the empirical measure. For $n$ i.i.d. tokens, with $n \geqslant 4\left(\frac{1}{\delta} - 1\right) n^{1/8}$, with probability at least $1 - \delta$,*

$$\left\| \mathbb{E}_n[f_n(x)] - \mathbb{E}[f(x)] \right\|_2 \; = \; P(\sqrt{\ln n}) \cdot O\!\left(n^{-\frac{1}{2(1+32\cdot H^2)}}\right).$$

*where $P$ is a polynomial function.*

The same result holds for Gaussian distribution, with a sharper rate (see Section D.8.6)

**Proposition D.5** (Mean convergence rate for Gaussian tokens). *For Gaussian tokens using the same notation we have,*

$$\left\| \mathbb{E}_n[f_n(x)] - \mathbb{E}[f(x)] \right\| = O\!\left( (\ln n)^{\frac{d+1}{2}} \, n^{-\frac{1}{2(1+H^2)}} \right).$$

## D.4. Mean squared error (MSE) convergence rate

**Corollary D.6** (Mean-squared convergence rate for Lipschitz observables under sub-Gaussian tokens). *Let $X$ be centered and sub-Gaussian with parameter matrix $\Sigma \succ 0$, and define $H$ as in Definition 5.1. Let $h$ be an $L_0$-Lipschitz function, squared integrable with respect to $\nu$. Let $\mathbb{E}$ denote expectation with respect to $\nu$. For $n$ i.i.d. tokens, assuming $n \geqslant 4\left(\frac{1}{\delta} - 1\right)n^{1/8}$, with probability at least $1 - \delta$,*

$$\mathbb{E}\!\left[ \left\| f_n(X) - f(X) \right\|_2^2 \right] \leqslant \frac{P(\sqrt{\ln n}, L_0)}{n^\beta},$$

*where $P$ is a polynomial function, and*

$$\beta := \frac{1}{2(1 + 32H^2)}.$$

The proof follows from the same truncation and bounded-region versus tail decomposition argument used to establish the moment bound.

## D.5. Deep composition of layers

Let

$$F := f^L \circ \cdots \circ f^1 \qquad \text{and} \qquad F_n := f_n^L \circ \cdots \circ f_n^1$$

be respectively a stack of $L$ attention layers and its sparse-attention approximation. For each $\ell \in \{1, \ldots, L\}$, assume that $f_n^\ell$ is constructed from $n$ i.i.d. subsampled tokens drawn from the layer-$\ell$ token distribution $\mu^\ell$.

Provided that the subsampling remains i.i.d. at each layer, local sketching errors accumulate through the network, with amplification governed by the Lipschitz constants of downstream attention layers.

**Corollary D.7.** *Let $F$ and $F_n$ be defined as above, and assume that all token distributions remain supported in a ball of radius $R$ after each layer $l \in [1, L]$. Then,*

$$\|F - F_n\|_{L^2(\mu^1)} = O\!\left( \sum_{\ell=1}^{L} \frac{R^{2(L-\ell)}}{n^{\beta_\ell}} \right).$$

*Proof.* Consider a stack of $L$ attention layers

$$F = f^L \circ \cdots \circ f^1$$

and its sparse attention approximation

$$F_n = f_n^L \circ \cdots \circ f_n^1,$$

where each $f_n^\ell$ is constructed from $n$ i.i.d. subsampled tokens drawn from the layer-$\ell$ token distribution $\mu^\ell$. Applying the single-layer convergence bound given by 5.3 at each depth yields

$$\|f^\ell - f_n^\ell\|_{L^2(\mu^\ell)} = O(n^{-\beta_\ell}),$$

where $0 < \beta_\ell < 1/2$ is the convergence exponent associated with layer $\ell$. If all token distributions remain supported in a ball of radius $R$, each attention map is $O(R^2)$-Lipschitz, and a composition argument gives

$$\|F - F_n\|_{L^2(\mu^1)} = O\left(\sum_{\ell=1}^{L} \frac{R^{2(L-\ell)}}{n^{\beta_\ell}}\right).$$

This concludes the proof. $\qquad\square$

### D.6. Covariance convergence rate for sub-Gaussian tokens

A similar convergence rate holds for the covariance matrix. Let

$$\mathrm{Cov}_n(f(\hat{X})) := \frac{1}{n}\sum_{i=1}^{n}(f_n(x_i) - \bar{f}_n)(f_n(x_i) - \bar{f}_n)^\top, \quad \bar{f}_n := \frac{1}{n}\sum_{i=1}^{n} f_n(x_i),$$

and

$$\mathrm{Cov}(f(X)) := \mathbb{E}\big[f(X)f(X)^\top\big] - \mathbb{E}[f(X)]\mathbb{E}[f(X)^\top].$$

Our main result regarding the convergence rate of the covariance is the following.

**Proposition D.8** (Covariance convergence rate for sub-Gaussian tokens)**.** *With the notation* $\mathrm{Cov}_n(f_n(\hat{X}))$ *and* $\mathrm{Cov}(f(X))$ *defined above, for $n$ i.i.d. tokens with $n \geqslant 4\left(\frac{1}{\delta} - 1\right)n^{1/8}$, with probability at least $1 - \delta$,*

$$\big\|\mathrm{Cov}_n(f_n(\hat{X})) - \mathrm{Cov}(f(X))\big\|_2 \leqslant P(\sqrt{\ln n})\, O\big(n^{-\frac{1}{2(1+16\cdot H^2)}}\big). \tag{31}$$

The proof follows the same steps as the one of Theorem 5.3.

*Proof.* To alleviate the notation, set $g_i := \Gamma_\nu(x_i)$ and $\hat{g}_i := f_n(x_i)$ for all $i \in \{1, \ldots, n\}$, as well as the empirical means $\bar{g}_n = \frac{1}{n}\sum_{i=1}^{n} g_i$ and $\bar{\hat{g}}_n = \frac{1}{n}\sum_{i=1}^{n} \hat{g}_i$. We also denote $g := f(x)$ and $\hat{g} := f_n(x)$. Now, decompose

$$\begin{aligned}
&\|\mathrm{Cov}_n(\hat{g}) - \mathrm{Cov}(g)\|_2 \\
&\leqslant \|\mathrm{Cov}_n(\hat{g}) - \mathrm{Cov}_n(g)\|_2 + \|\mathrm{Cov}_n(g) - \mathrm{Cov}(g)\|_2 \\
&= \left\|\frac{1}{n}\sum_{i=1}^{n}\big(\hat{g}_i\hat{g}_i^\top - g_i g_i^\top\big) - \big(\bar{\hat{g}}_n\bar{\hat{g}}_n^\top - \bar{g}_n\bar{g}_n^\top\big)\right\|_2 + \|\mathrm{Cov}_n(g) - \mathrm{Cov}(g)\|_2 \\
&\leqslant \mathcal{I} + \mathcal{J},
\end{aligned}$$

where

$$\mathcal{I} = \left\|\frac{1}{n}\sum_{i=1}^{n}\big(\hat{g}_i\hat{g}_i^\top - g_i g_i^\top\big) - \big(\bar{\hat{g}}_n\bar{\hat{g}}_n^\top - \bar{g}_n\bar{g}_n^\top\big)\right\|_2,$$
$$\mathcal{J} = \|\mathrm{Cov}_n(g) - \mathrm{Cov}(g)\|_2.$$

**Step 1: Analysis of the second term $\mathcal{J}$.** Recall from Lemma D.1 that,

$$\|f(x)\|_2 \leqslant \alpha\|X\|_2, \qquad \alpha := 2\|V\|_2\|\Sigma^{1/2}\|_2\|\Sigma^{1/2}A\|_2. \tag{32}$$

As $X$ follows a sub-Gaussian distribution with scalar parameter $\sigma = \|\Sigma^{1/2}\|_2$, $\|X\|_2$ is sub-Gaussian with parameter $\sigma\sqrt{d}$ (see Lemma 1 in (Jin et al., 2019)). Using the characterization of sub-Gaussianity via the tail bound $\mathbb{P}(\|X\|_2 \geqslant t) \leqslant 2e^{-t^2/(2\sigma^2)}$, we deduce from Lemma D.1 that $\|f(x)\|_2$ is sub-Gaussian. Moreover, since for every $u \in \mathbb{S}^{d-1}$, $|\langle u, f(x)\rangle| \leqslant \|f(x)\|_2 \leqslant \alpha\|X\|_2$, it follows that $f(x)$ is a sub-Gaussian random vector.

From Proposition 2.1 in (Vershynin, 2012b) applied to the sub-Gaussian vector $f(x)$,

$$\mathcal{J} \lesssim_{\sigma,\delta} \sqrt{\frac{d}{n}}.$$

**Step 2: Analysis of the first term $\mathcal{I}$.** Now bound $\mathcal{I}$,

$$
\begin{aligned}
\mathcal{I} &\leqslant \left\| \frac{1}{n}\sum_{i=1}^{n}\left(\hat{g}_i\hat{g}_i^\top - g_i g_i^\top\right) \right\|_2 + \left\| \hat{\bar{g}}_n\hat{\bar{g}}_n^\top - \bar{g}_n\bar{g}_n^\top \right\|_2 \\
&\leqslant \left\| \frac{1}{n}\sum_{i=1}^{n}\left(\hat{g}_i\hat{g}_i^\top - g_i\hat{g}_i^\top + g_i\hat{g}_i^\top - g_i g_i^\top\right) \right\|_2 + \left\| \hat{\bar{g}}_n\hat{\bar{g}}_n^\top - \hat{\bar{g}}_n\bar{g}_n^\top + \hat{\bar{g}}_n\bar{g}_n^\top - \bar{g}_n\bar{g}_n^\top \right\| \\
&\leqslant \frac{1}{n}\sum_{i=1}^{n}[\|\hat{g}_i - g_i\|_2(\|\hat{g}_i\|_2 + \|g_i\|_2)] + (\|\hat{\bar{g}}_n\|_2 + \|\bar{g}_n\|_2)\|\hat{\bar{g}}_n - \bar{g}_n\|_2 \\
&\leqslant \mathcal{I}_1 + \mathcal{I}_2.
\end{aligned}
$$

where

$$
\begin{aligned}
\mathcal{I}_1 &= \frac{1}{n}\sum_{i=1}^{n}[\|\hat{g}_i - g_i\|_2(\|\hat{g}_i\|_2 + \|g_i\|_2)] \\
\mathcal{I}_2 &= (\|\hat{\bar{g}}_n\|_2 + \|\bar{g}_n\|_2)\|\hat{\bar{g}}_n - \bar{g}_n\|_2.
\end{aligned}
$$

For $\mathcal{I}_1$,

$$
\begin{aligned}
\mathcal{I}_1 &\leqslant \frac{1}{n}\sum_{\|Bx_i\|_2 \leqslant R}[\|\hat{g}_i - g_i\|_2(\|\hat{g}_i\|_2 + \|g_i\|_2)] + \frac{1}{n}\sum_{\|Bx_i\|_2 > R}(\|\hat{g}_i\|_2 + \|g_i\|_2)^2 \\
&\leqslant \mathcal{I}_{1,\leqslant R} + \mathcal{I}_{1,>R}
\end{aligned}
$$

where

$$
\begin{aligned}
\mathcal{I}_{1,\leqslant R} &= \frac{1}{n}\sum_{\|Bx_i\|_2 \leqslant R}[\|\hat{g}_i - g_i\|_2(\|\hat{g}_i\|_2 + \|g_i\|_2)], \\
\mathcal{I}_{1,>R} &= \frac{1}{n}\sum_{\|Bx_i\|_2 > R}(\|\hat{g}_i\|_2 + \|g_i\|_2)^2.
\end{aligned}
$$

For $\|Bx_i\|_2 \leqslant R$, using the notation $N_R := \sum_{i=1}^{n}\mathbf{1}_{\{\|x_i\|_2 > R\}}$, and $M_n := \max_{1\leqslant i\leqslant n}\|x_i\|_2$, we have $\|\hat{g}_i\|_2 \leqslant M_n \leqslant R$, and $\|g_i\|_2 \leqslant \alpha M_n \leqslant \alpha R$, where $\alpha$ is defined in (32). Also, from Proposition D.3, $\|\hat{g}_i - g_i\|_2 \leqslant \frac{C_1}{\sqrt{n}}$. Hence,

$$
\mathcal{I}_{1,\leqslant R} \leqslant (1+\alpha)R\frac{C_1}{\sqrt{n}}.
$$

For $\|Bx_i\|_2 > R$,

$$
\mathcal{I}_{1,>R} \leqslant \frac{N_R}{n}M_n^2(1+\alpha)^2 \leqslant \left( e^{-c(\frac{R}{\sigma} - C\sqrt{d})^2} + \sqrt{\frac{1}{2n}\ln\!\left(\frac{4}{\delta}\right)} \right)\cdot c^2\sigma^2\left(\sqrt{d} + \sqrt{\ln\frac{2n}{\delta}}\right)^2(1+\alpha)^2,
$$

where we used the result of Proposition C.14.

For $\mathcal{I}_2$,

$$\mathcal{I}_2 = \left(\|\hat{\bar{g}}_n\|_2 + \|\bar{g}_n\|_2\right)\|\hat{\bar{g}}_n - \bar{g}_n\|_2$$

$$\leqslant \left(\frac{1}{n}\sum_{i=1}^{n}\|\hat{g}_i\|_2 + \frac{1}{n}\sum_{i=1}^{n}\|g_i\|_2\right)\frac{1}{n}\sum_{i=1}^{n}\|\hat{g}_i - g_i\|_2$$

$$\leqslant \left(\frac{1}{n}\sum_{\|x_i\|_2\leqslant R}\|\hat{g}_i\|_2 + \frac{1}{n}\sum_{\|x_i\|_2\leqslant R}\|g_i\|_2\right)\frac{1}{n}\sum_{\|x_i\|_2\leqslant R}\|\hat{g}_i - g_i\|_2$$

$$+ \left(\frac{1}{n}\sum_{\|x_i\|_2> R}\|\hat{g}_i\|_2 + \frac{1}{n}\sum_{\|x_i\|_2> R}\|g_i\|_2\right)\frac{1}{n}\sum_{\|x_i\|_2\leqslant R}\|\hat{g}_i - g_i\|_2$$

$$+ \left(\frac{1}{n}\sum_{\|x_i\|_2\leqslant R}\|\hat{g}_i\|_2 + \frac{1}{n}\sum_{\|x_i\|_2\leqslant R}\|g_i\|_2\right)\left(\frac{1}{n}\sum_{\|x_i\|_2> R}\|\hat{g}_i\|_2 + \frac{1}{n}\sum_{\|x_i\|_2> R}\|g_i\|_2\right)$$

$$+ \left(\frac{1}{n}\sum_{\|x_i\|_2> R}\|\hat{g}_i\|_2 + \frac{1}{n}\sum_{\|x_i\|_2> R}\|g_i\|_2\right)^2$$

$$\leqslant \mathcal{I}_{2,\leqslant R,\leqslant R} + \mathcal{I}_{2,> R,\leqslant R} + \mathcal{I}_{2,\leqslant R,> R} + \mathcal{I}_{2,> R,> R}.$$

where

$$\mathcal{I}_{2,\leqslant R,\leqslant R} = \left(\frac{1}{n}\sum_{\|x_i\|_2\leqslant R}\|\hat{g}_i\|_2 + \frac{1}{n}\sum_{\|x_i\|_2\leqslant R}\|g_i\|_2\right)\frac{1}{n}\sum_{\|x_i\|_2\leqslant R}\|\hat{g}_i - g_i\|_2$$

$$\mathcal{I}_{2,> R,\leqslant R} = \left(\frac{1}{n}\sum_{\|x_i\|_2> R}\|\hat{g}_i\|_2 + \frac{1}{n}\sum_{\|x_i\|_2> R}\|g_i\|_2\right)\frac{1}{n}\sum_{\|x_i\|_2\leqslant R}\|\hat{g}_i - g_i\|_2$$

$$\mathcal{I}_{2,\leqslant R,> R} = \left(\frac{1}{n}\sum_{\|x_i\|_2\leqslant R}\|\hat{g}_i\|_2 + \frac{1}{n}\sum_{\|x_i\|_2\leqslant R}\|g_i\|_2\right)\left(\frac{1}{n}\sum_{\|x_i\|_2> R}\|\hat{g}_i\|_2 + \frac{1}{n}\sum_{\|x_i\|_2> R}\|g_i\|_2\right)$$

$$\mathcal{I}_{2,> R,> R} = \left(\frac{1}{n}\sum_{\|x_i\|_2> R}\|\hat{g}_i\|_2 + \frac{1}{n}\sum_{\|x_i\|_2> R}\|g_i\|_2\right)^2.$$

The first term $\mathcal{I}_{2,\leqslant R,\leqslant R}$ is bounded by

$$\mathcal{I}_{2,\leqslant R,\leqslant R} \leqslant (1+\alpha)R\frac{C_1}{\sqrt{n}} + \left(\frac{N_R}{n}M_n(1+\alpha)\right)^2.$$

The crossed terms $\mathcal{I}_{2,> R,\leqslant R}$ and $\mathcal{I}_{2,\leqslant R,> R}$ are bounded by,

$$\mathcal{I}_{2,> R,\leqslant R} \leqslant \frac{N_R}{n}M_n(1+\alpha)\cdot\frac{C_1}{\sqrt{n}}.$$

$$\mathcal{I}_{2,\leqslant R,> R} \leqslant (1+\alpha)^2 M_n\frac{N_R}{n}\cdot\frac{C_1}{\sqrt{n}}.$$

And the last term is again bounded by

$$\mathcal{I}_{2,> R,> R} \leqslant \frac{N_R}{n}M_n^2(1+\alpha)^2.$$

Recall that $\frac{N_R}{n}M_n \leqslant \left(e^{-c\left(\frac{R}{\sigma}-C\sqrt{d}\right)^2} + \sqrt{\frac{1}{2n}\ln\left(\frac{4}{\delta}\right)}\right)\cdot c\sigma\left(\sqrt{d} + \sqrt{\ln\frac{2n}{\delta}}\right).$

Therefore, as in Theorem 5.3, optimizing over $R$ yields the optimal radius $R^\star$,

$$R^\star \approx \sqrt{\frac{\ln(n)}{16\,\|\Sigma^{1/2}AB^{-1}\|^2 + \frac{4c}{\|\Sigma B^\top\|_2}}}.$$

which again we optimize in $B$ by choosing $B := \Sigma^{-1/2}$. Finally, $\mathcal{I} + \mathcal{J} = P\left(\sqrt{\ln(n)}\right)\cdot O\left(n^{-\frac{1}{2(1+16\cdot H^2)}}\right)$ where $H := \|\Sigma^{1/2}A\Sigma^{1/2}\|_2$, is the horizon parameter and $P$ is a polynomial function of $\ln n$, which concludes the proof. $\qquad\square$

**Proposition D.9** (Uniform convergence of the attention map for compactly supported tokens). *Let $X \sim \nu$ be centered, with $\mathrm{Supp}(\nu) \subset B_{R_0}$. For any $R > 0$, $\delta > 0$, there exists a constant $C > 0$, such that for $n \geqslant n_{min}(\delta, A, R, R_0) := 4e^{2RR_0\|A\|_2} \left(\frac{1}{\delta} - 1\right)$, with probability at least $1 - \delta$,*

$$\sup_{x \in B_R} \left\|\Gamma_{\hat{\nu}_n}(x) - \Gamma_\nu(x)\right\|_2 \leqslant \frac{q_{V,A,R_0,d,\delta}}{\sqrt{n}} \cdot e^{2RR_0\|A\|_2}, \tag{33}$$

*where $q_{V,A,R_0,d,\delta} := \|V\|_2 \cdot \frac{c\sqrt{d}\,R_0(2\,R_0^2\|A\|_2 + R_0\,\|A\|_2 + 2)}{\delta}$.*

The proof of Proposition D.9 is the same as the proof of Theorem 4.2 in Appendix C.2, except for the step 2 of part 1, where bounds on the Lipschitz constant and envelope function are derived differently. More precisely, Lemma C.7, Lemma C.10 and Lemma C.8 needs to be replaced by Lemma D.10, Lemma D.11, and Lemma D.12 respectively.

**Lemma D.10** (Bound on Lipschitz constant of numerator for compactly supported $\nu$). *Let $f \in \mathcal{F}$, and $L$ defined in Lemma C.7, for $Y \sim \nu$, with support $\nu \subset B_{R_0}$, and $x \in B_R$,*

$$\|L\|_{L^2(P)} \leqslant R_0^2 \, e^{RR_0\|A\|_2}.$$

*Proof.* Consider $f(t, y) := y\,e^{\langle t, y\rangle}$. Then, with $t = Ax$, $\nabla_t f(t, y) = A^\top \nabla_x f(x, y)$. The Jacobian of $f(\cdot, y)$ at $t$ is

$$\nabla_t f(t, y) = y\,y^\top\,e^{\langle t, y\rangle} \in \mathbb{R}^{d \times d},$$

$$\begin{aligned}
\|\nabla_t f(t, y)\|_{\mathrm{L}^2(\nu)} &:= \left(\mathbb{E}_{Y \sim \nu}\|\nabla_t f(t, Y)\|_2^2\right)^{1/2} \\
&= \left(\mathbb{E}_{Y \sim \nu}\left[\|Y\|_2^4\,e^{2\langle t, y\rangle}\right]\right)^{1/2} \\
&\leqslant R_0^2\left(\mathbb{E}_{Y \sim \nu}\left[e^{2\langle Ax, Y\rangle}\right]\right)^{1/2} \\
&\leqslant R_0^2\,e^{RR_0\|A\|_2}
\end{aligned}$$

where we used Cauchy-Schwarz to bound $\langle t, z\rangle \leqslant \|Ax\|_2\|Y\|_2 \leqslant RR_0\|A\|_2$. $\qquad\square$

**Lemma D.11** (Bound on Lipschitz constant of denominator for compactly supported $\nu$). *Let $f \in \mathcal{F}'$, and $L$ defined in Lemma C.10, and let $Y \sim \nu$, with support $\nu \subset B_{R_0}$, and $x \in B_R$. Then,*

$$\|L\|_{L^2(P)} \leqslant \|A\|_2\,R_0\,e^{RR_0\|A\|_2}.$$

*Proof.* Consider $f(t, y) := e^{\langle t, y\rangle}$. Then, with $t = Ax$, $\nabla_x f(x, y) = A^\top \nabla_t f(Ax, y)$.
The Jacobian of $f(\cdot, y)$ at $t$ is

$$\nabla_t f(t, y) = y\,e^{\langle t, y\rangle} \in \mathbb{R}^{d \times d},$$

$$\begin{aligned}
\|\nabla_t f(t, y)\|_{\mathrm{L}^2(\nu)} &:= \left(\mathbb{E}_{Y \sim \nu}\|\nabla_t f(t, Y)\|_2^2\right)^{1/2} \\
&= \left(\mathbb{E}_{Y \sim \nu}\left[\|Y\|_2^2\,e^{2\langle t, y\rangle}\right]\right)^{1/2} \\
&\leqslant R_0\,e^{RR_0\|A\|_2}.
\end{aligned}$$

$\qquad\square$

**Lemma D.12** (Bound on envelope function for compactly supported $\nu$). *For $Y \sim \nu$, with support $\nu \subset B_{R_0}$, and $x \in B_R$,*

$$\left(\mathbb{E}[F(Y)^2]\right)^{1/2} \leqslant R_0\,e^{RR_0\|A\|_2}.$$

*Proof.*

$$\mathbb{E}[F(Y)^2] = \mathbb{E}\left[\|Y\|_2^2\,e^{2R\|A^\top Y\|_2}\right] \leqslant R_0^2\,e^{2RR_0\|A\|_2}.$$

$\qquad\square$

To apply those results to prove Proposition D.9, recall equation (10) for the numerator of attention in the proof of Theorem 4.2 in Appendix C.2,

$$\mathbb{E}[\|\mathbb{P}_n - \mathbb{P}\|_{\mathcal{F}}] \leqslant \frac{c\sqrt{d}}{\sqrt{n}} \Big( R_0 \, \mathbb{E}[\|L\|_{L^2(\mathbb{P}_n)}] + \mathbb{E}[\|F\|_{L^2(\mathbb{P}_n)}] \Big).$$

Applying Lemma D.10 and Lemma D.12 yields,

$$\mathbb{E}[\|\mathbb{P}_n - \mathbb{P}\|_{\mathcal{F}}] \leqslant \frac{c\sqrt{d}}{\sqrt{n}} \Big( \|A\|_2 \, R_0^3 \, e^{RR_0\|A\|_2} + R_0 \, e^{RR_0\|A\|_2} \Big)$$

$$\leqslant \frac{c\sqrt{d}}{\sqrt{n}} R_0 \, (\|A\|_2 \, R_0^2 + 1) \, e^{RR_0\|A\|_2}.$$

To control the denominator of attention, we apply Lemma D.11 to equation (12),

$$E[\|\mathbb{P}_n - \mathbb{P}\|_{\mathcal{F}'}] \leqslant \frac{c}{\delta\sqrt{n}} \|A\|_2 \, R_0 \, e^{RR_0\|A\|_2}. \tag{34}$$

Combining those results, the conclusion of part 3 in Appendix C.2 becomes,

$$\|N\|_2 = \|\mathbb{E}[Y e^{\langle Ax, Y\rangle}]\|_2 \leqslant R_0 \, e^{RR_0\|A\|_2},$$

$$\|\frac{N_n}{D_n} - \frac{N}{D}\|_2 \leqslant \|V\|_2 \cdot \frac{c\sqrt{d} \, R_0 (2\,R_0^2\|A\|_2 + R_0\,\|A\|_2 + 2)}{\delta} \cdot \frac{1}{\sqrt{n}} \cdot e^{2RR_0\|A\|_2}.$$

## D.7. Compactly supported tokens

From Proposition D.9, we deduce the following estimation in the compact support setting.

**Proposition D.13** (Lipschitz functional convergence rate for compactly supported tokens). *Let $X \sim \nu$ centered with* $\text{Supp}(\nu) \subset B_{R_0}$, *and let $h : \mathcal{L}^2(\nu) \to \mathcal{L}^2(\nu)$ be a Lipschitz function with Lipschitz constant $L_0$. Recall the definition of the empirical distribution $\nu_n = \frac{1}{n}\sum_{i=1}^n \delta_{Y_i}$. Let us denote by $\mathbb{E}$ the expectation w.r.t. $\nu$, and $\mathbb{E}_n$ the expectation w.r.t. the empirical measure. For $n$ i.i.d. tokens, with probability at least $1 - \delta$,*

$$\Big\|\mathbb{E}_n[h \circ f_n(\hat{X})] - \mathbb{E}[h \circ f(X)]\Big\|_2 \leqslant \frac{1}{\sqrt{n}} \cdot L_0 \cdot \Big( q_{V,A,R_0,d,\delta} \, e^{2 \cdot R_0^2\|A\|_2} + q'_{V,R_0,d,\delta} \Big)$$

*where $q_{V,A,R_0,d,\delta}$ is defined in Proposition D.9, and $q'_{V,R_0,d,\delta} := \|V\|_2 \, q_{d,1-\delta}^{1/2} R_0^2$.*

## D.8. Gaussian case

This section provides additional results for the Gaussian token distribution, including pointwise convergence rates, one-dimensional analysis, and detailed proofs of auxiliary lemmas used in the main text.

**Lemma D.14** (Exact linearity under Gaussian inputs (Castin and al. 2025)). *If $X \sim \mathcal{N}(0, \Sigma)$, then for all $x$, the log-MGF is quadratic and the population attention map is* exactly linear:

$$f(x) = V \, \nabla K(Ax) = V \Sigma Ax. \qquad K(t) := \ln \mathbb{E}[e^{\langle t, X\rangle}] = \tfrac{1}{2} t^\top \Sigma t.$$

*Hence the Gaussian family is preserved by attention, and first/second moments determine the dynamics.*

### D.8.1. POINTWISE CONVERGENCE

We begin with concentration bounds for fixed query vectors under Gaussian token distribution.

**Lemma D.15** (High-probability concentration (Gaussian constants)). *For any fixed query $x \in \mathbb{R}^d$ and $\delta \in (0, 1)$, with probability at least $1 - \delta$,*

$$\big\|f_n(x) - f(x)\big\|_2 \leqslant \frac{C(\Sigma, A, x, \delta, d)}{\sqrt{n}},$$

*where*

$$C(\Sigma, A, x, \delta, d) = \|V\|_2 \cdot q_{A,x,d,\delta}^{\mathcal{N}} \cdot e^{\frac{1}{2}\|\Sigma^{1/2}Ax\|_2^2}.$$

$q_{A,x,d,\delta}^{\mathcal{N}} := \sqrt{\|\Sigma\|_2 + \|\Sigma Ax\|_2^2} \; q_{d,1-\delta}^{1/2}.$

### D.8.2. DIMENSION 1

In the one-dimensional case, sharper convergence rates can be obtained with explicit constants.

**Lemma D.16** (Concentration in $d = 1$). *With probability at least $1 - \delta$,*

$$\left\|f_n(x) - f(x)\right\|_2 \leqslant \frac{C(\sigma, a, x, \delta)}{\sqrt{n}}, \qquad C(\sigma, a, x, \delta) = |V| \, q_{1-\delta/2} \, \sigma \, \sqrt{1 + \sigma^2 a_x^2} \, \exp\!\left(\tfrac{1}{2}\sigma^2 a_x^2\right).$$

**Proposition D.17** (Mean convergence rate in $d = 1$). *Let $H = (a\sigma) \cdot \sigma$. Then,*

$$\left\|\mathbb{E}_n[f_n(x)] - \mathbb{E}_{X \sim \hat{\nu}_n}[f(x)]\right\| = \Theta\!\left(\ln n \, n^{-\frac{1}{2(1+H^2)}}\right).$$

### D.8.3. UNIFORM CONVERGENCE

**Proposition D.18** (Uniform convergence of the attention map for Gaussian tokens). *Let $X \sim \nu$ be centered and sub-Gaussian with parameter matrix $\Sigma \succ 0$. For any $R > 0$, $\delta > 0$, there exists a constant $C > 0$, such that for $n \geqslant n_{min}(\delta, \Sigma, A, R) := 4e^{2R^2\|\Sigma^{1/2}A\|_2^2}\left(\frac{1}{\delta} - 1\right)$, with probability at least $1 - \delta$,*

$$\sup_{x \in B_R} \left\|f_n(x) - f(x)\right\|_2 \leqslant q^{\mathcal{N}}_{(\Sigma, A, V, R, d, \delta)} \cdot \frac{e^{R^2\|\Sigma^{1/2}A\|_2^2}}{\sqrt{n}}, \tag{35}$$

*where $q^{\mathcal{N}}_{(\Sigma, A, V, R, d, \delta)}$ is defined in Appendix D.8.4.*

### D.8.4. PROOF OF PROPOSITION D.18

The proof of Proposition D.18 is the same as the one of Theorem 4.2 in Appendix C.2, except for step 2 of part 1, where bounds on the Lipschitz constant and envelope function are tighter for Gaussian distributions. More precisely, Lemma C.7 , Lemma C.10 and Lemma C.8 needs to be replaced by Lemma D.19 , Lemma D.20 and Lemma D.21 respectively.

**Lemma D.19** (Bound on Lipschitz constant for numerator for Gaussian distribution). *Let $f \in \mathcal{F}$, and $L$ defined in Lemma B.7, for $Y \sim \mathcal{N}(0, \Sigma)$, and $x \in B_R$,*

$$\|L\|_{L^2(P)} \leqslant c(\Sigma, A, R) \exp\!\left(R^2\|\Sigma^{1/2}A\|_2^2\right),$$

*where $c(\Sigma, A, R)$ is a polynomial function in $\|A\|_2$, $\|\Sigma\|_2$, $R$, and $\operatorname{tr}(\Sigma)$.*

*Proof.* Consider $f(t, y) := y \, e^{\langle t, y \rangle}$ - we will then apply the result to $t = Ax$ using $\nabla_t f(t, y) = A^\top \nabla_x f_x(y)$.
The Jacobian of $f(\cdot, y)$ at $t$ is

$$\nabla_t f(t, y) \ = y \, y^\top \, e^{\langle t, y \rangle} \ \in \mathbb{R}^{d \times d},$$

Let $\|\nabla_t f(t, y)\|_{L^2(\nu)} := \left(\mathbb{E}_{Y \sim \nu}\|\nabla_t f(t, Y)\|_2^2\right)^{1/2} = \left(\mathbb{E}_{Y \sim \nu}\left[\|Y\|_2^4 \, e^{2\langle t, y \rangle}\right]\right)^{1/2}$, where we used the fact that $Y Y^\top$ is a rank one matrix, hence $\|Y Y^\top\| = \|Y\|_2^2$.

Notice that, for any measurable function $g$,

$$\begin{aligned}
\mathbb{E}[g(Y)e^{\langle t, y \rangle}] &= \int g(y)e^{\langle t, y \rangle}\phi_\Sigma(y)\,dy = \int g(y)\exp\!\left(\langle t, y \rangle - \frac{1}{2}y^\top\Sigma^{-1}y\right)\frac{dy}{(2\pi)^{d/2}\det(\Sigma)^{1/2}} \\
&= \int g(y)\exp\!\left(-\frac{1}{2}(y - \Sigma t)^\top\Sigma^{-1}(y - \Sigma t) + \frac{1}{2}t^\top\Sigma t\right)\frac{dy}{(2\pi)^{d/2}\det(\Sigma)^{1/2}} \\
&= e^{\frac{1}{2}t^\top\Sigma t}\int g(y)\phi_\Sigma(y - \Sigma t)\,dy = e^{\frac{1}{2}t^\top\Sigma t}\mathbb{E}\left[g(Y + \Sigma t)\right].
\end{aligned}$$

Using this property, we can write:

$$\|\nabla_t f(t, y)\|_{L^2(\nu)} = e^{t^\top\Sigma t}\left(\mathbb{E}_{Y \sim \nu}\left[\|Y + \Sigma t\|_2^4\right]\right)^{1/2},$$

Note that $t^\top \Sigma t = x^\top A^\top \Sigma A\, x \leqslant \|\Sigma^{1/2} A\|_2^2 \|x\|_2^2$, and take the supremum over $\|x\|_2 \leqslant R$:

$$\|L\|_{L^2(P)} := \sup_{\|x\|_2 \leqslant R} \|\nabla_x f_x(y)\|_{L^2(\nu)} \leqslant c(\Sigma, A)\, \exp\big(R^2 \|\Sigma^{1/2} A\|_2^2\big),$$

where $c(\Sigma, A)$ is a polynomial function in $\|A\|_2$, $\|\Sigma\|_2$, and $\mathrm{tr}(\Sigma)$. $\qquad\square$

**Lemma D.20** (Bound on Lipschitz constant for denominator for Gaussian distribution). *Let $f \in \mathcal{F}$, and $L$ defined in Lemma B.7, for $Y \sim \mathcal{N}(0, \Sigma)$, and $x \in B_R$*

$$\|L\|_{L^2(P)} \leqslant c'(\Sigma, A, R) \exp\big(R^2 \|\Sigma^{1/2} A\|_2^2\big),$$

*where $c'(\Sigma, A, R)$ is a polynomial function in $\|A\|_2$, $\|\Sigma\|_2$, $R$, and $\mathrm{tr}(\Sigma)$.*

*Proof.* Consider $f(t, y) := e^{\langle t, y\rangle}$ - we will then apply the result to $t = Ax$ using $\nabla_x f(x, y) = A^\top \nabla_t f(Ax, y)$. The Jacobian of $f(\cdot, y)$ at $t$ is

$$\nabla_t f(t, y) = y\, e^{\langle t, y\rangle} \in \mathbb{R}^{d\times d},$$

Let $\|\nabla_t f(t, y)\|_{L^2(\nu)} := \Big(\mathbb{E}_{Y\sim\nu}\|\nabla_t f(t, Y)\|_2^2\Big)^{1/2}$,

As derived in the proof of Lemma , for any mesurable function $g$,

$$\mathbb{E}[g(Y)e^{\langle t, y\rangle}] = e^{\frac{1}{2}t^\top \Sigma t}\mathbb{E}\left[g(Y + \Sigma t)\right].$$

Using this property,

$$\|\nabla_t f(t, y)\|_{L^2(\nu)} = e^{t^\top \Sigma t}\left(\mathbb{E}_{Y\sim\nu}\big[\|Y + \Sigma t\|_2^2\big]\right)^{1/2},$$

Use $t^\top \Sigma t = x^\top A^\top \Sigma A\, x \leqslant \|\Sigma^{1/2} A\|_2^2 \|x\|_2^2$, and take the supremum over $\|x\|_2 \leqslant R$:

$$\|L\|_{L^2(P)} := \sup_{\|x\|_2 \leqslant R} \|\nabla_x f_x(y)\|_{L^2(\nu)} \leqslant c'(\Sigma, A)\, \exp\big(R^2 \|\Sigma^{1/2} A\|_2^2\big),$$

where $c'(\Sigma, A, R)$ is a polynomial function in $\|A\|_2$, $\|\Sigma\|_2$, $R$, and $\mathrm{tr}(\Sigma)$. $\qquad\square$

**Lemma D.21** (Bound on envelope function for Gaussians). *For $Y \sim \mathcal{N}(0, \Sigma)$, and $x \in B_R$,*

$$\left(\mathbb{E}[F(Y)^2]\right)^{1/2} \leqslant \|\Sigma^{1/2}\|_2 \left(2^{d+1}d + \frac{\sqrt{2\pi}\, 2^{\frac{d}{2}+1}}{\Gamma(\frac{d}{2})}\left(2R\|\Sigma^{1/2} A\|_2\right)^{d+1}\right) e^{R^2\|\Sigma^{1/2} A\|_2^2}.$$

*Proof.* Write $Y := \Sigma^{1/2} Z$, where $Z \sim \mathcal{N}(0, I_d)$, and notice that $2R\|A^\top Y\|_2 \leqslant 2R\|\Sigma^{1/2} A\|_2 \|Z\|_2 := t\|Z\|_2$, where $t := 2R\|\Sigma^{1/2} A\|_2$. Then:

$$\mathbb{E}[F(Y)^2] = \mathbb{E}[\|Y\|_2^2\, e^{2R\|A^\top Y\|_2}] \leqslant \|\Sigma^{1/2}\|_2^2\, \mathbb{E}[\|Z\|_2^2 e^{t\|Z\|_2}] = \|\Sigma\|_2\, \mathbb{E}[\|Z\|_2^2 e^{t\|Z\|_2}].$$

As $\|Z\|_2 \sim \chi_d$, writing $c_d := \frac{2^{1-\frac{d}{2}}}{\Gamma(\frac{d}{2})}$, one can explicitly compute:

$$E[\|Z\|_2^2 e^{t\|Z\|_2}] = c_d \int_0^\infty r^{d+1} e^{-r^2/2} e^{tr}\, dr = c_d\, e^{t^2/2}\int_0^\infty r^{d+1} e^{-(r-t)^2/2}\, dr.$$

Making the change of variable $r - t = s$,

$$\int_0^\infty r^{d+1} e^{-(r-t)^2/2}\, dr = \int_{-t}^\infty (s+t)^{d+1} e^{-s^2/2}\, ds$$

$$\leqslant \int_{-\infty}^\infty (|s|+t)^{d+1} e^{-s^2/2}\, ds$$

$$\leqslant \int_{-\infty}^\infty \underbrace{2^d(|s|^{d+1} + t^{d+1})}_{(\star)} e^{-s^2/2}\, ds,$$

where we use $(a+b)^{d+1} \leqslant 2^d(a^{d+1}+b^{d+1})$ in $(\star)$.

Then use symmetry to compute

$$\int_{-\infty}^{\infty} |s|^{d+1}e^{-s^2/2}\,ds = 2\int_0^{\infty} s^{d+1}e^{-s^2/2}\,ds = 2\cdot 2^{\frac{d}{2}}\Gamma\left(\frac{d+2}{2}\right) = 2^{\frac{d+2}{2}}\Gamma\left(\frac{d+2}{2}\right).$$

Combining these results, we obtain that,

$$E[\|Z\|_2^2 e^{t\|Z\|_2}] \leqslant c_d\, 2^d(2^{\frac{d+2}{2}}\Gamma\left(\frac{d+2}{2}\right) + \sqrt{2\pi}\,t^{d+1})\,e^{t^2/2} = (2^{d+1}d + \frac{\sqrt{2\pi}\,2^{\frac{d}{2}+1}}{\Gamma(\frac{d}{2})}\,t^{d+1})\,e^{t^2/2}.$$

Finally, with $t = 2R\|\Sigma^{1/2}A\|_2$,

$$\left(\mathbb{E}[F(Y)^2]\right)^{1/2} \leqslant \|\Sigma^{1/2}\|_2\left(2^{d+1}d + \frac{\sqrt{2\pi}\,2^{\frac{d}{2}+1}}{\Gamma(\frac{d}{2})}\,(2R\|\Sigma^{1/2}A\|_2)^{d+1}\right)e^{R^2\|\Sigma^{1/2}A\|_2^2},$$

which is exactly the stated inequality. $\qquad\square$

Recall equation (10) for the numerator of attention in the proof of Theorem 4.2 in Appendix C.2,

$$\mathbb{E}[\|\mathbb{P}_n - \mathbb{P}\|_{\mathcal{F}}] \leqslant \frac{c\sqrt{d}}{\sqrt{n}}\left(R\,\mathbb{E}[\|L\|_{L^2(\mathbb{P}_n)}] + \mathbb{E}[\|F\|_{L^2(\mathbb{P}_n)}]\right).$$

Applying Lemma D.19 and Lemma D.21 yields,

$$\mathbb{E}[\|\mathbb{P}_n - \mathbb{P}\|_{\mathcal{F}}] \leqslant \frac{c\sqrt{d}}{\sqrt{n}}\left(R\,c(\Sigma, A) + C(d, \Sigma, A, R)\right)e^{R^2\|\Sigma^{1/2}A\|_2^2},$$

where $C(d, \Sigma, A, R) := \|\Sigma^{1/2}\|_2\,(2^{d+1}d + \frac{\sqrt{2\pi}\,2^{\frac{d}{2}+1}}{\Gamma(\frac{d}{2})}\,(2R\|\Sigma^{1/2}A\|_2)^{d+1})$.

To control the denominator of attention, applying Lemma D.20 to equation (12),

$$E[\|\mathbb{P}_n - \mathbb{P}\|_{\mathcal{F}'}] \leqslant \frac{c'(\Sigma, A)}{\delta\sqrt{n}}\,e^{R^2\|\Sigma^{1/2}A\|_2^2}. \tag{36}$$

Combining those results, the conclusion of part 3 in Appendix C.2 becomes,

$$\frac{\|N\|_2}{D} = \frac{\|\mathbb{E}[Ye^{\langle Ax, Y\rangle}]\|_2}{\mathbb{E}[e^{\langle Ax, Y\rangle}]} = \|\Sigma Ax\|_2 \leqslant \|\Sigma A\|_2\,R,\ \text{and}\ \|\frac{N_n}{D_n} - \frac{N}{D}\|_2 \leqslant \|V\|_2 \cdot \frac{c(\Sigma, A, R, d)}{\delta}\cdot\frac{1}{\sqrt{n}}\cdot e^{R^2\|\Sigma^{1/2}A\|_2^2},$$

where

$$q^{\mathcal{N}}_{(\Sigma, A, V, R, d, \delta)} = \frac{2\,c\|V\|_2\sqrt{d}}{\delta}\left(R\,c(\Sigma, A, R) + C(d, \Sigma, A, R) + \|\Sigma A\|_2\,R\,c'(\Sigma, A, R)\right).$$

### D.8.5. UNIFORM CONVERGENCE

**Proposition D.22** (Uniform convergence of the attention map for Gaussian tokens). *Let $X \sim \nu$ be centered and sub-Gaussian with parameter matrix $\Sigma \succ 0$. For any $R > 0$, $\delta > 0$, and invertible matrix $B$, there exists a constant $C > 0$, such that for $n \geqslant n_{min}(\delta, \Sigma, A, R) := 4e^{2R^2\|\Sigma^{1/2}AB^{-1}\|_2^2}\left(\frac{1}{\delta} - 1\right)$, with probability at least $1 - \delta$,*

$$\sup_{x\in B_R}\left\|f_n(x) - f(x)\right\|_2 \leqslant q^{\mathcal{N}}_{(\Sigma, A, V, R, d, \delta)}\cdot\frac{e^{R^2\|\Sigma^{1/2}AB^{-1}\|_2^2}}{\sqrt{n}}, \tag{37}$$

*where $q^{\mathcal{N}}_{(\Sigma, A, V, R, d, \delta)}$ is defined in Appendix D.8.4.*

The proof follows from the same argument as for Theorem C.11.

D.8.6. MEAN CONVERGENCE RATE

**Proposition D.23** (Mean convergence rate for Gaussian tokens)**.** *For Gaussian tokens using the same notation with* $H = \|\Sigma^{1/2}A\Sigma^{1/2}\|_2$ *we have,*

$$\left\| \mathbb{E}_n[f_n(x)] - \mathbb{E}[f(x)] \right\| = O\left( (\ln n)^{\frac{d+1}{2}} \, n^{-\frac{1}{2(1+H^2)}} \right).$$

The proof follows from applying Theorem C.11 with the same arguments as in Theorem 5.3.

# E. Improving the bounds

Our goal in this section is to show the following improvement of Theorem 4.2 in the compactly supported setting, where the exponential dependency of the bound in the radius $R$ is replaced by a polynomial dependency.

**Theorem E.1.** *Let* $Y \sim \nu$ *be centered and with compact support* $K \subset \mathbb{R}^d$*, such that* $\|Y\| \leqslant R_0$ *a.s. for some* $R_0 > 0$*. We assume that* $x \mapsto \sup_{y \in K}\langle Ax, y \rangle$ *is differentiable. Let* $\varphi : \mathbb{R}^d \to \mathbb{R}$ *defined by*

$$\varphi(t) = \sup_{y \in K}\langle t, y \rangle - \log \mathbb{E}[e^{\langle t, Y \rangle}].$$

*We assume*

$$\sup_{|t| \leqslant t_0} \varphi(t) \leqslant C_d \log(t_0) \tag{38}$$

*for some* $C_d > 0$ *as* $t_0 \to +\infty$*. Then for any* $R > 0$*,* $\delta > 0$*, there exists a constant* $C_d > 0$*, such that for* $n \geqslant n'_{min}(\delta, \Sigma, A, R)$*, with probability at least* $1 - \delta$*,*

$$\sup_{x \in B_R} \left\| f_n(x) - f(x) \right\|_2 \leqslant \frac{q'_{(\Sigma, A, V, R, R_0, d, \delta)}}{\sqrt{n}}, \tag{39}$$

*where* $q'_{(\Sigma, A, V, R, R_0, d, \delta)} := \|V\|_2 \frac{C\,\|\Sigma\|_2^{1/2}\,\sqrt{d}\,M}{\delta}\left[ (R+M)(\|\Sigma\|_2^{1/2}\,\sqrt{d}\,\|A\|_2 + \sup_{x \in B_R}\|\nabla_x \sup_{y \in K}\langle Ax, y \rangle\|_2) + 1 \right]$*, with* $M := (\|A\|_2 R R_0)^{C_d}$

*Remark* E.2. The condition (38) means that in each direction $v$, the pushforward of $\nu$ onto $\mathbb{R}v$ has a density close to its edges, and that this density has at most polynomial decay close to these edges. This is the case in most concrete examples, for example when $\nu$ is uniform on a regular open set $\Omega$, in any dimension. Note also that $x \mapsto \sup_{y \in K}\langle Ax, y \rangle$ is differentiable if, for instance $\partial K$ is $C^1$.

*Proof of Theorem E.1.* The main idea is to write the continuous attention map as

$$f(x) = \frac{\mathbb{E}[Y\,e^{\langle Ax, Y \rangle - \alpha(x)}]}{\mathbb{E}[e^{\langle Ax, Y \rangle - \alpha(x)}]}$$

with $\alpha(x)$ defined for any $x \in \mathbb{R}^d$ as

$$\alpha(x) = \sup_{y \in K}\langle Ax, y \rangle.$$

Then instead of (5) and (6) we let

$$f_x : y \mapsto y e^{\langle Ax, y \rangle - \alpha(x)} \tag{40}$$

and

$$F : y \mapsto \sup_{x \in B_R} \|f_x(y)\|_2 \leqslant \|y\|_2.$$

Lemma C.6 is unchanged. Lemma C.7 becomes:

**Lemma E.3** (Bound on Lipschitz constant for numerator)**.** *Let* $R > 0$*. Let* $f \in \mathcal{F} := \{f_x, x \in B_R\}$ *as defined in* (40)*. With an abuse of notation, denote* $f : (x, y) \mapsto y\,e^{\langle Ax, y \rangle - \alpha(x)}$*. Then,* $f(\cdot, y)$ *is Lipschitz with respect to* $x$*. Let* $y \mapsto L(y)$ *be*

*its Lipschitz constant. Consider $Y \sim \nu$, a sub-Gaussian random vector in $\mathbb{R}^d$ of matrix sub-Gaussian parameter $\Sigma$, and scalar sub-Gaussian parameter $\sigma := \sqrt{\|\Sigma\|_2}$. Then, $L$ verifies*

$$\|L\|_{L^2(\nu)} = \sup_{x \in B_R} \|\nabla_x f_x\|_{L^2(\nu)} \leqslant C\sigma\sqrt{d}\,(\sigma\sqrt{d}\|A\|_2 + \sup_{x \in B_R} \|\nabla\alpha(x)\|_2),$$

*for some constant $C > 0$.*

*Proof.* The proof is essentially the same as that of Lemma C.7. The only difference is that

$$\nabla_x f_x(y) = y(A^\top y - \nabla\alpha(x))^\top e^{\langle Ax, y \rangle - \alpha(x)}.$$

Using that $\alpha(x) \geqslant \langle Ax, y \rangle$ and the Cauchy-Schwarz inequality, we get

$$
\begin{aligned}
\|\nabla_x f_x\|_{L^2(\nu)} &\leqslant \sqrt{2\big(\mathbb{E}_{Y \sim \nu}[\|Y\|_2^4] \cdot \|A\|_2^2 + \mathbb{E}_{Y \sim \nu}[\|Y\|_2^2] \cdot \|\nabla\alpha(x)\|_2^2\big)} \\
&\leqslant \sqrt{2}\big(\mathbb{E}_{Y \sim \nu}[\|Y\|_2^4]^{1/2} \cdot \|A\|_2 + \mathbb{E}_{Y \sim \nu}[\|Y\|_2^2]^{1/2} \cdot \|\nabla\alpha(x)\|_2\big) \\
&\leqslant C\big(\sigma^2 d\|A\|_2 + \sigma\sqrt{d} \cdot \|\nabla\alpha(x)\|_2\big)
\end{aligned}
$$

Hence the statement follows. $\qquad\square$

Lemma C.8 is replaced by the bound $\mathbb{E}[F(Y)^2]^{1/2} \leqslant c\sigma\sqrt{d}$. Finally Proposition C.5 becomes:

**Proposition E.4** (Bound on numerator). *With probability at least $1 - \delta$,*

$$\|\mathbb{P}_n - \mathbb{P}\|_{\mathcal{F}} \leqslant C_1'(R) := c \cdot \frac{\sigma\sqrt{d}}{\delta\sqrt{n}} \cdot \big[R\,(\sigma\sqrt{d}\,\|A\|_2 + \sup_{x \in B_R} \|\nabla\alpha(x)\|_2) + 1\big].$$

This is a consequence of (10).

We turn to bounding the denominator $\mathbb{E}(e^{\langle Ax, y \rangle - \alpha(x)})$. For the same reasons we find,

**Proposition E.5** (Bound on denominator). *With probability at least $1 - \delta$,*

$$\|\mathbb{P}_n - \mathbb{P}\|_{\mathcal{F}'} \leqslant C_2'(R) := \frac{C_2}{\delta\sqrt{n}}(\sigma\sqrt{d}\|A\|_2 + \sup_{x \in B_R} \|\nabla\alpha(x)\|_2).$$

As in Part 3 of Appendix C.2, we can define

$$
\begin{aligned}
\mathcal{E}_1 &= \{\sup_{x \in B_R} \|N_n(x) - N(x)\|_2 \leqslant C_1'(R)\} \\
\mathcal{E}_2 &= \{\sup_{x \in B_R} |D_n(x) - D(x)| \leqslant C_2'(R)\}.
\end{aligned}
$$

Let

$$c_{\min} = \exp\left(\sup_{x \in B_R} \varphi(Ax)\right)$$

where $\varphi$ is introduced in the statement. Let also

$$
\begin{aligned}
\mathcal{E}_3 &= \{\sup_{x \in B_R} |D_n(x) - D(x)| \leqslant \frac{1}{2c_{\min}}\} \\
\mathcal{E}_4 &= \{\forall x \in B_R, D_n(x) \geqslant \frac{1}{2c_{\min}}\} = \{\inf_{x \in B_R} D_n(x) \geqslant \frac{1}{2c_{\min}}\}.
\end{aligned}
$$

In particular, with this definition of $c_{\min}$, we have for all $x \in B_R$,

$$D(x) = \mathbb{E}[e^{\langle Ax, y \rangle - \alpha(x)}] \geqslant \frac{1}{c_{\min}}.$$

By hypothesis, for $C_d > 0$,

$$\sup_{|t| \leqslant t_0} \varphi(t) \leqslant C_d \log(t_0)$$

Hence, for $t_0 = \|A\|_2 R R_0$, we have

$$c_{\min} \leqslant (\|A\|_2 R R_0)^{C_d}.$$

By Proposition E.5, if

$$C_2'(R) := \frac{C_2}{\delta_1 \sqrt{n}} (\sigma \sqrt{d} \|A\|_2 + \sup_{x \in B_R} \|\nabla \alpha(x)\|_2).$$

then the event

$$\mathcal{E}_2 = \{ \sup_{x \in B_R} |D_n(x) - D(x)| \leqslant C_2'(R) \}$$

is verified with probability $1 - \delta_1$. Then, for

$$n \geqslant n_{\min}'(\delta, \Sigma, A, R, d) := 4 \frac{C_2^2 (\|A\|_2 R R_0)^{2C_d}}{\delta_1^2} (\sigma \sqrt{d} \|A\|_2 + \sup_{x \in B_R} \|\nabla \alpha(x)\|_2)^2,$$

one has $C_2(R) \leqslant \frac{1}{2 c_{\min}}$, hence $\mathcal{E}_2 \subseteq \mathcal{E}_3$, and therefore $\mathbb{P}(\mathcal{E}_3) \geqslant \mathbb{P}(\mathcal{E}_2) \geqslant 1 - \delta_1$. On $\mathcal{E}_3$, the event $\mathcal{E}_4$ is also verified by the inclusion $\mathcal{E}_3 \subseteq \mathcal{E}_4$, which allows us to bound $D_n(x) \geqslant \frac{1}{2c_{\min}}$ uniformly in $x \in B_R$. Therefore, with probability $1 - \delta_1$, and with $n \geqslant n_{min}'(\delta_1, \Sigma, A, R, d)$,

$$\sup_{x \in B_R} \|\frac{N_n}{D_n} - \frac{N}{D}\|_2 \leqslant 2 c_{\min} ( \sup_{x \in B_R} \|N_n - N\|_2 + \sup_{x \in B_R} \|N\|_2 \cdot C_2'(R) c_{\min}). \tag{41}$$

where we used $\sup_{x \in B_R}(\|N\|_2 \cdot |D_n(x) - D(x)|) \leqslant \sup_{x \in B_R} \|N\|_2 \cdot \sup_{x \in B_R} |D_n(x) - D(x)|$, and $\sup_{x \in B_R} |D_n(x) - D(x)| \leqslant C_2'(R)$ on $\mathcal{E}_2$.

By Proposition C.5, if

$$C_1'(R) := c \cdot \frac{\sigma \sqrt{d}}{\delta \sqrt{n}} \cdot \left[ R (\sigma \sqrt{d} \|A\|_2 + \sup_{x \in B_R} \|\nabla \alpha(x)\|_2) + 1 \right]$$

the event

$$\mathcal{E}_1 = \{ \sup_{x \in B_R} \|N_n(x) - N(x)\|_2 \leqslant C_1'(R) \}$$

is verified with probability $1 - \delta_2$.

Then, the same derivation as before yields $\|N\|_2 = \|\mathbb{E}[Y e^{\langle Ax, Y \rangle - \alpha(x)}]\|_2 \leqslant c\sigma\sqrt{d}$. Combining events $\mathcal{E}_1$ and $\mathcal{E}_2$, and for $n \geqslant n_{min}'(\delta, \Sigma, A, R, d)$, we have with probability $1 - \delta$ (taking $\delta_1 = \delta_2 = \frac{\delta}{2}$),

$$\sup_{x \in B_R} \|\frac{N_n}{D_n} - \frac{N}{D}\|_2 \leqslant \|V\|_2 \cdot \frac{C \sigma \sqrt{d} (\|A\|_2 R R_0)^{C_d} \left[ (R + (\|A\|_2 R R_0)^{C_d})(\sigma \sqrt{d} \|A\|_2 + \sup_{x \in B_R} \|\nabla \alpha(x)\|_2) + 1 \right]}{\delta \sqrt{n}}.$$

where we recall that $\sigma = \sqrt{\|\Sigma\|_2}$. This concludes the proof of Theorem E.1. $\qquad \square$

