# OpenReview forum: "Token Sample Complexity of Attention"
_ICML.cc/2026/Conference — ICML 2026 regular_

### Official Review · Reviewer_dhGu · 2026-03-10

**Soundness:** 2
**Presentation:** 2
**Significance:** 3
**Originality:** 2
**Overall Recommendation:** 4
**Confidence:** 4

**Summary:**

This paper introduces the concept of token sample complexity to characterize how the output of a Transformer's attention mechanism converges to its theoretical infinite-token limit. It establishes convergence bounds at two primary levels: pointwise uniform convergence, and moment convergence.

**Compliance With Llm Reviewing Policy:**

Affirmed.

**Final Justification:**

I hold that interdependent tokens are significant to be considered. Anyway, the authors have solved my majority of concerns, considering the authors' work on additional experiments during this period, I have raised the score to 4.

**Key Questions For Authors:**

1. Regarding the i.i.d. assumption, it might mean that you can directly use the classic uniform convergence tool on training data to sequence tokens. Is there some theoretical difficulities in analyzing token sample complexity?

2. In real-world sequences, tokens are interdependent. If this dependency are truly modeled rather than simply assuming i.i.d., is it possible that the slower convergence could be derived as early as Section 4, without having to wait until the distributional analysis to discover this issue? I think this way is more important given that the i.i.d. assumption rarely holds true in practice.

3. Theorems 4.2 and 5.3 are derived based on dense attention, while the experiments utilize BigBird (sparse attention). How can we distinguish whether the slower convergence ($\beta < 0.5$) observed in the experiments is caused by attention horizon or token distribution your mentioned, or merely because BigBird's sparse approximation fails to rapidly converge to the true mean-field limit?

**Limitations:**

Yes.

**Strengths And Weaknesses:**

Advantages:
1. This paper introduces the novel concept of token sample complexity, in contrast to traditional statistical learning, which focuses on the sample complexity of training data.
2. This paper explains why the convergence rate of attention in practice is often lower than the ideal value through theoretical analysis.

Weaknesses:
1. The main theory is built upon the assumption that tokens are drawn i.i.d. from a distribution, however, actual natural language exhibits extremely strong contextual and Markovian dependencies.
2. The authors claim to prove a logarithmic convergence rate in the Hardmax limit. However, this proof (in Section 5.4) only applies to a highly simplified one-dimensional ($d=1$) Gaussian distribution.
3. The mismatch between theoretical models and experimental models (See Question 3).

---

> ### Author Rebuttal · Authors · 2026-03-31
>
> Thank you for reviewing our paper and for raising these important questions. We address each of your points below, hoping our responses will convince you to increase your score.
>
> ### Weaknesses
>
> > **W1.**
>
> We would like to emphasize that there are two distinct aspects to this concern.
>
> First, having tokens drawn from a distribution with Markovian dependencies does not
> contradict our theory: as long as tokens are sampled i.i.d. from that distribution
> — which is the case in our real-data experiments — the theorem applies directly.
>
> Second, we fully agree that in practice, subsampling is often structured, e.g.
> via windowing, and in that case the i.i.d. assumption no longer holds. Obtaining
> theoretical guarantees in this setting is significantly harder. We therefore address
> this concern empirically, using windowing-based subsampling on BigBird (see Figure~3
> of pdf in **[https://anonymous.4open.science/r/Additional_figures-0C79/additional_figures.pdf]**),
> and observe the same slow-down in convergence, suggesting that our theoretical
> predictions remain informative beyond the i.i.d. setting.
>
> > **W2.**
>
> We agree with this point. However, extending the result to higher-dimensional
> Gaussian settings appears to be beyond reach at present, as it would require
> nonstandard central limit theorems for maxima (rather than sums), which are outside
> the scope of this paper. The key difficulty is the following: in dimension $1$, the
> hardmax limit concentrates all attention on just two tokens — the maximum and
> minimum — whose attraction basins (positive and negative tokens respectively) are
> easy to characterize. In higher dimensions, these basins depend on the full
> configuration of the $n$ sampled tokens and are significantly harder to analyze,
> even for the simplest multivariate distributions. We will elaborate on this
> limitation in the final version.
>
> > **W3.**
>
> See response to question 3
>
> ### Questions
>
> > **Q1.**
>
> There are several non-trivial technical difficulties in analyzing token sample
> complexity that prevent us from simply using classical uniform convergence tools,
> and this even under the i.i.d. assumption. We will add some remark about this before the sketch of proof of the main theorems:
> - First, the attention map is a ratio of empirical averages whose kernel
> $e^{\langle Ax, y\rangle}$ is not uniformly Lipschitz, leading to a function
> class $\mathcal{F} = \{y \mapsto y e^{\langle Ax, y\rangle}, x \in B_R\}$ with
> unbounded envelope $F(y) = \|y\|_2 e^{R\|A^\top y\|_2}$, outside the scope of
> standard empirical process theory.
> - Furthermore, although non-standard uniform convergence over $\mathcal{F}$ can
> be guaranteed, no existing result yields quantitative bounds with explicit
> dependence on $H$, $\Sigma$ and $R$. We build these from scratch via Dudley's
> entropy integral and covering numbers, with an additional non-standard adaptation
> to the vector-valued setting on $\mathcal{G} = \mathcal{F} \times B_1$.
> - A further difficulty is that the empirical denominator can approach zero,
> precluding any uniform delta-method: numerator and denominator must be controlled
> separately and recombined uniformly in $x$.
> - Finally, in self-attention, queries, keys and values all originate from the same
> token set, making uniform control over all queries mandatory — a single poorly
> controlled query can dominate the aggregate and invalidate moment-level convergence.
>
> > **Q2.**
>
> This is an interesting point, but handling such dependencies theoretically would
> require moving beyond i.i.d. subsampling to structured schemes (such as
> window-based sampling), which is currently beyond the scope of our theory.
> Furthermore this slow-down is not a consequence of statistical dependence between
> tokens: it already appears in the i.i.d. setting of Theorem 4.2 itself, for
> realistic values of $R$ and $H$.
>
> We have conducted additional experiments on real data (Figure~3 of the link in W1),
> where such dependencies are naturally present, as well as with structured sampling
> schemes. The observed slowdown remains consistent with our theoretical predictions,
> suggesting that $H$ explains, to some extent, the magnitude of this effect —
> indicating that, despite the presence of nontrivial dependencies, our i.i.d.
> analysis still provides meaningful insight. We will include this discussion in the
> final version.
>
> > **Q3.**
>
> We thank the reviewer for raising this important and legitimate concern. To
> disentangle the two effects, we ran the same experiment on BERT-base-uncased (dense
> attention) (see Figures~1--2 for details in **[https://anonymous.4open.science/r/Additional_figures-0C79/additional_figures.pdf]**). We observe the same slow-down in convergence for both the mean and the covariance, confirming that the phenomenon is not an artifact of BigBird's sparse approximation, but really linked to $H$ and $\Sigma$. Note that the range of $H$ values differs slightly from the BigBird setting, as expected from the different model parameterization.

---

> > ### Author Rebuttal · Reviewer_dhGu · 2026-04-06
> >
> > Thank you! I hold that interdependent tokens are significant to be considered. Considering that the authors' work on additional experiments, I have raised the score to 4.

---

> > > ### Author Response · Authors · 2026-04-06
> > >
> > > Thank you very much for taking the time to read our responses carefully and for raising your score.

---

### Official Review · Reviewer_EdZE · 2026-03-12

**Soundness:** 4
**Presentation:** 3
**Significance:** 3
**Originality:** 4
**Overall Recommendation:** 4
**Confidence:** 2

**Summary:**

The paper introduces the notion of token sample complexity to characterize the convergence of the attention outputs as the length of the sequence approaches infinity, regarding finite-n attention as an approximation of a mean-field operator. The authors show that the attention map converges uniformly at a rate of $O(1/\sqrt{n})$ for sub-Gaussian tokens. And, with the analysis of the transformed token distribution, the convergence rate is $O(n^{-\beta})$ with $\beta<1/2$ depending on attention horizon and spectral properties of the token distribution. The analysis can also be extended to the hardmax attention regime to get the rate of $O(1/\sqrt{\ln n})$. In addition, the authors conduct both synthetic and real-world experiments to support the theoretical findings.

**Compliance With Llm Reviewing Policy:**

Affirmed.

**Final Justification:**

My concerns have been addressed. I will maintain my positive score.

**Key Questions For Authors:**

See Weaknesses.

**Limitations:**

The authors have discussed the limitations.

**Strengths And Weaknesses:**

Strengths:
1. The topic is interesting and novel. The paper introduces the notion of token sample complexity to investigate the complexity when the model is already trained.
2. Although the paper mainly focuses on the theory, it also provides empirical results on synthetic data and real-world data to support the theoretical findings.
3. The paper is well-written and presents solid and clear mathematical analysis and proofs.

Weaknesses:
1. The experimental validation on synthetic data is restricted to Gaussian distributions. The theory is built on the broader class of sub-Gaussian assumptions. To demonstrate the generality of the theory, the experiments should be extended to other sub-Gaussian distributions. In addition, how does the attention mechanism behave when the distribution violates the sub-Gaussian assumption? An analysis of heavy-tailed distributions would be a valuable extension.
2. The analysis is strictly confined to a single attention layer. In practice, the error from the first layer will interact with subsequent layers, normalization, and feed-forward networks. The authors can discuss more about what will happen on more complicated models.

---

> ### Author Rebuttal · Authors · 2026-03-31
>
> ### Comment
> Thank you very much for your positive review and your suggestions.  Please see our responses to your concerns below:
> ### Weaknesses
>
> > **W1.** The experimental validation on synthetic data is restricted to Gaussian distributions. The theory is built on the broader class of sub-Gaussian assumptions. To demonstrate the generality of the theory, the experiments should be extended to other sub-Gaussian distributions. In addition, how does the attention mechanism behave when the distribution violates the sub-Gaussian assumption?
>
> Thank you for raising these relevant points. We conducted additional experiments on uniform distributions — relevant in practice, e.g. after projection onto the unit ball — and observe the same slow-down in convergence as in the Gaussian case. We propose to add these to the appendix (see Figure~4 of **[https://anonymous.4open.science/r/Additional_figures-0C79/additional_figures.pdf]**).
>
> >  An analysis of heavy-tailed distributions would be a valuable extension.
>
> We will add in the final version the following simple example to explain why one cannot hope to even have convergence in this case. The key idea follows the same decomposition as in the proof of
> Proposition~5.4: the empirical attention mean splits into a fluctuation
> term $\tfrac{M_n^+ - M_n^-}{\sqrt{n}}$ and a mean-shift term
> $\tfrac{M_n^+ + M_n^-}{2}$. In the Gaussian case, symmetry forces the
> latter to zero. For heavy-tailed distributions, this term fails to
> converge, as we now show.
>
> Consider a distribution with density $\mu(x) = c_k(1+|x|)^{-k}$ with $k \geq 3$. Taking the same notation as in Appendix B.6, for $x > 0$:
> $$
> \mathbb{P}(M_n^+ > x) = 1 - \left(1 - c_k' x^{-k+1}\right)^n.
> $$
> Denoting by $p_n(\alpha,\beta)$ the probability that $M_n^+ \in [(\alpha n)^{1/(k-1)}, (\beta n)^{1/(k-1)}]$:
> $$
> p_n(\alpha,\beta) = \left(1-\frac{c_k'\alpha}{n}\right)^n -
> \left(1-\frac{c_k'\beta}{n}\right)^n \underset{n\to+\infty}{\longrightarrow}
> e^{-c_k'\alpha} - e^{-c_k'\beta}.
> $$
> In other words, $n^{-1/(k-1)}M_n^+$ converges in law to a non-trivial explicit distribution. Since $n^{-1/(k-1)}M_n^-$ converges to the same law,
> $$
> \frac{M_n^+ - M_n^-}{\sqrt{n}} \to 0 \quad \text{a.s.}
> $$
> However, $\frac{M_n^+ + M_n^-}{2}$ does not converge to $0$: it has strictly positive variance and the probability that it exceeds $1$ is positive. Therefore studying the rate of convergence makes no sense in this setting: $\mathbb{E}\,\Gamma_n$ does not converge. This shows that the sub-Gaussian assumption is not merely a technical convenience — it is necessary for the convergence of the attention output to hold.
>
> > **W2.** The analysis is strictly confined to a single attention layer. In practice, the error from the first layer will interact with subsequent layers, normalization, and feed-forward networks. The authors can discuss more about what will happen on more complicated models.
>
> We refer the reviewer to our response to Q2 of reviewer xt8m, where we derive a composition bound for a stack of $L$ attention layers, clarifying how errors across different layers accumulate.
>
> From a practical perspective, we have also conducted additional experiments showing that these theoretical findings persist in deep architectures, even under realistic (non-i.i.d.) subsampling patterns and for downstream classification tasks. We refer the reviewer to (Figures~5--7) of **[https://anonymous.4open.science/r/Additional_figures-0C79/additional_figures.pdf]**.

---

> > ### Author Rebuttal · Reviewer_EdZE · 2026-04-04
> >
> > Thank you for your response. I think my concerns have been addressed. I will maintain my positive score.

---

> > > ### Author Response · Authors · 2026-04-06
> > >
> > > Thank you for your response and for taking the time to read our replies carefully. We are glad that all your concerns have now been fully resolved, and we thank you again for your positive review.

---

### Official Review · Reviewer_xt8m · 2026-03-13

**Soundness:** 3
**Presentation:** 3
**Significance:** 3
**Originality:** 3
**Overall Recommendation:** 4
**Confidence:** 3

**Summary:**

This paper studies the token sample complexity of attention, namely, how fast attention computed on n tokens converges to its infinite-token limit as context length increases.
The authors formulate self-attention in a measure-theoretic framework, where a finite token sequence is represented by an empirical distribution and the infinite-context limit is represented by a continuous operator on the underlying token distribution.
They first establish a uniform convergence result for the attention map over a bounded query region, obtaining a 1/sqrt(n) rate under sub-Gaussian assumptions, although the associated constant scales exponentially with the query radius.
To derive more informative guarantees, the paper then analyzes convergence of moments of the transformed token distribution, including the mean and covariance.
The main theoretical result shows sub-parametric convergence rates of the form n^(-beta), where beta < 1/2 and depends on both the attention geometry and the spectral properties of the token distribution.
The paper also studies the hardmax regime and shows that, in a one-dimensional Gaussian setting, the convergence rate can degrade to 1/sqrt(log n).
Empirically, the authors validate the theory on both synthetic Gaussian data and real token embeddings extracted from BERT-base-uncased on large text corpora.
Overall, the paper provides a clean theoretical perspective on why increasing context length may yield substantially slower gains than the standard parametric intuition would suggest.

**Compliance With Llm Reviewing Policy:**

Affirmed.

**Key Questions For Authors:**

1. Can the authors connect the moment-convergence results more explicitly to downstream task-level error, for example through bounds on prediction drift or representation mismatch across layers?
2. How robust are the observed convergence behaviors to modeling components that are abstracted away in the theory, such as residual connections, LayerNorm, positional encodings, masking, and multi-layer composition?

**Limitations:**

yes

**Strengths And Weaknesses:**

Strengths:
1. The paper introduces a clear and novel problem formulation—token sample complexity at inference time—which is well motivated by the trend toward increasingly long context windows.
2. The theoretical development is well structured, with a useful distinction between uniform convergence of the attention map and convergence of downstream-relevant output moments.
3. The hardmax analysis is conceptually interesting, and the empirical section goes beyond purely synthetic validation by testing the theory on representations from a pretrained language model.

Weaknesses:
1. The scope is somewhat limited relative to modern Transformer practice, since the analysis focuses on attention operators and output moments rather than end-to-end task performance, multi-layer composition, or realistic autoregressive long-context settings.
2. Some of the assumptions appear restrictive, especially the dependence on sub-Gaussian or compact-support conditions and the one-dimensional Gaussian setup used in the hardmax analysis.
3. Although the experiments support the theoretical claims, they do not yet demonstrate that the derived convergence rates translate into predictive power for downstream accuracy, retrieval, or reasoning quality.

---

> ### Author Rebuttal · Authors · 2026-03-31
>
> ### Comment
>
> Thank you very much for your positive review and for the insightful remarks. All additional figures are in **[https://anonymous.4open.science/r/Additional_figures-0C79/additional_figures.pdf]**. Please see our responses below.
>
> ### Weaknesses
>
> > **W1.**
>
> We agree with this point, and this is inherent to the difficulty of the
> analysis — even the single-layer i.i.d. setting required substantial new tools.
> We nonetheless propose two concrete additions.
>
> On the theoretical side, we will add a corollary bounding the MSE
> $\mathbb{E}\|f(X) - f_n(X)\|^2$, following for free from our proof strategy,
> and refer to Q2 for the multi-layer composition bound.
>
> On the experimental side, we add a downstream classification experiment
> (Figures~5--7) that studies the behavior of our bounds outside the strict
> theoretical setting — on real Transformers, multiple layers, and realistic token
> distributions. The observed trends are consistent with our theoretical predictions,
> suggesting that the key phenomena identified by the theory persist in more
> realistic settings.
>
> > **W2.** Some of the assumptions [...] or compact-support conditions
>
> We would like to stress that the sub-Gaussian assumption is the least restrictive
> one can make: after LayerNorm/RMSNorm, tokens are bounded, hence sub-Gaussian,
> so our assumptions apply directly to real token distributions. The sub-Gaussian
> model goes beyond compact-support by introducing $\Sigma$, which captures data
> anisotropy and leads to strictly more informative bounds — as confirmed by our
> experiments. We will explain this in the final version.
>
> > and the one-dimensional Gaussian setup used in the hardmax analysis.
>
> We agree. Extending to higher dimensions requires nonstandard CLTs for maxima,
> which is outside the scope of this paper. We will elaborate in the final version.
>
> > **W3.**
>
> We provide a downstream classification experiment (Figures~5--7):
> BigBird-RoBERTa-base fine-tuned on arxiv-classification ($11$ classes,
> $86.4\%$ dense accuracy), varying $n$ i.i.d.\ keys per query from $64$ to $4096$.
> RMSE ($\beta=0.27$), pooling error ($\beta=0.30$), logit error ($\beta=0.37$) and
> disagreement rate ($\beta=0.46$) all follow $n^{-\beta}$ with $\beta<\tfrac12$,
> confirming the slow-down propagates to downstream quality. With $25\%$ of keys
> ($n=1101$), accuracy reaches $84.5\%$; doubling $n$ yields only $+0.7\%$.
>
> ### Questions
>
> > **Q1.**
>
> Thank you for raising this relevant and interesting point. The mean moment error
> is exactly the mean pooling error, directly used as input to the final layer in
> classification or clustering tasks. Controlling Lipschitz moments more generally
> is equivalent to controlling MMD distances between token distributions — relevant
> for domain adaptation, representation alignment, or hallucination detection.
>
> For tasks requiring a decoder or per-token predictions, our theory does not cover
> it, but it remains important to control individual embedding quality. We therefore
> propose to extend our results to the $L^2$/MSE loss $\mathbb{E}\|f_n(X)-f(X)\|^2$,
> which comes essentially for free from our proof strategy. We refer to our response
> to W3 for the numerical study.
>
> > **Q2.**
>
> Our experimental setup already includes residual connections, LayerNorm, and
> positional encodings, and the observed convergence behavior confirms that our
> theoretical predictions are robust to these components. From a theoretical
> standpoint, it suffices to consider already-normalized embeddings with positional
> encodings added — compact support is encompassed by our framework. Residual
> connections can also be handled but would make expressions more involved.
> We will add this to the final version.
>
> > masking
>
> We acknowledge that masking is a clear limitation, as it completely breaks the
> i.i.d. structure of the sampling.
>
> > and multi-layer composition?
>
> We will add in the final version that the control of the mean squared error can also be extended to deep compositions of layers, provided that the subsampling remains i.i.d. at each layer. More precisely, consider a stack of $L$ attention layers $F = f^L \circ \cdots \circ f^1$ and its sparse attention approximation $F_n = f_n^L \circ \cdots \circ f_n^1$, where each $f_n^\ell$ is constructed from $n$ i.i.d. subsampled tokens drawn from the layer-$\ell$ token distribution $\mu^\ell$. Applying the single-layer convergence bound at each depth yields $\lVert f^\ell - f_n^\ell \rVert_{L^2(\mu^\ell)} = O(n^{-\beta_\ell})$, where $0 < \beta_\ell < \tfrac{1}{2}$ is the convergence exponent associated with layer $\ell$. If all token distributions remain supported in a ball of radius $R$, each attention map is $O(R^2)$-Lipschitz. A composition argument then gives an overall error of order $O\bigl(\sum_{\ell=1}^L R^{2(L-\ell)} n^{-\beta_\ell}\bigr)$. Thus, local sketching errors accumulate through the network, with amplification governed by the Lipschitz constants of downstream attention layers. We plan to include this result in the appendix.

---

### Official Review · Reviewer_AxEP · 2026-03-13

**Soundness:** 2
**Presentation:** 2
**Significance:** 2
**Originality:** 3
**Overall Recommendation:** 4
**Confidence:** 3

**Summary:**

This paper studies token-sample complexity, which is the required number of tokens such that the attention output converges to its infinite-token limit. The authors first show the uniform convergence of the attention map as an exponential function of the radius $R$ over $\sqrt{n}$, and then prove the convergence of moments for the transformed token distribution as a polynomial of $R$ over $n\^\beta$. Experiments are conducted to support the theory.

**Compliance With Llm Reviewing Policy:**

Affirmed.

**Final Justification:**

The two new follow-up questions are well addressed. I have increased my rating to 4.

**Key Questions For Authors:**

1. Why do you consider $F=\\{y \mapsto ye\^{yAx}\\}$ (line 198)? How is it related to the attention structure in Eqn 1?

2. Why is Chinese Wikipedia anisotropic (line 427)? Can you cite any work to support this?

3. Can you summarize the technical challenges and novelty of the analysis?

**Limitations:**

Some limitation of this work is discussed in the Conclusion.

**Strengths And Weaknesses:**

Strengths:

1. The theoretical analysis is impressive and solid.

2. The studied topic is interesting and significant.

Weaknesses:

1. The insights from the theoretical conclusions are not clear or significant enough. First, Theorem 5.3 studies the convergence of the moment of the attention output. However, it is not clear what practical insights are from that result. The authors mention average pooling as an application, but it is only used in limited settings. Second, the convergence should highly depend on the characteristics of the attention map, while the analysis considers a mild condition of a compact support. Therefore, the bound could be loose and cannot lead to insights that are highly related to practical training and inference.

2. The writing needs improvement. The proof ideas of the main theorems are provided after each theorem. However, the overview is still too technical and does not include the discussion of insights or the novelty.

3. Some discussion is confusing. For example, in line 433, it says that $H$ encodes the anisotropy of the token distribution.  Why is it? The anisotropy should be reflected by the internal structure, like the condition number of $\Sigma$. From the definition of $H$, I cannot see this, and only the largest singular value of $\Sigma$ matters.

---

> ### Author Rebuttal · Authors · 2026-03-31
>
> ### Comment
>
> We thank the reviewer for the review and relevant remarks, which we will carefully address in the revised version. Please see
> our responses below, hoping they will convince you to increase your score.
>
> ### Weaknesses
>
> > **W1.** The [...]. First, [...], [...] settings.
>
> We agree that moment convergence is not suitable for all downstream tasks. However,
> controlling Fourier moments provides bounds on Maximum Mean Discrepancy (MMD)
> distances between token distributions — this metric is widely used in practice for
> domain adaptation, representation alignment, and hallucination detection.
>
> To broaden the applicability of our results, we will include a corollary bounding
> the MSE $\mathbb{E}\bigl|f_n(X_i) - f(X_i)\bigr|^2$ — relevant for tasks relying
> on individual token embeddings — whose proof follows for free from the same
> truncation argument.
>
> Finally, we provide a downstream classification experiment (Figures~5--7 of **[https://anonymous.4open.science/r/Additional_figures-0C79/additional_figures.pdf]**):
> BigBird-RoBERTa-base fine-tuned on arxiv-classification, for varying $n$ i.i.d.\ keys per query.
> RMSE, pooling error, logit error and disagreement rate — fraction of examples where
> sparse and dense predictions disagree — all follow $n^{-\beta}$ with $\beta <
> \tfrac{1}{2}$, confirming the slow-down reaches downstream quality. The
> sub-parametric convergence directly translates into insight on accuracy: with
> $25\%$ of keys ($n=1101$), accuracy reaches $84.5\%$; doubling $n$ yields only
> $+0.7\%$.
>
> > Second, [...] inference.
>
> It is true that compact-support analyses suffer from this limitation — but this is
> precisely what our sub-Gaussian framework overcomes: it introduces $\Sigma$ and $H
> = \|\Sigma^{1/2} A\|_2$ to account for data anisotropy and its interaction with
> the attention map, yielding a convergence rate that depends explicitly on $H$.
> Empirically, the bounds are tight on synthetic data, $H$ governs convergence on
> real data, and our new downstream experiment (see **[https://anonymous.4open.science/r/Additional_figures-0C79/additional_figures.pdf]**) shows how attention
> sparsification affects inference accuracy in practice.
>
> > **W2.**
>
> We propose adding the following paragraphs after the main theorems.
>
> **Theorem 5.3** establishes a subparametric convergence rate with exponent
> $\beta < 1/2$ depending on $H$, providing the first quantitative characterization
> of how sparse attention is harder to approximate with few tokens. $H$ governs
> the approximation rate even for downstream tasks, giving concrete insight on the
> required number of tokens per layer to achieve a desired level of accuracy.
>
> > **W3.**
>
> Indeed, we fully agree: it is $\Sigma$, not $H$, that encodes the anisotropy.
> $H = \|\Sigma^{1/2}A\|_2$ captures the alignment between the eigenspace of the
> token distribution and $A$, encoding the interaction between data geometry and
> the model, not anisotropy alone. We will clarify this.
>
> ### Questions
>
> > **Q1.**
>
> This class arises naturally from rewriting self-attention as a ratio of empirical
> averages: the numerator is $N(x) = V \int y e^{\langle Ax,y\rangle} d\mu(y)$
> and the denominator $D(x) = \int e^{\langle Ax,y\rangle} d\mu(y)$. The core
> empirical-process object is $\mathcal{F} = \{y \mapsto y e^{\langle Ax,y\rangle}\}$
> that must be controlled uniformly in $x$.
>
> > **Q2.** Why is Chinese Wikipedia anisotropic (line 427)? Can you cite any work to support this?
>
> This was a minor empirical remark for this particular dataset and embeddings.
> What we wanted to convey is that, for a fixed model, two datasets with similar
> $H$ can still exhibit different convergence rates if their token distributions
> have different anisotropy — i.e. different maximum eigenvalues of $\Sigma$.
> We will clarify this.
>
> > **Q3.** Can you summarize the technical challenges and novelty of the analysis?
>
> We will add the following two paragraphs in the final version:
>
> At a conceptual level, our main contribution is to introduce, for the first time,
> a statistical framework that quantitatively captures the difficulty of approximating
> attention maps for large context. Our analysis brings out $H$ and $\Psi$ as key
> quantities linking convergence rates to data anisotropy and model parameters,
> confirmed by downstream experiments (Figures~5--7 of **[https://anonymous.4open.science/r/Additional_figures-0C79/additional_figures.pdf]**).
>
> At a mathematical level, our contributions do not follow from classical PAC theory.
> This is not only because token-level sample complexity differs conceptually from
> standard settings, but also because the SoftMax operator is not uniformly
> Lipschitz, requiring a dedicated treatment. We develop quantitative bounds from
> first principles: a vector-valued extension of Dudley's entropy integral on
> $\mathcal{G} = \mathcal{F} \times B_1$, control of the unbounded envelope
> $F(y) = \|y\|_2 e^{R\|A^\top y\|_2}$ via high-order moments, and a
> numerator–denominator decomposition avoiding a uniform delta method.

---

> > ### Author Rebuttal · Reviewer_AxEP · 2026-04-02
> >
> > Thank you for your response. Most of the questions are addressed. However, I am still not clear how you derive $y\mapsto y \mapsto y e^{\langle Ax,y\rangle}$ in Q1. For Q2, do you mean the statement that Chinese Wikipedia is anisotropic is a speculation or a corollary, since you do not provide citations? I think previously, you were restricted by the space limit of the initial rebuttal. I hope you can now provide a detailed answer to Q1 and Q2 in this round.

---

> > > ### Author Response · Authors · 2026-04-06
> > >
> > > Thank you very much for your response and for taking the time to read our responses. We were indeed limited by the space limit before. Please find below more expanded explanations:
> > >
> > > >Q1
> > >
> > > The main point is that $y \mapsto y e^{\langle Ax,y\rangle}$ is not introduced ad hoc: it is exactly the numerator kernel of attention, viewed as a function of the sampled token $y$, once self-attention is rewritten as an empirical average and the numerator is controlled uniformly in the query $x$.
> > >
> > > To be more precise, Eq.~(1) corresponds to the **self-attention** case: given a context $(y_1,\dots,y_n)$, one picks a token $y_i$ as query and computes its attention with respect to the same context. More generally, before specializing to $x=y_i$, we consider the attention map for an arbitrary deterministic query $x\in\mathbb{R}^d$:
> > >
> > > $$
> > > f_n(x)=V \frac{\sum_{j=1}^n y_j e^{\langle Ax,y_j\rangle}}{\sum_{j=1}^n e^{\langle Ax,y_j\rangle}}.
> > > $$
> > >
> > > Eq.~(1) is recovered by evaluating this map at $x=y_i$.
> > >
> > > For fixed $x$, the numerator is a weighted average of the values $v_j=Vy_j$. Since the value matrix $V$ is fixed, we factor it out and study
> > > $$
> > > \widetilde N_n(x)=\frac{1}{n}\sum_{j=1}^n y_j e^{\langle Ax,y_j\rangle},
> > > \qquad
> > > \widetilde N(x)=\mathbb{E}\left[Y e^{\langle Ax,Y\rangle}\right]
> > > =\int y e^{\langle Ax,y\rangle}\,d\mu(y).
> > > $$
> > > Thus the relevant integrand is exactly $y\mapsto y e^{\langle Ax,y\rangle}$.
> > >
> > > The first step of the proof is to control how fast $\widetilde N_n(x)$ converges to $\widetilde N(x)$, uniformly in $x$. On a ball $B_R$, this means bounding
> > >
> > > $$\sup_{x\in B_R}\left\lVert \widetilde N_n(x)-\widetilde N(x)\right\rVert_2 = \sup_{x\in B_R}\left\lVert \frac{1}{n}\sum_{j=1}^n y_j e^{\langle Ax,y_j\rangle} - \mathbb{E} \left[Y e^{\langle Ax,Y\rangle}\right] \right\rVert_2.
> > > $$
> > >
> > > This is a uniform empirical-process problem indexed by $x\in B_R$. Accordingly, for each fixed $x$ we introduce
> > > $$
> > > g_x(y)=y e^{\langle Ax,y\rangle},
> > > $$
> > >
> > > and the class
> > > $$
> > > \mathcal{F} = \lbrace g_x:x \mapsto y e^{\langle Ax,y\rangle},x\in B_R \rbrace.
> > > $$
> > >
> > > Then the previous display becomes
> > > $$
> > > \sup_{g_x\in\mathcal F}
> > > \left\lVert
> > > \frac1n\sum_{j=1}^n g_x(y_j)-\mathbb E[g_x(Y)]
> > > \right\rVert_2.
> > > $$
> > >
> > > So $\mathcal F$ is exactly the family of numerator integrands arising when attention is rewritten as a ratio of empirical averages and the numerator is studied uniformly over $x\in B_R$. The denominator is handled analogously with the scalar class $\lbrace h_x: y \mapsto e^{\langle Ax,y\rangle},\ x\in B_R \rbrace$, and the two bounds are then recombined to control $f_n(x)-f(x)$.
> > >
> > > Finally, this uniform control is needed precisely because of the self-attention structure: in self-attention, the query is itself one of the random tokens of the context. One therefore first proves a bound uniform in $x$, and then evaluates it at the random queries $x=y_i$.
> > >
> > > We will incorporate this clarification in the final version to make the passage from Eq.~(1) to the class $\mathcal F$ explicit.
> > >
> > > >Q2
> > >
> > > Thank you for pointing this out. This statement was indeed not meant as a corollary of the theory, nor as a general claim about Chinese Wikipedia supported by an external citation. Rather, it was intended as an **empirical observation** based on our own measurements for the specific datasets considered in our experiments. In particular, for the Chinese Wikipedia dataset, the covariance matrix $\Sigma$ of the token representations appears more anisotropic than for the other datasets we studied.
> > >
> > > To make this point more precise, we have added a figure - in our previous link - showing, across layers, the effective rank of the covariance matrix for all datasets (see Figure 8 page 7 of additional_figures.pdf:  **[https://anonymous.4open.science/r/Additional_figures-0C79/additional_figures.pdf]**). Since the effective rank is directly related to anisotropy, this figure makes the empirical claim explicit.
> > >
> > > What we intended to convey is that our bounds are sensitive to the geometry of the token distribution through $\Sigma$, in particular to the distribution of its eigenvalues, and hence to anisotropy. This dependence is also reflected in the horizon
> > > $$
> > > H=\left\lVert \Sigma^{1/2}A \right\rVert_2,
> > > $$
> > > which does not measure anisotropy alone, but rather the interaction between the token distribution and the attention parameter $A$. More precisely, $H$ captures how the dominant eigenspaces of $\Sigma$ align with $A$, which is more informative than the looser decoupled bound $\left\lVert \Sigma^{1/2}\right\rVert_2 \cdot \left\lVert A \right\rVert_2$. On the other hand, our constants are also sensitive to the scale of the largest eigenvalues of $\Sigma$, which is also significantly higher for the Chinese Wikipedia dataset in our experiments.
> > >
> > > We will revise the wording in the paper to make it fully clear that this point is an empirical observation, not a formal corollary.
> > >
> > > Note: if the link takes time to load, please click on "additional_figures.pdf" on the left.

---

### Decision · Program_Chairs · 2026-04-30

**Decision:**

Accept (regular)

**Comment:**

This paper introduces the notion of token sample complexity and studies how pretrained attention converges as the context length approaches infinity. It provides theoretical convergence guarantees at both the level of the attention map and the moments of the transformed token distribution under sub-Gaussian assumptions.

Reviewers agree that the work is novel and interesting. Solid theoretical analysis and empirical results are provided to support the main claims. During the rebuttal and discussion phases, the authors addressed many of the reviewers’ concerns and provided additional experiments and clarifications. Several reviewers updated their assessments toward acceptance.